

**The high-resolution version of TM5-MP for optimised satellite retrievals: Description and Validation.**

**J. E. Williams**[1]**, K. F. Boersma**[1,2]**, P. Le Sager**[1]**, W. W. Verstraeten**[1,2,3]

[1] {KNMI, De Bilt, The Netherlands}
[2] {Meteorology and Air Quality Group, Wageningen University, Wageningen, The Netherlands}
[3] {KMI, Ukkel, Brussels, Belgium}

**Abstract**

We provide a comprehensive description of the high-resolution version of the TM5-MP global Chemistry-Transport Model, which is to be employed for deriving highly resolved vertical profiles of nitrogen dioxide ($NO_2$), formaldehyde ($CH_2O$), and sulphur dioxide ($SO_2$) for use in satellite retrievals from platforms such as the Ozone Monitoring Instrument (OMI) and the Sentinel-5 Precursor, the TROPOspheric Monitoring Instrument (tropOMI). Comparing simulations conducted at horizontal resolutions of 3° x 2° and 1° x 1° reveals differences of ±20% exist in the global seasonal distribution of $^{222}Rn$, being larger near specific coastal locations and tropical oceans. For tropospheric ozone ($O_3$), analysis of the chemical budget terms shows that the impact on globally integrated photolysis rates is rather low, in spite of the higher spatial variability of meteorological data fields from ERA-Interim at 1° x 1°. Surface concentrations of $O_3$ in high-$NO_x$ regions decrease between 5-10% at 1° x 1° due to a reduction in $NO_x$ recycling terms and an increase in the associated titration term of $O_3$ by NO. At 1° x 1°, the net global stratosphere-troposphere exchange of $O_3$ decreases by ~7%, with an associated shift in the hemispheric gradient. By comparing NO, $NO_2$, $HNO_3$ and PAN profiles against measurement composites, we show that TM5-MP captures the vertical distribution of $NO_x$ and long-lived $NO_x$ reservoirs at background locations, again with modest changes at 1° x 1°. Comparing monthly mean distributions in lightning $NO_x$ and applying ERA-interim convective mass fluxes, we show that the vertical re-distribution of lightning $NO_x$ changes with enhanced release of $NO_x$ in the upper troposphere. We show that surface mixing ratios in both NO and $NO_2$ are generally underestimated in both low and high $NO_x$ scenarios. For Europe, a negative bias exists for [NO] at the surface across the whole domain, with lower biases at 1° x 1° at only ~20% of sites. For $NO_2$, biases are more variable, with lower (higher) biases at 1° x 1° occurring at ~35% (~20%) of sites, with the remainder showing little change. For $CH_2O$, the impact of higher resolution on the chemical budget terms is rather modest, with changes less than 5%. The simulated vertical distribution of $CH_2O$ agrees reasonably well with measurements in pristine locations, although column-integrated values are generally underestimated relative to satellite measurements in polluted regions. For $SO_2$, the performance at 1° x 1° is principally governed by the quality of the emission inventory, with limited improvements in the site specific biases with most showing no significant improvement. For the vertical column, improvements near strong source regions occur which reduce the biases in the integrated column. For remote regions missing biogenic source terms are inferred.

## 1.    Introduction

One application of Chemistry Transport Models (CTM) is to provide accurate vertical and horizontal global distributions of trace gases such as ozone ($O_3$), nitrogen dioxide ($NO_2$), sulphur dioxide ($SO_2$) and formaldehyde ($CH_2O$) that are used as *a-priori* best-guesses in the retrievals of tropospheric abundances from instruments mounted on Earth-orbiting satellites such as the Tropospheric Emission Sounder (TES; Worden et al., 2007) Global Ozone Monitoring Experiment (GOME), SCanning Imaging Absorption spectroMeter for Atmospheric CHartographY (SCIAMACHY; De Smedt et al., 2008), the Ozone Monitoring Instrument (OMI; Boersma et al., 2011), and GOME-2 (Valks et al., 2011). To date, although high-resolution regional models have been employed for selected regions such as the US and Europe (e.g. Russell et al., 2011; Zhou et al., 2012; Vinken et al., 2014), at the global scale the CTMs resolutions employed are still rather coarse (between 1.1-4.0º latitude and 1.1-6º longitude), resulting in 'footprints' which aggregate hundreds of kilometers in area. This has limitations as the resulting total columns are sensitive to topography, surface albedo and the shape of the *a-priori* vertical profiles themselves. Using rather coarse resolution leads to substantial errors in the retrievals (e.g. Boersma et al., 2007; Heckel et al., 2011; Russell et al., 2011) and imposes limitations towards capturing the regional scale variability in short-lived trace gas abundances observed from high-resolution satellite instruments such as OMI.

This lack of spatial detail is particularly relevant for situations where strong spatio-temporal variability in the vertical distribution of $NO_2$, $SO_2$, and $CH_2O$ can be expected. Examples include shipping lanes in the relatively unpolluted marine boundary layer (e.g. Vinken et al., 2014) and coal-fired power plant $SO_2$ pollution (e.g. Fioletov et al., 2015). Moreover, during the day the local lifetime and mixing ratios of trace gases such as nitric oxide (NO) and $NO_2$ are critically dependent on a host of variables e.g. temperature, surface albedo, cloud cover (via photolysis), chemical conversion (i.e. $NO/NO_2$ ratio) and the extent of mixing by convective upwelling (i.e. land type) and advective transport. Thus, the information provided for the retrievals is affected by the coarsening of the high-resolution meteorological data used to drive the CTM. Recently, Heckel et al. (2011) demonstrated that there is an associated uncertainty of ~2 using a-priori data from a global CTM rather than a regional CTM, principally due to loss of spatial information. Two other studies focusing on the impact of horizontal resolution on the retrieval of vertical column densities of $NO_2$ suggested that errors of up to ~50% exist (Yamaji et al., 2014; Lin et al., 2014). This problem becomes accentuated for the next generation of Earth-orbiting satellites such as the Tropospheric Monitoring Instrument (tropOMI), which has a smaller footprint compared to its predecessors (Veefkind et al., 2012). Applications of TM5 include the retrieval of $NO_2$, $CH_2O$, and $SO_2$ column densities from OMI and tropOMI (e.g. van Geffen et al., 2016), where studies related to the influence of horizontal resolution have been limited principally to $NO_2$.

The dominant tropospheric loss terms for $CH_2O$ are photolysis and scavenging into cloud droplets (wet deposition; Jacob, 2000). Thus the atmospheric lifetime of $CH_2O$ is highly sensitive to the extent of cloud cover and the vertical profiles of the photolysis rates. A dominant application of $CH_2O$ retrievals is to provide constraints on tropical and sub-tropical isoprene emission fluxes (e.g. Palmer et al., 2006; Stavrakou et al., 2009; Marais et al., 2012). The resulting emission estimates are highly sensitive to the stoichiometric yield of $CH_2O$ from isoprene oxidation, the chemical lifetime of $CH_2O$ and spatial differences in land cover. Other applications include estimating emissions released during Biomass Burning (BB) episodes (Gonzi et al., 2011),

whose spatial location is also smeared via coarsening in TM5-MP. For $SO_2$, which predominantly originates
from point sources, an adequate spatial distribution of such sources is crucial for estimating accurate biases in
existing emission inventories.
In this paper we provide a comprehensive description of the global, high-resolution 1° x 1° version of the TM5
CTM tailored for the application of satellite retrievals (hereafter referred to as TM5-MP). In Sect. 2 we give
details related to the modifications which have been made to the TM5 model compared to previous versions,
the emission inventories employed, updates that have been made to the modified CB05 chemical mechanism,
the stratospheric boundary conditions, the photolysis scheme, the heterogeneous conversion and the overall
model structure. In Sect. 3 we analyse the impact on convective and advective transport of trace species from
the BL of both increased horizontal resolution and use of ERA-interim convective mass-fluxes, as derived
using radon ($^{222}$Rn) distributions. In Sect. 4 we investigate the effects on regional and global photolysis
frequencies. In Sects. 5-9 we examine the differences in the vertical and horizontal distributions of
tropospheric $O_3$, $NO_x$, Lightning induced $NO_x$, N-containing species (i.e. nitric acid ($HNO_3$), peroxy-acetyl-
nitrate (PAN) and lumped organic nitrates (ORGNTR)), $CH_2O$ and $SO_2$, where we make comparisons against
both surface and aircraft measurements to validate mixing ratios. Finally, in Sect. 10, we present our
conclusions.

**2    Description of TM5-MP**

Previous versions of TM5 (TM5-chem-v3.0, Huijnen et al., 2010) included a two-way nested zooming option
as described by Krol et al. (2005). This option allowed high-resolution simulations to be performed over any
pre-defined regional domain, with boundary conditions being determined by the global simulation at coarser
resolution. Typically, global simulations at 3° x 2° with zoom regions at 1° x 1° were performed to alleviate the
long runtime of a global 1° x 1° run. In the new version of TM5 (hereafter referred to as TM5-MP; the
massively parallel version), the usage of the Message Passing Interface (MPI) has been totally rewritten. Zoom
regions are no longer available, but data sets are distributed along longitudes and latitudes, instead of model
levels and tracers. The advantages of that overhaul towards domain decomposition are a smaller memory
requirement and the possibility to use more processors making global 1° x 1° simulations feasible in terms of
runtime and affordable in terms of computing resources. A TM5-MP global 3° x 2° (1° x 1°) run is ~6 (~ 20)
times faster than the previous version of TM5 (Huijnen et al., 2010) for similar resources. The following model
description pertains to both 3° x 2° and 1° x 1° simulations discussed in this manuscript.
Here we provide a comprehensive description of the modifications and updates introduced into TM5-MP
compared to TM5 v3.0 (Huijnen et al., 2010). The model is driven using the ERA-interim meteorological re-
analysis (Dee et al., 2011) and updated every 3 hours, with interpolation of fields for the intermediate time
periods. Although TM5-MP can adopt all 60 vertical levels provided by the ECMWF ERA-Interim reanalysis,
we employ 34 vertical levels for this study with higher resolution in the troposphere and Upper Troposphere-
Lower Stratosphere (UTLS). Convective mass-fluxes and detrainment rates are taken from ERA-interim
dataset to describe the updraft velocities from the Boundary Layer (BL) into the free troposphere, which
replaces the parameterization of Tiedtke (1989) used in previous versions. The vertical diffusion in the free
troposphere is calculated according to Louis (1979), and in the BL by the approach of Holtslag and Boville
(1993). Diurnal variability in the BL height is determined using the parameterization of Vogelezang and
Holtslag (1996). We use the first-order moments scheme with an iterative time-step to prevent too much mass
being transported out of any particular grid-cell during the time-step according to the preservation of the
Courant-Friedrichs-Lewy (CFL) criterium (Bregman et al., 2003), which is especially relevant when reducing
the size of grid-cells as done here.
The gas-phase chemistry in TM5-MP is described by an expanded version of the modified CB05 chemical
mechanism (hereafter mCB05; Williams et al., 2013). We have placed emphasis on updating and expanding
the fast $NO_x$ chemistry to account for an accurate partitioning of nitrogen for higher $NO_x$ regimes than those
occurring at coarser horizontal resolutions. All reaction rate data is now taken from the latest IUPAC
recommendations (sited at http://iupac.pole-ether.fr/; last access June 2016) using updated formulations for
third-body collisions, where the rate data for fast $NO_x$ and $CH_2O$ chemistry is given in Table 1. This includes
the recent update to the formation rate of $HNO_3$ determined by Möllner et al. (2010). The most relevant
modifications are: (i) The yield of $CH_2O$, methanol ($CH_3OH$) and the hydro-peroxy radical ($HO_2$) from the
self-termination of the methyl-peroxy radical ($CH_3O_2$) is increased according to Yarwood et al. (2005), (ii) the
direct formation of $CH_2O$ from the reaction of $CH_3O_2 + HO_2$ is added using the temperature dependent
branching ratio defined in Atkinson et al. (2004), (iii) the production of $HNO_3$ during the oxidation of di-
methyl sulphide (DMS) by the $NO_3$ is now included, (iv) explicit organic peroxy radicals have been introduced
as products from the oxidation of propene ($C_3H_6$) and propane ($C_3H_8$) by OH, which are lost by either the
reaction with nitric oxide (NO) or $HO_2$ allowing the in-situ chemical formation of acetone ($CH_3COCH_3$) and
higher aldehydes (ALD2), respectively, following the stoichiometry given in Emmons et al. (2010), (v) a
second product channel for $N_2O_5$ photolysis is added producing NO, (vi) the formation and photo-dissociation
of HONO has been included, (vi) the formation and transport of methyl peroxy nitrate ($CH_3O_2NO_2$) is also
included (Browne et al., 2011), and (vii) modifications to the gas-phase chemistry involving $NH_3$ have been
introduced following the stoichiometry given in Hauglestaine et al. (2014). This version of the modified CB05
chemical mechanism is hereafter referred to as mCB05v2.
The calculation of height resolved photolysis rates ($J$ values) is performed using a tailored version of the
Modified Band Approach (MBA). The implementation and performance of this parameterization in TM5 has
been fully described in Williams et al. (2012). For the calculation of the height-resolved actinic fluxes at the
seven specific wavelengths used for calculating the $J$ values (these being 205.1nm, 287.9nm, 302.0nm,
311.0nm, 326.5nm, 385.0nm and 610.0nm), the 2-stream radiative transfer solver of Zdunkowski et al. (1980)
is embedded into TM5-MP. Details regarding the parameterizations used to account for the scattering and
absorption introduced by gaseous molecules, aerosols and clouds the reader is referred to Williams et al.
(2012). For aerosols, the climatology of Shettle and Fenn (1979) is included. The calculation of the effective
radius ($r_{eff}$) of cloud droplets is now performed using the approach of Martin et al. (1994), where different
parameter values are used for over the land and ocean using cloud condensation nuclei concentrations of 40
and 900, respectively. Due to potentially erroneous values at low horizontal resolution, we weight the final $r_{eff}$
value using the land fraction in each grid-cell. We apply limits between 4-16μm on the resulting $r_{eff}$ values.
This improves the representation of the scattering component due to cloud droplets used for the calculation of
the actinic flux in the lower troposphere (LT; not shown). For the scattering effects from cloud droplets, we
subsequently downsize the physical $r_{eff}$ by ~0.5-2µm to account for the relationship between the optical and
physical $r_{eff}$ values.
For aerosols, an aerosol scheme is available for use within TM5-MP (aan den Brugh et al., 2010), but we
choose not to use it for the purpose of satellite retrievals due to the extra computational expense needed when
performing high resolution simulations that would potentially hinder operational use. We acknowledge that the
description of aerosols in this study is rather crude and increasing scattering could have an impact under
instances of low cloud coverage. For the application of TM5-MP towards satellite retrieval, it is preferable to
use any advancements in computational performance on further increases in the horizontal resolution
employed. Therefore it is not currently envisaged that a full description of aerosol processes will be included
during operational satellite retrievals.
However, heterogeneous conversion processes still need the description of the total reactive Surface Area
Density (SAD) from aerosols. In TM5-MP this is assumed as the cumulative value of contributions from
sulphate, nitrate, ammonium and methane sulphonic acid as calculated by the EQuilibrium Simplified Aerosol
Model (EQSAM) approach (Metzer et al., 2002), thus the secondary organic aerosol component is not
included. The distribution of these aerosol species is calculated online and coupled to the respective gaseous
precursors. The density of each aerosol type (1.7 g/cm$^3$) and $r_{eff}$ (of between 0.18-0.2µm) is prescribed as in
Huijnen et al. (2014). Swelling at higher relative humidities ($> 70\%$) is crudely accounted for by increasing $r_{eff}$
between 0.25-0.27µm. The contributions due to sea-salt, black carbon and organic carbon towards
heterogeneous loss are not accounted for. Temperature dependent gas-phase diffusion co-efficients ($D_g$) are
used in the derivation of the pseudo first-order heterogeneous rate constants based on the theory of Schwartz
180 (1986).

For $N_2O_5$, the uptake coefficient ($\gamma$) is calculated using the parameterization of Evans and Jacob (2005),
therefore dependent on both temperature and relative humidity. Once a surface reaction with $H_2O$ occurs two
molecules of $HNO_3$ are formed. No uptake on cirrus particles is included for $HNO_3$, which can lead to de-
nitrification of the upper troposphere (Lawrence and Crutzen, 1998; von Kuhlmann and Lawrence, 2006). For
$HO_2$ we adopt a fixed $\gamma_{HO2}= 0.06$ across all aerosol types as taken from Abbatt et al. (2012) and for $NO_3$ we
adopt a fixed $\gamma_{NO3} = 10^{-3}$ as recommended by Jacob (2000). For $HO_2$, heterogeneous conversion forms 0.5
molecules of Hydrogen Peroxide ($H_2O_2$), whereas for $NO_3$ it forms one molecule of $HNO_3$ following Emmons
et al. (2010). For the SAD associated with cloud droplets we use the $r_{eff}$ values that are calculated by Martin et
al. (1994) thus maintaining consistency between the size of the cloud droplets used for the scattering
component in the calculation of $J$ values and heterogeneous loss rates on the clouds. By using the ECMWF
cloud fraction for each respective grid-cell, we assume that instantaneous mixing throughout the grid-cell does
not occur in order to avoid exaggerated conversion rates on cloud surfaces.
As TM5-MP contains no explicit stratospheric chemistry, we apply constraints above the tropopause to ensure
realistic Stratosphere-Troposphere Exchange (STE) of $O_3$ and for constraining the incoming radiation reaching
the troposphere needed for the MBA (Williams et al., 2012). For stratospheric $O_3$, we use total column values
derived from the assimilation of satellite observations as provided in the improved version of the Multi-Sensor
Re-analysis (MSR, van der A., 2010), which is vertically distributed according to the climatology of Fortuin
and Kelder (1998). Three distinct zonal bands are used for nudging the stratospheric $O_3$ fields, these being
30°S-30°N, 30-66°S/N and > 66°S/N, where nudging occurs at pressure levels <45hPa, <95hPa and <120hPa,
with relaxation times of 2.5 days, 3 days and 4 days, respectively.
For stratospheric $CH_4$ we use the monthly 2D climatological fields provided by Grooß and Russell (2005), with
the nudging heights and relaxation times being identical to those used for stratospheric $O_3$. For stratospheric
CO and $HNO_3$ we constrain mixing ratios by using monthly mean ratios of $CO/O_3$ (Dupuy et al., 2004) and
$HNO_3/O_3$ (Jégou et al., 2008; Urban et al., 2009) based on the latitudinal climatologies derived from ODIN
observations using data for 2003/2004 (CO) and 2001-2009 ($HNO_3$). In order to avoid jumps in the nudging
constraints between months, we gradually change between ratios using the total monthly difference/number of
days in the month. These ratios are applied using the monthly mean stratospheric $O_3$ distribution in TM5-MP,
which is constrained by the MSR dataset (van der A et al., 2010). For both species, model fields are nudged at
5.5hPa, 10hPa and 28hPa using relaxation times of 5, 10 and 60 days, respectively. Previous versions of TM5
used a $HNO_3$ climatology from the UARS MLS instrument and applied nudging constraints at 10hPa only
(Huijnen et al., 2010).
For our study on the impact of horizontal resolution on the performance of TM5-MP, we present simulations
for the year 2006, which has been used for previous benchmarking studies (Huijnen et al., 2010; Williams et
al., 2012). We use a one year spin-up from the same initial conditions, where the initial conditions are
representative of the state-of-the-atmosphere for January 2005 taken from a previous simulation (see Zeng et
al., 2015). The model is run using 34 levels, as it will be used operationally for satellite retrievals, where
details of the pressure levels being given in Huijnen et al., 2010.

**2.2 Emission inventories**

All emission inventories applied in TM5-MP are yearly specific meaning that the year-to-year variability in
emission fluxes due to changes in anthropogenic activity, biogenic activity and burning extent are taken into
account. For the anthropogenic emission of $NO_x$, CO, $SO_2$, $NH_3$ and Non-Methane Volatile Organic
Compounds (NMVOC), we adopt the MACCity emission estimates described in Granier et al. (2011). The lack
of sector-specific information complicates the use of daily cycles for e.g. the road transport component, where
a bi-sinusoidal distribution could be applied peaking in the morning and late afternoon to represent variability
in traffic volume. Aircraft emissions are included only for NO, using a homogenous hourly flux estimate not
related to regional flight times. For volcanic $SO_2$ emissions, the estimated emission flux has been scaled up to
10 Tg S yr$^{-1}$ based on Halmer et al. (2002). For the biogenic component, where available we use the CLM-
MEGANv2.1 emission inventories produced for the Southern Hemispheric Multi-model Intercomparison
Project (SHMIP) as described in Zeng et al. (2015), with the missing trace species (e.g. ethane, propane, higher
organics) coming from alternative MEGAN simulations as outlined in Sindelarova et al. (2014). A diurnal
cycle is imposed on the isoprene emissions and introduced into the first ~50m between 20ºS-20ºN, whereas for
other latitudes a continuous daily flux is applied. The BB emissions are taken from the monthly estimates
provided by the GFEDv3 inventory (van der Werf et al., 2010) and latitude dependent injection heights and a
tropical burning cycle are implemented following Huijnen et al. (2010). All emission inventories are provided
on a 0.5º x 0.5º resolution and subsequently coarsened onto the horizontal resolution employed in any
simulation. In TM5-MP all $NO_x$ emissions are introduced as NO, rather than specifying a fraction that is
emitted directly as $NO_2$ (Carslaw and Beevers, 2005). Global $NO_x$ emissions for the year 2006 total 49 Tg N
$yr^{-1}$ (including lightning). Other notable species include CO (1081 Tg CO $yr^{-1}$), $SO_2$ (117 Tg S $yr^{-1}$), $CH_2O$
(13.5 Tg C $yr^{-1}$) and isoprene (510 Tg C $yr^{-1}$). An overview of the global and zonal emissions terms used in the
simulations analysed here are given in Table 3.
For lightning $NO_x$ we use the parameterization which uses convective precipitation fields (Meijer et al., 2001)
and constrain the annual global emission term at ~6 Tg N $yr^{-1}$. This uses the convective flux values meaning
that re-scaling of the nudging term was necessary in order to achieve similar total lightning $NO_x$ emissions
across simulations. An example of the resulting horizontal distributions in lightning $NO_x$ at ~400hPa for the
tropics for both horizontal resolutions is shown in the top panel of Fig. S1a in the supplementary material.
Although the spatial variability increases at 1º x 1º, the global distribution remains essentially the same, where
the constraints on annual lightning $NO_x$ emissions homogenize the total emission flux between resolutions.
One other factor affecting the vertical distribution of lightning $NO_x$ emissions is the convective
parameterization which is used. The lower panels of Fig. S1a show that, at this altitude, the Tiedke (1989)
approach increases the $NO_x$ emissions at this level by ~14%, accompanied by a significant re-distribution
between regions (c.f. SH below 30ºS and an significant increase in the tropical component in the Tiedke
simulation). However this is altitude dependent, where the absolute differences in the vertical distribution in
the monthly $NO_x$ emissions for a selection of latitudes are shown in Fig. S1b. Here comparisons are shown
both over the continents (e.g. 50.5ºN) and the oceans (e.g. 59.5ºS). Although the differences in the integrated
monthly global emission $NO_x$ flux is only around ~1-2%, the temporal and vertical distribution can be quite
different between convective schemes. Profiles show that in the upper troposphere ERA-interim consistently
results in higher $NO_x$ emissions around 300hPa, especially for July.
Latitudinal constraints on $CH_4$ global distributions are applied using the methodology given in Banda et al.
(2015) with a 3-day relaxation time. We also introduce similar constraints based on the appropriate surface
measurements for $H_2$ in order to account for the latitudinal gradient and variability across seasons, which
replaces the fixed global value of 550ppb used in previous versions. Finally, for Radon ($Rn^{222}$) emissions we
apply the estimates of Schery (2004), whose global distribution is given in Zhang et al. (2011).

**2.3 Observations**

Although the performance of mCB05 in TM5 v3.0 has been validated for selected NMVOC, $O_3$, $CH_2O$, CO
and NOy in both hemispheres (Williams et al., 2013; 2014; Fisher et al., 2015; Zeng et al., 2015), the
significant changes made to both the chemical scheme and the rate parameters in mCB05v2 necessitate
independent validation at both 3º x 2º and 1º x 1º. We choose a range of ground-based and airborne
measurements taken at diverse locations during the year 2006 representing different chemical regimes. Here
we briefly describe the observations utilised for this purpose.
For validation of simulated surface concentrations we use measurements of gaseous $O_3$, NO, $NO_2$, $HNO_3$ and
$SO_2$ available from the European Monitoring and Evaluation program (EMEP, www.emep.int), where we
exploit measurements taken at various background sites in Norway, Finland, The Netherlands, Belgium,
Poland, the Czech republic, Germany, Great Britain, Spain, Slovakia, Italy and Portugal. The number of sites
used for comparisons of trace species other than $O_3$ is smaller due to data availability. For the model
composites we extract data from 3 hourly instantaneous output in order to assemble both the weekly and
monthly mean values from the simulations. For the weekly comparisons of $NO_2$ and $SO_2$ we use values
extracted at 13:00 local time, close to the overpass time of the OMI instrument (e.g. Boersma et al, 2008). The
selected stations allow validation of the seasonality for both rural regions (FI37) and urban regions (NL09),
where we include identical stations where possible for both species. For $HNO_3$ we assemble the weekly values
from the daily averages.
Measured $[O_3]$ in the EMEP network are obtained using UV monitors (Aas et al. 2001). For all species, spatial
interpolation of model data is performed accounting for the height of the measurement station and by
weighting using the distance of the station from the surrounding grid-cells. The wide range of measurement
sites chosen ensures that both background and polluted cases are assessed.
For validating the vertical distribution of relevant trace species such as $O_3$, $SO_2$ and $CH_2O$, we use
measurements by the DC-8 aircraft during the Intercontinental Chemical Transport Experiment B (INTEX-B;
Singh et al., 2009) that took place between March and May 2006. Observations of a host of co-located
nitrogen-containing species are available (namely NO, $NO_2$, PAN and $HNO_3$). These flights were conducted
over a wide region, and we use all three months of measurements. Each month sampled a different region
representing different meteorological conditions and local emission sources, namely: the Gulf of Mexico (90-
100ºW, 15-30ºN), the remote Pacific (176-140ºW, 20-45ºN) and to the south and west of Alaska over the
ocean (160-135ºW, 20-60ºN). Measurements cover altitudes up to 10.5km, and we bin the values with respect
to pressure using 50 hPa bins or less in the LT. We interpolated three-hourly output against measurements for
each respective day, similar to the comparisons performed in previous evaluations of TM5 (e.g. Huijnen et al.,
2010), but we segregate our comparisons into the three distinct regions. For details relating to the location of
each flight the reader is referred to the campaign overview of Singh et al. (2009).
For tropospheric $O_3$, we supplement the INTEX-B comparisons with measurements taken over more polluted
regions as part of the Measurement of Ozone, water vapour, carbon monoxide and nitrogen oxides by Airbus
In-service aircraft initiative (MOZAIC; Thouret et al., 1998). We aggregate the measurements as seasonal
means for December-January-February (DJF) and June-July-August (JJA) in order to provide a robust number
of samples for each location. Here we choose to use profiles representative of the Northern mid-latitudes,
namely: London (0.2ºW, 51.2ºN), Vienna (16.5ºE, 48.1ºN), Washington (77.5ºW, 38.9ºN), Portland (122.6ºW,
45.6ºN), Shanghai (121.8ºE, 31.2ºN) and Tokyo (140.4ºE, 35.8ºN).
We also make comparisons of $O_3$, NO, $NO_2$, selected N-reservoir species, $SO_2$ and $CH_2O$ profiles using
measurements made aboard the NOAA WP-3D aircraft as part of the Second Texas Air Quality Study
(TexAQS II; Parrish et al, 2009), which was conducted over the Texas sea-board during September and
October 2006. This allows the assessment of TM5-MP over a region with higher NMVOC emissions and
industrial activity. These measurements were typically sampled at altitudes below 500hPa, therefore no
measurements in the UTLS are available from this campaign.

**3 The Effect on Atmospheric Transport**

Here we analyse the differences in convective transport out of the BL by analysing the vertical and horizontal
distribution of $^{222}$Rn, which is a diagnostic typically used for assessing the differences in transport in CTMs
(e.g. Jacob et al., 1997). $^{222}$Rn is emitted at a steady rate and exhibits a half-life of ~3.8 days, which is long
enough to be transported from the BL into the FT due to chemical passivity, with loss via wet scavenging and
dry deposition being negligible. Therefore, it acts as an ideal tracer to assess differences in convective transport
from the surface out of the BL. The representation of BL dynamics for TM5-MP has recently been assessed at
1° x 1° using $^{222}$Rn distributions for both the Tiedtke (1989) scheme and when adopting convective mass
transport values from the ERA-Interim meteorological data (Koffi et al, 2016).
Figure 1 shows seasonal mean horizontal global distributions of $^{222}$Rn for DJF and JJA in the 1° x 1° simulation
averaged between 800 and 900hPa (i.e. sampling the LT). Also shown are the associated percentage
differences against the re-binned 3° x 2° $^{222}$Rn distribution, allowing a direct comparison. Resolution dependent
differences result from the cumulative effects of the use of higher resolution mass-fluxes from the ERA-interim
meteorological data for describing convective activity and the more accurate temporal distribution of regional
$^{222}$Rn emissions at 1° x 1°. In general it can be seen that seasonal differences of ±20% exist, typically with
increases over continents and decreases over oceans in the 1° x 1° simulations. Maximum differences of >60%
occur near selected coastal regions (California, West Africa, Madagascar) or in outflow regions such as off
South America and Africa, where differences exhibit a strong seasonal dependency. This is due to the large
differences in convective strength due to the variability in heating rates, and thus temperatures, between land
and ocean (e.g. Sutton et al., 2007).
A comparison of the ratio of the monthly mean $^{222}$Rn profiles (1° x 1° /3° x 2°) extracted above selected
European cities for January (black) and July (blue) 2006 are shown in Fig. S3 in the Supplementary Material.
The typical tropospheric profile of $^{222}$Rn exhibits an exponential decay from the LT to the FT (not shown). In
order to homogenise the emission flux in the comparison, we coarsen the 1° x 1° data onto the 3° x 2° grid by
averaging the six individual values into a representative mean column. The extent of the changes in the vertical
distribution of $^{222}$Rn is somewhat site specific meaning an in depth analysis is beyond the scope of this paper.
In summary, the 1° x 1° simulation generally provides stronger convective activity for January, with the main
impact occurring below 700hPa (e.g. London and Paris). The changes in $^{222}$Rn in the LT range between 2 and
10% (i.e. ratios of 0.9 to 1.1), implying both weaker and stronger convective transport depending on changes in
location (e.g. orography and land type). The impact at Berlin is larger than e.g. Barcelona which shows that,
surprisingly, the inclusion of a large ocean fraction (with weaker convective mixing) in the 3° x 2° cell does not
seem to introduce dominating effects. Recently Koffi et al. (2016) have shown that comparisons of $^{222}$Rn at
coastal sites in Europe at 1° x 1° exhibit significant discrepancies compared to more continental stations. For
July the changes in the vertical distribution extend into the FT up to 500hPa, although changes in the upper FT
have a significant component due to changes in long-range transport. The magnitude of the changes are similar
to those exhibited during January, although maybe of the opposite sign (e.g. Rome). Thus the influence on e.g.
$NO_2$, $CH_2O$ and $SO_2$ *a-priori* vertical profiles will be non-negligible and diverse.
For the tropical cities located in regions where convective mixing is stronger, the corresponding differences
between resolutions can reach ± 20%, especially near the surface (e.g. Caracas and Karachi). There is a site-
specific seasonal dependency in the magnitude of the changes related to the regional land characteristics (e.g.
Lagos versus Kuala Lumpar) and the extent of ocean within any particular grid-cell. Thus, differences in *a-*
*priori* vertical profiles of trace gases using a resolution of 1º x 1º can be considerable compared to those
provided at a 3º x 2º resolution.
We also show ratios of profiles from 1º x 1º simulations using the convective scheme of Tiedtke (1989) against
those using the convective mass-fluxes from the ERA-interim meteorological dataset (Fig. S4), defined as
ERA(1º x 1º)/Tiedtke (1º x 1º). For this comparison no daily averaging is employed with $^{222}$Rn profiles
extracted from 3 hourly instantaneous sampling, with the profiles shown being interpolated directly above
urban conurbations (with high trace gas emissions). The ratios show that the significant differences exist, with
the convective mass-fluxes from ERA-interim being somewhat weaker than those calculated online using
Tiedtke (1989) (i.e) the ratio is typically less than 1, especially during July. In the recent study by Koffi et al.
(2016) performed at 1º x 1º for the European domain, there was no appreciable improvement in the correlation
co-efficients when distributions of $^{222}$Rn were compared against measurements resulting in no strong
conclusion towards which of the parameterizations results in better atmospheric transport.

**4 The Impact on tropospheric photolysis frequencies**

The changes in the spatio-temporal distribution of tropospheric clouds and surface albedo have the potential to
alter the incident flux of photolysing light reaching the surface, and thus photochemical production and
destruction terms. When present, clouds dominate the integrated optical density in the tropospheric column.
TM5-MP uses a random overlap method for determining the impact of clouds on actinic flux, which is
weighted by cloud cover (Williams et al., 2012). Comparing seasonal mean cloud coverage for DJF and JJA
(Fig. S5), we show that there are significant increases in the fractional cloud cover (*fcc*) at 1º x 1º resulting in
*fcc* values ranging between 0.1-0.8 (c.f. 0.1-0.5 for 3º x 2º). Moreover, the definition of tropical equatorial
cloud systems becomes much more defined and there are significant differences in the cloud distributions
around the west coast of Southern America. For DJF, the largest changes occur at high latitudes over the tundra
and oceans, but correspond with low intensity incident radiation due to the polar winter. For the SH, the
seasonal *fcc* increases significantly, which will potentially impose effects on Antarctic oxidative capacity (see
Sect. 5). For JJA, most increases in *fcc* do not occur directly above high $NO_x$ sources but rather over the
oceans. This limits the impact on the lifetime of chemical pre-cursors (e.g. $NO_2$) as discussed in Sect. 6.
Examining similar plots for surface albedo (not shown) reveals that maximum differences (increases at 1º x 1º)
again occur in the polar regions under low temperatures related to sea-ice and snow coverage typically during
polar winters. For mid-latitudes and tropics, although differences in the absolute albedo value can be
significant (±50%) values are typically below 0.1, which will contribute to the perturbations in the final J value
tropospheric profiles as discussed below. The monthly mean comparisons in surface J values provided in Fig.
S6 show that any differences in instantaneous cloud cover are moderated to the order of a few percent when
looking at longer periods.
The similarity in the monthly mean photolysis frequencies for $O_3$ and $NO_2$ across resolutions (hereafter
denoted $J_{O3}$ and $J_{NO2}$, respectively) are shown in Fig. S6 of the Supplementary Material. Comparisons of the
monthly mean $J_{O3}$ and $J_{NO2}$ values are shown at five different locations identical to those shown in Williams et
al. (2012). For $J_{O3}$ the impact of increasing resolution is limited to a few percent in the monthly mean values,
even for regions which have high surface albedo. At the global scale this leads to a reduction of ~2% in the
total mass of $O_3$ photolysed (not shown). For $J_{NO2}$, the corresponding differences become more appreciable,
with 1° x 1° exhibiting ~5-10% higher values at high Northern latitudes (associated with locations with high-
$NO_x$ regimes). Focusing on $J_{NO2}$ and comparing seasonal mean values near the surface shows that very similar
large-scale spatial patterns occur for both simulations at the global scale (c.f. Fig. S7). The highest $J_{NO2}$ values
occur over the tropical oceans and high altitude regions (e.g. Nepal), with a latitudinal shift related to seasonal
changes in daylight hours. Although more regional fine-structure can be seen at 1° x 1° (e.g. South-Western US
and South-West China for DJF), these seasonal averages show that the small perturbations in $J_{NO2}$ shown in
Fig. S6 extend to the global scale, leading to only modest changes in the tropospheric lifetime of $NO_2$ (see
Sect. 6).
Comparisons of monthly mean vertical profiles of $J_{O3}$ and $J_{NO2}$ as sampled over selected tropical cities are
shown in the Figs. S8a and b, respectively, in the Supplementary Material. Here no averaging is performed
towards an identical horizontal resolution, therefore values are representative of the $J$ values directly above the
selected urban centres. The $J_{O3}$ profiles are affected to a larger extent than the $J_{NO2}$ profiles, due to the
characteristic absorption spectra of each species which makes $J_{O3}$ more sensitive to the additional scattering
introduced due to clouds. Profiles over Dubai act as a proxy for clear-sky conditions, where values of unity
exist in the residual of $J_{O3}$ and $J_{NO2}$ calculated through most of the column. The small difference at the surface
is due to changes in the surface albedo between resolutions, with Dubai being situated on the coast meaning
that a sharp horizontal gradient exists in surface albedo. For other cities, the largest perturbations occur away
from the surface (e.g. Jakarta, Nairobi and Lagos) around the altitude where tropospheric clouds are most
abundant. There are typically changes of between ±5-10% in the monthly mean profiles. The changes in $J_{NO2}$
reflect those simulated for $J_{O3}$, with somewhat smaller perturbations.

**5 Implications for oxidative capacity and tropospheric $O_3$**

The partitioning of reactive N between the short- and long-lived chemical N-reservoirs included in TM5-MP
depends on the oxidative capacity simulated for the troposphere via competition between the various different
radicals (i.e. OH, $CH_3C(O)O_2$, $NO_3$ and $CH_3O_2$). Therefore, changes to the distribution and resident mixing
ratios of tropospheric $O_3$ subsequently impose changes in the fractional composition of the $NO_y$ budget
(Olszyna et al., 1994) and also the efficiency of the $NO_x$ recycling terms by altering the chain length (Lelieveld
et al., 2004). In this section we analyse the global and zonal chemical budget terms for tropospheric $O_3$ to
highlight the inter-hemispheric differences which occur (i.e. under low and high-$NO_x$ environments). An
overview of the resulting near-surface global distribution of tropospheric $O_3$ for May 2006 is shown in Figure
3, which also includes the location of the regional comparisons presented below in a larger global context. In
general, the pattern of minimum and maximum mixing ratios in $O_3$ occur in similar locations, with the long-
range transport component being more clearly defined in the 1° x 1° simulation. There is a distinct latitudinal
gradient in $O_3$ mixing ratios imposed by the global distribution of $NO_x$ emissions.
Table 4 provides the zonally segregated chemical budget terms for tropospheric $O_3$, from which the global
component due to STE can be determined by closing the budget terms following the methodology given in
Stevenson et al. (2006). The chemical tropopause calculated for 3° x 2° is applied for the analysis of 1° x 1°
budget terms to ensure that a valid comparison is performed, (i.e. the same mass of air is accounted for). For
computational efficiency the budget terms are aggregated in 10º latitudinal bins for each vertical level and
summed across all longitudes providing the cumulative latitudinal terms.
The most significant change with resolution concerns STE. By using a dedicated tagged stratospheric $O_3$ tracer
(which only undergoes photo-chemical destruction and deposition in the troposphere; hereafter denoted as
$O_3S$) changes in the zonal mean STE can be determined. The stratospheric burden of $O_3$ ($BO_3$ (strat)) exhibits a
strong hemispheric gradient with much more downwelling occurring in the NH peaking during boreal
springtime. At the global scale the STE exchange is 579 Tg $O_3$ $yr^{-1}$, which agrees well with the multi-model
mean for STE of 556±154 Tg $O_3$ $yr^{-1}$ in Stevenson et al. (2006), with observational estimates being ~550±140
Tg $O_3$ $yr^{-1}$ (Olsen et al., 2001). The ~7% reduction of STE at 1º x 1º is encouraging considering that previous
studies using TM5 have concluded that STE in TM5 at 3º x 2º was biased high compared to STE inferred from
TES and MLS satellite observations (Verstraeten et al. (2015)). The increase in STE in the SH, with an
associated  decrease in the NH (see below), implies that there is a shift in circulation patterns at 1º x 1º even
though $BO_3$ (strat) remains essentially unchanged. Previous studies have shown that in order to resolve the
correct spatial and temporal Stratosphere-Troposphere flux, high resolution is required both in the horizontal
and the vertical gridding (e.g. Meloen et al., 2002). The NH STE diagnosed with TM5-MP is an order of
magnitude smaller than estimates derived in a CTM study also conducted at a 1º x 1º resolution (Tang et al,
2011; ~200 $TgO_3$ $yr^{-1}$), which identified deep convection as important for STE. Here we use a different vertical
grid and meteorological dataset to drive TM5-MP, both of which affect the ability towards capturing an
accurate STE flux (Meloen et al., 2002). For the 1º x 1º simulation using the Tiedtke scheme, there is a further
reduction in the STE component of 21 Tg $O_3$ $yr^{-1}$, resulting in an STE component almost identical to the multi-
model mean in Stevenson et al. (2006).
The zonal seasonal means of the fraction of $O_3S$ to $O_3$ ($O_3S/O_3$) for both simulations are shown in Fig. 2 for
DJF and JJA. There is a clear seasonal zonal shift in the fractional contribution due to the $O_3$ transported
downwards from the Stratosphere exhibiting a longer lifetime in the winter hemisphere reflecting a lower
photochemical destruction rate. At 1º x 1º the largest increase in STE occurs in the Southern Hemisphere (SH)
during JJA. Here ~20-25% of tropospheric $O_3$ is transported down from the Stratosphere. Comparing the 0.2
contour for the NH mid-troposphere shows significant changes, extending further down towards the surface
during boreal wintertime leading to the higher total mass of $O_3S$ in the troposphere. The extent of nudging
towards the MSR climatology is essentially constant across simulations (c.f. Table 4). Interestingly, less $O_3S$
reaches the surface in the tropics at 1º x 1º due to the enhanced chemical destruction term in the Free
Troposphere. Approximately 10% of the global deposition term for $O_3$ is associated with $O_3$ that originates
from the Stratospheric at 1º x 1º (c.f. ~5% at 3º x 2º). For the NH, this contributes to the simulated increase in
deposition of ~9%.
For tropospheric $O_3$ there are similarities that occur between the NH, tropics and SH (i.e.) high and low-$NO_x$
scenarios, resulting in a cumulative decrease in $O_3$ production of ~2-4% across zones. For the chemical loss
terms there is a modest decrease of ~3% (~2%) in the NH (SH) reflective of the changes discussed for $J_{O3}$,
which acts as the primary destruction term. Therefore, in the SH the significant differences shown for *fcc* do
not significantly impose a lower photochemical destruction term on the annual tropospheric $O_3$ budget. There
is a zonal gradient in the tropospheric burden of $O_3$ ($BO_3$) following the zonal gradient in $NO_x$ emissions (c.f.
Fig. 3). Comparing terms shows that $BO_3$ decreases at 1º x 1º by a few percent at the global scale (~7 Tg $O_3$)

making a rather small impact on oxidative capacity. This is of the same order of magnitude as that found in previous studies concerned with horizontal resolution (e.g. Wild and Prather, 2006). Interestingly, changes in the deposition flux of $O_3$ are rather small, even though there is a larger amount of variability in the land surfaces and better-resolved land-sea contrast at 1º x 1º, although differences in regional deposition fluxes can be more significant. Multi-model inter-comparisons of surface deposition terms across models have shown previous versions of TM5 to be at the low end of the model spread in terms of $O_3$ (Hardacre et al., 2015), suggesting that the surface deposition flux should be increased by ~10% in TM5-MP towards the multi-model mean value. This can be partly attributed to the large uncertainty which exists related to the loss of $O_3$ to the ocean (Hardacre et al., 2015).

Figure 4 shows comparisons of simulated and observed mass mixing ratios of surface $O_3$ at EMEP sites across Europe (www.emep.int; Aas et al. 2001), with stations chosen to cover a range of latitudes. Previous comparisons using mCB05 have revealed high biases in surface $O_3$, especially during boreal summertime (Williams et al., 2013). These high biases originate from cumulative effects associated with the accuracy of the emission inventories, the convective and turbulent mixing component, the underestimation of the scattering and absorption of photolysing light due to aerosols and the chemical mechanism that is employed. For the emission component it should be noted that even at 1º x 1º coarsening is performed, where emission inventories are typically supplied at 0.5º x 0.5º resolution. The seasonal cycle in surface $O_3$ is captured to a large degree, and the high bias exhibited by the model is generally reduced by ~2-5 ppb (or ~20%) at 1º x 1º. This is associated with perturbations in the $NO_x$ recycling terms, chemical titration by NO, changes to the turbulent diffusion and convective mixing out of the BL. In that the improvement in biases is largest during boreal summertime is associated with the shorter chain length of the $NO_x$ recycling term during boreal wintertime. However, there is still a significant monthly mean bias in both simulations when compared against observations throughout the year, especially for locations impacted by a large anthropogenic $NO_x$ source. This is partly due to the low $NO/NO_2$ ratio as discussed in Sect. 6 below.

Comparing vertical profiles from composites assembled from the MOZAIC measurements for DJF and JJA (Figs. S9a and S8b, respectively), INTEX-B (Singh et al, 2009; Fig. S10) and TexAQS II (Parrish et al, 2009; Fig. S11) show consistently that differences are small between simulations across regions, and typically mimic those which occur at the surface. There is a general positive bias of ~20-40% in mixing ratios exhibited across all comparisons, although the variability in the vertical gradients across regions is captured rather well. Such positive biases have consequences for both the $NO_x$ recycling terms and $HNO_3$ formation discussed in the sections below.

**6 Implications for the distribution of NO and $NO_2$**

Table 5 provides the zonally segregated annual $NO_x$ recycling terms involving the main peroxy-radicals and the direct titration term involving NO for the 1º x 1º simulation. The conversion rate of NO back into $NO_2$ decreases by ~2-3% across zones as a consequence of an associated increase in the titration term and re-partitioning of N into long-lived reservoir species (see below). For the titration term involving NO, although the globally integrated flux remains relatively constant, there is contrasting behaviour for the two most important zones (TR, NH), which exhibit a lower and higher titration term, respectively. It has been shown that

for regions such as Europe the increased titration results in lower surface $O_3$ mixing ratios (c.f. Fig. 4),
improving the boreal summertime high bias at the surface.
Important model uncertainties include the quality of the MACCity $NO_x$ emission inventory, the lifetime of $NO_2$
simulated in TM5, BL mixing and the $NO_x$ recycling term via the chemical titration of $O_3$. Figure 3 provides an
illustration of the global distribution in surface $NO_2$ during May 2006 for both the 3º x 2º and 1º x 1º
simulations, where the short-lifetime means that the maximum mixing ratios occur directly near the strong
source regions. Most $NO_x$ is anthropogenic in origin, therefore there is strong latitudinal gradient between the
NH and SH, with ship-tracks also visible. The regions where validation occurs are also superimposed in the
figure, including the extent of the EMEP domain over which $NO_2$ weekly comparisons are made.
Figures 5 and 6 shows comparisons of weekly [NO] and [$NO_2$] surface measurements against the
corresponding composites from both simulations, sampled at 13:00 local time which is close to the local
overpass time for both OMI and tropOMI. Although the number of EMEP sites conducting $NO_x$ measurements
is smaller than those measuring $O_3$, we choose stations located throughout Europe in both high and low $NO_x$
regimes. To supplement these comparisons we provide the seasonal mean biases for DJF and JJA from both
simulations in Tables 6 and 7, respectively, calculated using weekly binned data from all EMEP sites that
measure hourly [NO] and [$NO_2$]. Here we perform an analysis across sites rather than focusing on the
behaviour at selected individual locations.
For the determination of [$NO_2$], the reduction of NO on a Molybdenum convertor takes place with subsequent
detection by chemi-luminescense, with an associated detection limit of ~0.4ppb. Previous studies have shown
that some bias can result due the oxidation of nitrogen reservoirs such as PAN (Dunlea et al., 2007;
Steinbacher et al., 2007). In TM5-MP all $NO_x$ emissions are introduced as NO, although a fraction for road
transport is known to be emitted directly as $NO_2$ (e.g. Carslaw and Beevers, 2005). Many studies have been
performed comparing satellite $NO_2$ columns with model values, implying that inadequacies in emission
inventories are somewhat region specific (e.g. Zyrichidou et al., 2015; Pope et al., 2015).
Table 6 shows a negative bias of a few µg m$^{-3}$ in TM5-MP in seasonal surface [NO] in Europe. This is a
cumulative effect of the accuracy of the MACC $NO_x$ emission estimates, an overestimate in daytime vertical
mixing (Koffi et al., 2016) (enhanced dilution) and, to a lesser extent, too high surface [$O_3$] (increasing the
oxidation rate of NO to $NO_2$). As anthropogenic emissions are the principle source of NO, there is no
significant seasonal cycle in the monthly emission estimates in the NH. Seasonal differences in convective
mixing (i.e. lower BL heights) do cause somewhat higher surface [NO] during DJF for approximately equal
emission terms. This is captured by TM5-MP, although under night-time conditions TM5-MP has been shown
to overestimate nocturnal BL heights (Koffi et al., 2016). For ~80% of the EMEP sites we do not observe any
significant change in the quality of the comparisons. For ~20% of sites, simulations of [NO] at 1º x 1º
introduce significant improvements over those at 3º x 2º and there is an improvement regarding the extent of
seasonal variability (Fig. 4).
Table 7 shows that for [$NO_2$] the biases are more variable being typically in the range of ±0-6 µg m$^{-3}$, with both
positive and negative biases occurring across sites. Both the conversion efficiency from NO, loss to reservoir
compounds (e.g. $HNO_3$), photo-dissociation rate, convective mixing and emission estimates contribute to these
biases. The seasonal biases show improvements at 1º x 1º for ~35% of the EMEP sites, accompanied with
degradations at ~20% of the sites. The maximum biases in [$NO_2$] at 1º x 1º can be approximately double those
for [NO]. For the corresponding NO/NO$_2$ ratio, there will generally be an under prediction in the model due to
the negative biases shown for the [NO] comparisons. Analyzing the corresponding seasonal correlation co-
efficients (not shown) shows in ~25% of the cases there is little seasonal correlation between the weekly [NO$_2$]
in TM5-MP and the measurements regardless of resolution for both seasons (Pearson's $r$ in the range -0.3 to
0.3). In ~30% of cases there is actually a degradation in $r$ between resolutions, the changes somewhat reflect
those seen in the seasonal biases i.e. simultaneous changes to both the meteorology and local emission fluxes
do not necessarily improve the performance of the model. Comparing 1° x 1° values both with and without the
Tiedtke convection scheme shows that for the most convective regions (e.g. south of 45°N) increases in $r$
generally occur during JJA when employing the ERA-interim mass-fluxes. Conversely for e.g. Finland the
correlation becomes worse.
Beyond Europe, we compared monthly mean TM5-MP vertical distributions of NO and NO$_2$ between March
and May 2006 against measurements taken during the INTEX-B campaign in Fig. 7. In general differences
between 1° x 1° and 3° x 2° simulations are the order of a few percent, with NO$_2$ biased low in the LT by ~70-
80%. This is partially associated with the take-off and landing of the aircraft from polluted airfields, where
point sources of high anthropogenic emissions cannot be resolved even at 1° x 1°. For March, there is a strong
signature from biomass burning plumes in the middle troposphere which is not captured using the monthly
burning estimates. For the FT, TM5-MP captures the observed gradient to reasonable degree. In the UT there is
a consistent high bias for NO and an associated low bias for NO$_2$ suggesting that the conversion term is too low
and the NO$_x$ cycle is out of synch at these cold temperatures despite the addition of new reservoir species (i.e.
CH$_3$O$_2$NO$_2$) and application of new rate data.
One important gauge as to whether the chemical mechanism can capture the correct recycling efficiency of NO
into NO$_2$ is to examine their ratio, which is presented in the third column of Fig. 7. In the LT (< 900 hPa)
NO/NO$_2$ ratios of 0.1-0.2 exist that are captured quite well by TM5-MP, with negligible differences between 3°
x 2° and 1° x 1 ° simulations. For the FT, TM5-MP consistently overestimates the ratio in spite of a high bias in
O$_3$ (c.f. Fig. 4) which is imposed by the overestimates in NO$_2$. This implies the chemical conversion is too slow
and, assuming representative $J_{NO2}$ values, indicates a low bias in HO$_2$ or an under-estimation in the mixing
ratios of other long-lived and short-lived NOy compounds (see Sect. 7).
Finally in Fig. 8 we show the corresponding comparisons against measurements taken during the TexAQS II
campaign (Parrish et al, 2009) for both September and October 2006. As for the EMEP comparisons shown in
Figs. 5 and 6, there is a significant underestimation in NO and NO$_2$ mixing ratios, with both model profiles
being outside the 1-σ variability in the observational mean. This is clearly related to the emission estimates for
this region being underestimated in the current emission inventories (e.g. Kim et al., 2011). For the resulting
NO/NO$_2$ ratio, TM5-MP captures the correct ratio in the lowest few hundred meters of the BL, but
overestimates the ratio at higher altitudes as for more pristine environments, although there is marked
improvement in the ratios simulated for October.

**7 Changes in the NOy budget**

**7.1 Long-lived reservoirs**

The resolution dependent changes in the temporal distribution of $[NO_2]$, and associated differences in NMVOC
chemical pre-cursor emissions have the potential to alter the partitioning of reactive $NO_x$ between the three
main chemical reservoirs included in mCB05v2 (i.e. $HNO_3$, PAN and ORGNTR). The differences in both the
deposition efficiency and tropospheric lifetimes between trace species at 1° x 1° suggests that the fraction of
$NO_x$ that can be transported out of source regions could change significantly. Here we briefly examine the
zonally integrated nitrogen budget terms between simulations to quantify the effect of applying a higher spatial
resolution. The seasonal distribution of these three dominant reservoir species at 1° x 1° and their individual
contributions to total NOy are shown in Figs. S12-S15 for DJF and JJA, respectively. Here we define $NO_y$ as
the cumulative total of NO, $NO_2$, $NO_3$, $HNO_3$, PAN, $CH_3O_2NO_2$, HONO, $2*N_2O_5$, lumped organic nitrates
(ORGNTR) and $HNO_4$. It should be noted that methyl-nitrate is not in this version of TM5-MP. These figures
are provided as reference for the reader to aid understanding of the discussion below.
Table S1 in the Supplementary Material provides a zonal decomposition of the tropospheric chemical budget
terms for $HNO_3$, PAN and ORGNTR. For $HNO_3$, even though the recent kinetic rate parameters increase
(decrease) the chemical production term at the surface (UTLS) compared to older rate data (e.g. Seltzer et al.,
2015), changes in the integrated column term are small. The changes at 1° x 1° are somewhat latitude
dependant (corresponding to low and high $NO_x$ regimes), with only small increases occurring in the NH and
associated decreases in the tropics related to lower [OH] (i.e. chemical production). Loss by cumulative
deposition terms only changes by a few percent, due to wet scavenging being so efficient for $HNO_3$ any
associated change in the Surface Area Density (SAD) of cloud droplets ($cm^2/cm^3$) introduced by changes in the
liquid water product
For PAN, both the production and destruction terms decrease marginally by ~1-3% across all zones, meaning
the transport of $NO_x$ out of the main source regions remains relatively robust. The total mass of N cycled
through PAN is ~four times that sequestrated as $HNO_3$, although the lifetime of PAN is shorter due the
efficient thermal decomposition. The changes in the production term due to temporal increases in $NO_2$ near
high $NO_x$ source regions (c.f. Fig. 3) are partially offset by a reduction in the mixing ratios of the acetyl-peroxy
radical ($C_2O_3$ in Table 1) due to e.g. increased dry deposition of organic precursors at 1° x 1°. Although the
chemical budget terms only exhibit small changes, it can be expected that the global distribution of PAN is
somewhat different due the changes in the convective and advective mixing due to the application of higher
resolution meteorological data (c.f. Sect. 3).
For ORGNTR, there is a 5% reduction in the annual production term at 1° x 1°, with an associated decrease in
the loss by deposition. Both the largest production and, thus, destruction terms occur in the tropics related to
the strongest source of ORGNTR being biogenic pre-cursors in mCB05v2. Thus at 1° x 1°, this intermediate
trace species becomes less important as a $NO_x$ reservoir, .
Finally, the one additional intermediate not shown is $CH_3O_2NO_2$, which is primarily a stable vehicle for
transporting $NO_x$ from the surface up to the UTLS, where at cold temperatures it accounts for a significant
fraction of $NO_2$ speciation along with $HNO_4$ (Browne et al., 2011). At the global scale three times as much
nitrogen cycles through $CH_3O_2NO_2$ compared to PAN, although the thermal stability is low at temperatures >
255°K thus resident mixing ratios are typically small. This results in maximum mixing ratios occurring in the
cold upper troposphere (up to ~0.2 ppb) and subsequently dissociates primarily by thermal decomposition
(photolytic destruction accounting for <0.1% of all destruction). At 1° x 1° there is a few percent decrease in
the chemical production term as a result of lower $CH_3O_2$ mixing ratios and more variability in the temporal
temperature distribution.
Comparisons of weekly $[HNO_3]$ at the surface in Europe are shown in Fig. 9 against measurements from the
EMEP network. It has recently been determined that $HNO_3$ measurements are also sensitive to ambient night-
time $[N_2O_5]$, which could result in a positive bias in the observations (Phillips et al., 2013). In general, the
modelled seasonal cycle is not evident in the measurements, which exhibit a rather homogeneous variation in
mixing ratios throughout the year typically, thus being somewhat decoupled from variability in photochemical
activity. Comparisons show an underestimation in TM5-MP during March and an overestimation during JJA.
No such seasonal pattern is observed for $[NO_2]$ (c.f. Fig. 6), thus seasonal $[OH]$ variability due to variations in
photo-chemical activity and $[H_2O_{(g)}]$ and/or an incorrect wash-out term which could both act as likely causes.
The impact of resolution on $[HNO_3]$ is rather muted for most weeks resulting in no significant changes to the
seasonal biases (not given), as constrained by the improvements in surface $[NO_2]$ (c.f. Fig. 6). The
heterogeneous scavenging of $HNO_3$ into ammonium nitrate can act as a moderator toward gaseous $HNO_3$.
Although this heterogeneous conversion process is included in TM5-MP as described by the EQSAM
approach, low concentrations of e.g. ammonium nitrate (not shown) typically result. Thus, gaseous $[HNO_3]$
remains too high due to too little conversion into particles and subsequent deposition.
For other regions outside Europe, we make comparisons of vertical profiles of $HNO_3$ and PAN between March
and September 2006 against those measured during INTEX-B (Figure 10) and TexAQSII (Fig. S16). PAN is a
good marker for transport in the free-troposphere due to the relatively long-lifetime at colder temperatures. For
all regions the vertical gradients for both species are captured quite well, although some fine-structure is lost
due to the vertical resolution of TM5-MP and insufficient pre-cursor emissions. This implies that the
underestimation in $NO_2$ simulated in the UTLS (Fig. 7) is not due to insufficient transport of $NO_2$ away from
source regions, but rather should be attributed to either missing chemistry or insufficient transport down from
the stratosphere. Finally, for more polluted regions, the vertical gradient of $HNO_3$ is rather less steep than that
observed, with significant low biases in the lower troposphere related to the low bias in $NO_2$ shown in Fig. 8.
The impact of the 1° x 1° resolution only results in a marginal improvement in the LT for $HNO_3$ (again similar
to $NO_2$). For PAN the vertical profile in TM5-MP agrees remarkably well, although somewhat anti-correlated
around 900hPa in both simulations and the rapid decrease in the lowest kilometre not being captured
sufficiently. This overestimation would likely be larger if the emission estimates were increased as required to
consolidate the $NO_2$ comparisons for the same campaign.

**7.2 Short-lived reservoirs**

Here we briefly discuss the perturbations introduced for the short-lived N-reservoirs, namely HONO, $HNO_4$
and $N_2O_5$, where the chemical budget terms for all three species are provided in Table S2 in the Supplementary
Material. For HONO it should be noted that many tropospheric CTMs have difficulty in simulating observed
mixing ratios (e.g. Goncalves et al, 2012) suggesting missing (heterogeneous) source terms. The global
production for HONO is an order of magnitude less than that for the other short-lived N-reservoirs. At 1° x 1°
there is ~10% more chemical production of HONO in high $NO_x$ regions and no appreciable effect in low $NO_x$
regions. Thus the impact of increased resolution on HONO production is rather small, which is surprising
considering the higher NO mixing ratios that occur in high $NO_x$ regions (c.f. Fig 5). The muted response is due
to competing oxidative processes, which effectively lower the OH available. For $HNO_4$, approximately the
same mass of N cycles through this species as for PAN, although the shorter lifetime means that it is more
important at regional scale. Again, the impact of resolution on this species is small, where decreases in $[HO_2]$
result in no significant net change in production for the NH. The most significant changes occur for the global
production and heterogeneous conversion of $N_2O_5$, with enhanced chemical production of ~12% at the global
scale, increasing the heterogeneous sink term by ~6%, although the changes in the total mass of N converted
are small. In general, this is due to an increase in the production of the $NO_3$ radical by ~10% at 1° x 1° (not
shown) resulting in enhanced $N_2O_5$ mixing ratios.

**8 Implications for tropospheric $CH_2O$ retrieval**

The implications of applying a higher resolution CTM for the global distribution of $CH_2O$ are rather modest.
Fig. S17 shows the near-surface global distributions of $CH_2O$ for May 2006, where maximum mixing ratios
occur near forested regions due to the link with isoprene oxidation (e.g. Palmer et al., 2006). The tropospheric
lifetime of $CH_2O$ is of the order of a few days, meaning that transport has little impact between simulations
apart from in low emission areas. Also shown are the locations where regional validation occurs. In Table 8,
we show zonally integrated chemical production and destruction terms for $CH_2O$, which suggests changes of
the order of a few percent at the global scale. The most notable difference is the increase in the cumulative
deposition term of ~4% at 1° x 1°, thus reducing the atmospheric lifetime of $CH_2O$ in TM5-MP. Again, this low
impact shows that the increase in the temporal variability of the meteorological data at 1° x 1°, and thus the
local variability of cloud SAD, only changes the net deposition term by a few percent. Even though the
temporal distribution of the surface mixing ratios shows more variability at 1° x 1° due to the better
representation of regional pre-cursor sources terms (e.g.) isoprene and terpene, only moderate improvements
occur to the simulated profiles and total columns due to changes in transport. For instance, when analysing
individual production terms (not given) for the tropics, decreases are related to small changes in the dominating
chemical source terms (e.g. oxidation of $CH_3OOH$; a reduction of ~3-5 Tg $CH_2O$ $yr^{-1}$). For the chemical
destruction term, the relative insensitivity of the photolysis of $CH_2O$ towards resolution (similar to $J_{O3}$; c.f. Fig
S4) results in small net decreases in line with changes in the chemical production term.
Figure 11 compares monthly mean tropospheric profiles of $CH_2O$ measured during INTEX-B (Singh et al.,
2009) with those from both TM5-MP simulations for March to May 2006. In general, there is a fair
representation of the vertical gradient of $CH_2O$ by TM5-MP for all months shown, although surface mixing
ratios are typically too high suggesting loss by deposition to the ocean is underestimated (potentially related to
underestimations in surface area due to lack of 3D wave structure) or that the chemical production term is too
efficient. Moreover, there appears to be a missing (chemical) source term in the UTLS in TM5-MP leading to a
~ 30-50% (~0.05 ppb) low bias above 600hPa, i.e. there is no significant improvement to the underestimation
in the SH $CH_2O$ column in TM5-MP when compared to mCB05 (Zeng et al, 2015). Comparing profiles shows
that the changes in the vertical distribution of $CH_2O$ at 1° x 1° are minimal in the chemical background
compared to 3° x 2°, with the main differences originating from more efficient transport out of source regions
(c.f. March). These findings are further confirmed by the comparisons of TM5-MP against TexAQS II
measurements for September and October 2006 (Fig. S18).

**9 Implications for tropospheric SO$_2$ retrieval**

The global distribution of near-surface SO$_2$ mixing ratios for both the 3° x 2° and 1° x 1° simulations are shown
in the bottom panels of Fig. S17, where the distribution shows the land-based point sources as applied from the
MACCity emission inventory. The high mixing ratios of SO$_2$ correlate with the location of strong
anthropogenic emission sources due to the relatively short atmospheric lifetime of SO$_2$ (varying between ~2
days during winter and ~19 hours during summer (Lee et al, 2011)), being rapidly oxidized to sulphate (SO$_4^=$).
Although the regional distributions are similar, the 1° x 1° simulation is able to differentiate point sources to a
much better degree, which enhances the ability of deriving more accurate emission fluxes. The *in-situ* chemical
production term of SO$_2$ from the oxidation of oceanic di-methyl sulphide (DMS) is low thus there are very low
SO$_2$ mixing ratios in the chemical background. Also shown are the regions used for validating the SO$_2$ surface
concentrations and vertical profiles (insets in FigS17; see below).
In Figure 12, we compare weekly [SO$_2$] for 2006 at a number of EMEP sites in Austria (AT02, forested), The
Netherlands (NL09, rural), Great Britain (GB43, rural) and Spain (ES10, rural), with most sites being
positioned away from strong point sources. For SO$_2$ in Europe, the main emission source is anthropogenic (e.g.
from the energy sector). High [SO$_2$] has been observed throughout the EMEP network in e.g. The Netherlands
and Spain, which is significantly higher than that measured in Central Europe (Tørseth et al., 2012). Although
the measurement uncertainty is somewhat site specific due to the different methodologies employed, it is
typically around ~1.3 ug/m$^3$ (e.g. Hamad et al., 2010). Comparing weekly averages shows that for most sites
shown there is a significant low bias at 3° x 2°, indicating inaccuracies in the MACC emission inventory and
the effect of coarsening to the model resolution. At 1° x 1° significant improvements occur in the correlations
as a result of the better temporal distribution of anthropogenic emission sources.
Table 9 provides an overview of the seasonal biases for all of the EMEP sites that measure hourly [SO$_2$], with
the biases calculated for the overpass time of tropOMI aggregated on a weekly basis. Improvements occur at 1°
x 1° for ~20% of the sites during both seasons, with the majority (~50%) of sites showing no significant
improvement (< 5%). In such instances the local [SO$_2$] is determined more by long-range transport (thus
sensitive to wash-out) rather than a local emission source, where strong mitigation practises have been
implemented in Europe over the last few decades reducing resident [SO$_2$] significantly (Tørseth et al., 2012).
For some sites there is a notable increase in biases at 1° x 1° (20% DJF, 25% JJA) indicating that too strong
local emission sources occur in the MACC inventories (e.g. ES13 and GR01). For others (e.g. ES08 and NL07)
significantly low biases occur suggesting the opposite problem.
Finally, for the vertical profiles, we make comparisons against monthly mean composites assembled from
measurements taken during INTEX-B (Fig. S19) and TexAQS II (Fig. 13) as for the other trace gas species.
For the more pristine locations there are typically low biases at 3° x 2° for all months, especially at the surface
during March indicating a significant underestimation in the emission fluxes of SO$_2$. Increasing to 1° x 1° only
provides an improved correlation for March, due to the transport in the FT being described better as shown for
NO$_2$ in Fig. 7. For April, the comparison shows a significant underestimation in the column for both
simulations, where corresponding comparisons of the vertical profiles of DMS, which acts as a key source of
$SO_2$ in the Equatorial Pacific (Alonza Gray et al., 2011), agree quite well (not shown). This points to an
appreciable biogenic source term that is currently missing from the inventories as proposed for organic nitrates
(Williams et al., 2014). For May again no significant improvement occurs at 1° x 1°, although both simulations
capture the peak in $SO_2$ mixing ratios at the top of the BL. More relevant for satellite based retrievals is the
observed column near strong anthropogenic source regions as shown in Fig. 13 over Texas during September
and October 2006. Here a clear improvement occurs at 1° x 1°, with the low bias in the BL being reduced
significantly although the integrated column is still too low. Again this is due to the underestimation in the
source emission fluxes in the anthropogenic emission inventory employed.

**10 Conclusions**

In this paper we have provided a comprehensive description of the high-resolution 1° x 1° version of TM5,
which is to be used for the purpose of providing *a-priori* columns for the satellite retrieval of trace gas columns
of $NO_2$, $CH_2O$ and $SO_2$. By performing identical simulations at a horizontal resolution of 3° x 2° and 1° x 1°,
and comparing the resulting global distributions of trace gas species, photolysis frequencies and chemical
budget terms, we quantify and validate both the near-surface and vertical distributions for the evaluation year
of 2006.
Comparing the seasonal distribution in $^{222}$Rn between resolutions we show differences in the vertical
distribution of up to ±20% at the global scale, with significantly larger impacts for specific coastal regions and
tropical oceans. In order to assess the changes in convective activity above strong $NO_x$ sources, we show that
differences of between ~2-10% (~10-20%) exist for the Northern mid-latitudes (tropics) at higher resolution,
with both weaker and stronger upwelling occurring depending on the region and the season. The magnitude of
the changes are site specific being affected by local orography. We have also made comparisons using a 1° x 1°
simulation applying the Tiedtke (1989) convection scheme, showing that ERA-interim mass-fluxes result in
less transport of $^{222}$Rn out of the boundary layer, where it has been shown that the use of ERA-interim mass-
fluxes introduces inconclusive improvements in surface $^{222}$Rn distributions (Koffi et al., 2016).
Although the impact of resolution on daily photolysis rates maybe appreciable, analysing global monthly mean
$J_{O3}$ and $J_{NO2}$ surface values over a range of conditions shows that effects are limited to ~2% and ~5-10%,
respectively. One contributing factor to this rather muted impact is that the changes in surface albedo that
occur at 1° x 1° are largest at the poles during winter, which has no impact on photo-chemistry due to the
absence of photolysing light (not shown). For cloud cover, a dominant term for determining total optical depth,
there are significant increases at 1° x 1° over the oceans, although generally related to instances of low
photochemical activity. Examining the resulting changes in $J_{O3}$ and $J_{NO2}$ which occur throughout the
tropospheric column reveals that significant differences of >10% can occur at the top of the BL at tropical
locations. Such modest changes associated with this dominant loss term result in the change in the integrated
chemical budget terms for tropospheric $O_3$ and $NO_2$ to be rather low.
Analysing the chemical budget terms for tropospheric $O_3$ shows (i) a reduction in the stratosphere-troposphere
exchange flux of ~7% to 597 Tg $O_3$ yr$^{-1}$, (ii) a repartitioning of the contribution from stratospheric down-
welling in both the Northern and Southern hemispheres, (iii) no significant change in the tropospheric burden
of $O_3$ and (iv) modest changes in the integrated chemical production and destruction terms. Comparing
simulated mixing ratios against surface measurements in Europe shows that the positive bias present in TM5
decreases by ~20% at 1° x 1° between 2-5 ppb/month. This positive bias persists throughout the vertical
column across diverse global regions regardless of the local $NO_x$ emissions, although the vertical gradient in
tropospheric $O_3$ through the tropospheric column is captured quite well.
For NO and $NO_2$ increasing horizontal resolution results in only modest differences in the zonal mean
recycling terms and the loss of $O_3$ by chemical titration. Comparisons against surface measurements in Europe
shows that there is a consistent negative bias in weekly [NO] of a few µg m$^{-3}$ associated with both too high
surface $O_3$ (enhanced NO titration) and the inaccuracy of the $NO_x$ emission inventories. For $NO_2$, the biases in
the weekly concentrations are larger and can be both positive and negative. Increasing horizontal resolution has
little effect on reducing the NO biases, but results in improvements for $NO_2$ at ~35% of the available sites, with
~45% of sites showing limited changes. Examining correlation co-efficients shows that although there is
typically a higher correlation at 1° x 1°, many sites still exhibit very low correlation or anti-correlation for
some seasons. For the tropospheric column the improvement in the comparisons is only by a few percent, with
a significant underestimation in both NO and $NO_2$ throughout the tropospheric column. Analysing the $NO/NO_2$
ratio and comparing against observations shows that although partitioning is captured in the BL there is a
significant overestimation in the upper troposphere.
Finally for $CH_2O$ and $SO_2$, which can also be retrieved from satellite measurements, the effect of increased
resolution is rather modest due to compensating changes towards the chemical budget terms. When compared
against observations there is a persistent low bias for tropospheric $CH_2O$ due to missing production terms
especially on the Free Troposphere. For $SO_2$ comparison with surface observations in Europe shows lower
biases at 20% of sites due to more accurate local emission fluxes, whereas for the majority of cases (~50%)
there is no significant change. Comparing vertical profiles shows a significant underestimation in the
tropospheric column likely associated with either missing precursors or an underestimation in the direct
emission terms.
Future updates to TM5-MP will most likely focus on developing an online Secondary Organic Aerosol
scheme, tropospheric halogen chemistry and incorporating an updated isoprene oxidation scheme involving
more intermediate species. It will also be applied in the context of an Earth System Model (EC-EARTH) for
allowing future studies concerning chemistry-climate feedbacks. When computing resources allow more
expensive simulations can be performed using the 60 vertical levels as defined in the ERA-Interim
meteorological dataset, approximately doubling the resolution of the simulations presented here. An additional
update to improve the STE would be to apply the second-order moments scheme (Prather, 1986), whose
application has been shown to capture the seasonality and magnitude of STE exchange to a better degree
(Bönisch et al., 2008). In terms of oxidative capacity, one means of reducing the tropospheric near-surface $O_3$
mixing ratios would be improve loss to land-surfaces (Hardacre et al., 2010), although mixing ratios have been
shown to be insensitive to the additional loss term to oceans, which is currently missing from many CTM's
(Ganzeveld et al., 2009). Our comparisons of $CH_2O$ and $SO_2$ show that there is a significant uncertainty of
chemical processes that affect distributions in the pristine marine environment. For instance, the physical
process of deposition seems to be under-represented possibly due to too low surface area of the surface i.e. lack
of a flat surface. The significant underestimates in $SO_2$ suggest missing biogenic sources terms, therefore more
understanding of biogenic emission terms is necessary.

**Code Availability**
The TM5-MP code can be downloaded from the SVN server hosted at KNMI, The Netherlands. A request to
generate a new user account for access can be made by e-mailing sager@knmi.nl. Any new user groups need to
agree to the protocol set out for use, where it is expected that any developments are accessible to all users after
publication of results. Attendance at 9-monthly TM5 international meetings is encouraged to avoid duplicity
and conflict of interests.

**Acknowledgements**
This research has been supported by FP7 project Quality Assurance for Essential Climate Variables
(QA4ECV), no. 607405. We thank M. van Weele for processing the MSR2 stratospheric ozone data record
used for constraining the overhead $O_3$ field and T. P. C. van Noije for updating the $SO_x$ emission estimates. We
thank V. Huijnen for providing estimates on the heterogeneous uptake co-efficients.

Table 1: Details of the reaction rate data applied for $NO_x$ and nitrogen reservoirs. The $k_0$ terms are multiplied by
the relevant air density to calculate the correct forward and backward rate constants. The reaction data and
stiochiometery are taken from Atkinson et al. (2004) accommodating the latest evaluation at http://iupac.pole-
ether.fr.

| Reactants | Products | Rate parameters |
|---|---|---|
| $NO + O_3$ | $NO_2$ | $3.0 \times 10^{-12} *\exp(-1500/T)$ |
| $NO_2 + O_3$ | $NO_3$ | $1.4 \times 10^{-13} *\exp(-2470/T)$ |
| $NO + HO_2$ | $NO_2 + OH$ | $3.3 \times 10^{-12} *\exp(270/T)$ |
| $NO + CH_3O_2$ | $CH_2O + HO_2 + NO_2$ | $2.8 \times 10^{-12} *\exp(300/T)$ |
| $OH + NO_2$ | $HNO_3$ | $k_0 = 3.2 \times 10^{-30}*(300/T)^{4.5}$ $k_\infty = 3.0 \times 10^{-11}$ |
| $NO + NO_3$ | $NO_2 + NO_2$ | $1.8 \times 10^{-11} *\exp(110/T)$ |
| $NO_2 + NO_3$ | $N_2O_5$ | $k_0 = 8.0 \times 10^{-27}*(300/T)^{3.5}$ $k_\infty = 3.0 \times 10^{-11}*(300/T)^{1.0}$ |
| $N_2O_5 + M$ | $NO_2 + NO_3$ | $k_0 = 1.3 \times 10^{-3}*(300/T)^{3.5}* \exp(-11000/T)$ $k_\infty = 9.7 \times 10^{14}*(300/T)^{-0.1}* \exp(-11080/T)$ |
| $HO_2 + NO_2$ | $HNO_4$ | $k_0 = 1.4 \times 10^{-31}*(300/T)^{3.1}$ $k_\infty = 4.0 \times 10^{-12}$ |
| $HNO_4 + M$ | $HO_2 + NO_2$ | $k_0 = 4.1 \times 10^{-5}*\exp(-10650/T)$ $k_\infty = 6.0 \times 10^{15}*\exp(-11170/T)$ |
| $OH + HNO_4$ | $NO_2$ | $1.3 \times 10^{-12} *\exp(380/T)$ |
| $OH + NO + M$ | $HONO$ | $k_0 = 7.0 \times 10^{-31}*(300/T)^{4.4}$ $k_\infty = 3.6 \times 10^{-11}*(300/T)^{0.1}$ |
| $HONO + h\upsilon$ | $OH + NO$ | |
| $OH + HONO$ | $NO_2$ | $2.5 \times 10^{-12} *\exp(260/T)$ |
| $NO_2 + CH_3C(O)O_2$ | $PAN$ | $k_0 = 3.28 \times 10^{-28}*(300/T)^{6.87}$ $k_\infty = 1.125 \times 10^{-11}*(300/T)^{1.105}$ |
| $PAN$ | $NO_2 + CH_3C(O)O_2$ | $k_0 = 1.1 \times 10^{-5}*\exp(-10100/T)$ $k_\infty = 1.9 \times 10^{17}*\exp(-14100/T)$ |

| | | |
|---|---|---|
| PAN + hυ | CH$_3$C(O)O$_2$ + NO$_2$ | |
| | CH$_3$O$_2$ + NO$_3$ | |
| CH$_3$O$_2$ + NO$_2$ | CH$_3$O$_2$NO$_2$ | $k_0$= 2.5 x 10$^{-30}$*(300/T) |
| | | $k_\infty$= 1.8 x 10$^{-11}$ |
| CH$_3$O$_2$NO$_2$ | CH$_3$O$_2$ + NO$_2$ | $k_0$= 9.0 x 10$^{-5}$*exp(-9690/T) |
| | | $k_\infty$= 1.1 x 10$^{16}$*exp(-10560/T) |
| NO$_3$ + HO$_2$ | HNO$_3$ | 4.0 x 10$^{-12}$ |


Table 2: Details of updates made to the reaction data and stoichiometry of the modified CB05 chemical
mechanism for other reactions. Data is taken from the following: [1] Atkinson et al. (2004) accommodating the
latest evaluation at http://iupac.pole-ether.fr, [2] branching ratio (R) equal to $1/(1+498.*exp(-1160./T)$, [3]
Yarwood et al. (2005), [4] Sander et al. (2011), [5], Atkinson et al. (2006), [6] Emmons et al. (2010), [7]
Hauglustaine et al. (2014), [8] rate assumed equal to $NH_2$ analogue, [9] assumed to be equal to $HNO_4$ after
Browne et al. (2011) and E is an estimated value.

| Reactants | Products | Rate expression | Ref. |
|---|---|---|---|
| $CH_3O_2 + HO_2$ | $CH_3OOH$ | $3.8 \times 10^{-13}*exp(750/T)*R$ | [1],[2] |
| $CH_3O_2 + HO_2$ | $CH_2O$ | $3.8 \times 10^{-13} *exp(750/T)$ $*(1-R)$ | [1],[2] |
| $CH_3O_2 + CH_3O_2$ | $1.37CH_2O + 0.74HO_2 + 0.63CH_3OH$ | $9.5 \times 10^{-14}*exp(390/T)$ | [3],[4] |
| $OH + C_3H_8$ | $i\text{-}C_3H_7O_2$ | $7.6 \times 10^{-12} *exp(-585/T)$ | [5],[6] |
| $NO + IC_3H_7O_2$ | $0.82CH_3COCH_3 + HO_2 + 0.27ALD2$ $+ NO_2$ | $4.2 \times 10^{-12} *exp(180/T)$ | [6] |
| $HO_2 + IC_3H_7O_2$ | $ROOH$ | $7.5 \times 10^{-13} *exp(700/T)$ | [6] |
| $OH + C_3H_6$ | $C_3H_6O_2$ | $k_0= 8.0 \times 10^{-27}*(-300/T)^{3.5}$ $k_\infty=3.0 \times 10^{-11}*(-300/T)^{1.0}$ | [5],[6] |
| $NO_3 + C_5H_8$ | $0.2ISPD + XO_2 + 0.8HO_2 +$ $0.8ORGNTR + 0.8ALD2 +$ $2.4 PAR + 0.2 NO_2$ | $2.95 \times 10^{-12}*exp(465/T)$ | [5] |
| $NO + C_3H_6O_2$ | $ALD2 + CH_2O +$ $HO_2 + NO_2$ | $4.2 \times 10^{-12} *exp(180/T)$ | [6] |
| $HO_2 + C_3H_6O_2$ | $ROOH$ | $7.5 \times 10^{-13} *exp(700/T)$ | [6] |
| $NO_3 + DMS$ | $SO_2 + HNO_3$ | $1.9 \times 10^{-13} *exp(520/T)$ | [1] |
| $NH_2 + OH$ | | $3.4 \times 10^{-11}$ | [E] |
| $NH_2 + HO_2$ | $NH_3$ | $3.4 \times 10^{-11}$ | [4],[7] |
| $NH_2 + O_3$ | $NH_2O_2$ | $4.3 \times 10^{-12}*exp(-930/T)$ | [4],[7] |
| $NH_2 + O_2$ | $NO$ | $6.0 \times 10^{-21}$ | [1],[7] |
| $NH_2O_2 + NO$ | $NH_2 + NO_2$ | $4.0 \times 10^{-12}*exp(450/T)$ | [7],[8] |
| $NH_2O_2 + O_3$ | $NH_2$ | $4.3 \times 10^{-12}*exp(-930/T)$ | [7],[8] |
| $NH_2O_2 + HO_2$ | $NH_2$ | $3.4 \times 10^{-11}$ | [8] |
| $CH_3O_2NO_2 + h\upsilon$ | $CH_3O_2 + NO_2$ | | [9] |

| | | |
|---|---|---|
| $CH_3O_2NO_2 + h\upsilon$ | $CH_2O + HO_2 + NO_3$ | [9] |
| $HO_2$ + aero | $0.5H_2O_2$ | |
| $NO_3$ + aero | $HNO_3$ | [6] |


Table 3: The zonally segregated emission totals introduced into TM5-MP for the year 2006. All
organic hydrocarbons are given in Tg C yr$^{-1}$, except for CO (Tg CO yr$^{-1}$), CH$_2$O (Tg CH$_2$O yr$^{-1}$) and
CH$_3$OH (Tg CH$_3$OH yr$^{-1}$). All NO$_x$ emissions are introduced as NO (Tg N yr$^{-1}$). For SO$_2$ emission
totals are given as Tg SO$_2$ yr$^{-1}$ and for NH$_3$ as Tg NH$_3$ yr$^{-1}$. No direct emissions occur for HNO$_3$, PAN,
ORGNTR, HONO, N$_2$O$_5$, NO$_2$, CH$_3$O$_2$NO$_2$ or O$_3$.

| Species Tg Yr$^{-1}$ | Global | 30-90°S | 30S-30°N | 30-90°N |
|---|---|---|---|---|
| CO | 1081.0 | 24.4 | 755.1 | 301.27 |
| NO$_x$ (as N) | 49.0 | 1.5 | 24.0 | 23.6 |
| SO$_2$ | 117.0 | 3.0 | 49.2 | 64.3 |
| DMS (as S) | 19.2 | 6.7 | 9.3 | 3.2 |
| NH$_3$ | 56.6 | 3.1 | 27.9 | 25.6 |
| CH$_2$O | 13.5 | 0.3 | 10.5 | 2.7 |
| PAR | 34.1 | 0.7 | 18.5 | 14.9 |
| OLE | 22.4 | 0.9 | 16.6 | 4.9 |
| ALD2 | 13.4 | 0.4 | 11.2 | 1.8 |
| CH$_3$CHCHO | 2.2 | 0.0 | 1.2 | 1.0 |
| CH$_3$OH | 100.7 | 3.3 | 82.5 | 14.9 |
| CH$_3$CH$_2$OH | 70.4 | 2.8 | 52.6 | 15.1 |
| C$_2$H$_4$ | 25.9 | 1.0 | 19.0 | 5.9 |
| C$_2$H$_6$ | 6.1 | 0.3 | 5.3 | 1.5 |
| C$_3$H$_8$ | 5.6 | 0.4 | 3.6 | 1.6 |
| C$_3$H$_6$ | 19.6 | 0.9 | 14.8 | 3.9 |
| CH$_3$COCH$_3$ | 27.4 | 0.8 | 22.0 | 4.6 |
| HCOOH | 1.8 | 0.0 | 1.5 | 0.3 |
| CH$_3$COOH | 7.1 | 0.1 | 6.0 | 1.0 |
| C$_5$H$_8$ | 510.0 | 23.2 | 441.9 | 45.0 |
| C$_{10}$H$_{16}$ | 85.4 | 2.3 | 70.2 | 12.9 |


Table 4: The tropospheric chemical budget terms and burden for $O_3$ during 2006 for the 1° x 1°
simulation, with all quantities being given in Tg $O_3$ yr$^{-1}$. The associated percentage changes are given
when comparing against the 3° x 2° simulation (equal to (1° x 1°)-( 3° x 2°)/3° x 2°). The definition of
the chemical tropopause and the calculation of the STE are calculated using the methodology outlined
in Stevenson et al. (2006). The stratospheric nudging term refers to total change in the mass of $O_3$ in
the stratospheric column when constraining zonal distributions towards observational values from the
MSR (Huijnen et al., 2010). The contribution to each term from the SH extra-tropics/tropics/NH
extra-tropics regions (defined as 90-30°S/30°S-30°N/30-90°N) are provided.

| Term | Global | % | SH | % | Tropics | % | NH | % |
|---|---|---|---|---|---|---|---|---|
| Net STE | 579 | -6.7 | | | | | | |
| Strat. Nudging | 1440 | -0.7 | -224 | 2.8 | 1615 | - | 49 | 5.8 |
| Trop.Chem.Prod | 5532 | -1.9 | 389 | -2.2 | 3938 | -3.5 | 1206 | -2.2 |
| Trop.Chem.Loss | 5162 | -2.4 | 440 | -1.0 | 3869 | -2.5 | 853 | -2.8 |
| $BO_3$ | 378 | -2.0 | 72 | 1.7 | 203 | -2.3 | 104 | -3.4 |
| Strat $BO_3$ | 80 | -2.0 | 23 | 9.1 | 38 | -6.5 | 24 | -2.0 |
| | | | | | | | | |
| Deposition | 949 | 0.8 | 115 | 0.6 | 465 | - | 369 | 1.9 |
| $O_3S$ Deposition | 97 | 5.0 | 19 | 7.5 | 37 | -1.2 | 42 | 10.0 |

Table 5: The annual NO to $NO_2$ re-cycling terms involving peroxy-radicals given in Tg N yr$^{-1}$ for
2006 at 1° x 1° resolution. In mCB05v2 $XO_2$ represents lumped alkyl-peroxy radicals (Yarwood et al,
2005). The NO + $RO_2$ term is an aggregate of numerous specific peroxy-radical conversion terms in
the modified CB05 mechanism (Williams et al., 2013; Tables 1 and 2). Also provided are the
approximate percentage differences when comparing with 3° x 2° (equal to (1° x 1°)-( 3° x 2°)/3° x 2°).
The chemical tropopause is defined using the methodology outlined in Stevenson et al. (2006).

| Reaction | Global | % | SH | % | Tropics | % | NH | % |
|---|---|---|---|---|---|---|---|---|
| NO + $HO_2$ | 1058 | -1.2 | 79 | -1.2 | 740 | -1.9 | 239 | 0.6 |
| NO + $CH_3O_2$ | 407 | -2.2 | 31 | -2.6 | 294 | -2.8 | 82 | -0.5 |
| NO + $XO_2$ | 147 | -2.1 | 7 | -3.6 | 111 | -2.6 | 29 | 0.2 |
| NO + $RO_2$ | 9.4 | -4.4 | 0.4 | -2.6 | 6.3 | -4.4 | 2.7 | -4.9 |
| | | | | | | | | |
| NO + $O_3$ | 5403 | 0.1 | 518 | 7.5 | 2933 | -3.9 | 1953 | 4.0 |


Table 6: The seasonal mean absolute biases as calculated using weekly [NO] values (µg m$^{-3}$). The weekly
means are composed from daily measurements taken at 13:00 for DJF and JJA (given as the difference in the
measurements-model). Values are shown for both the 3º x 2º and 1º x 1º simulations for all stations with
available data. Those with differences < 5% are considered to exhibit no discernible change in the bias.

| EMEP Station | Lat | Lon | DJF 3º x 2º | DJF 1º x 1º | JJA 3º x 2º | JJA 1º x 1º |
|---|---|---|---|---|---|---|
| CH01 | 46.32 | 7.59 | -0.01 | -0.01 | 0.00 | -0.01 |
| CZ03 | 49.35 | 15.50 | -4.05 | -3.30 | -1.61 | -1.35 |
| DE43 | 47.48 | 11.10 | -2.37 | -2.36 | -0.47 | -0.48 |
| DK05 | 54.44 | 10.44 | -2.51 | -2.61 | -1.29 | -1.51 |
| ES07 | 58.23 | 21.49 | -3.80 | -3.84 | -1.45 | -1.48 |
| ES08 | 43.26 | -4.51 | -2.08 | -2.09 | -1.00 | -1.01 |
| ES09 | 41.16 | -3.80 | -0.93 | -0.93 | -1.07 | -1.07 |
| ES10 | 38.28 | 3.19 | -1.14 | -1.24 | -0.75 | -0.88 |
| ES11 | 39.50 | -6.55 | -1.07 | -1.07 | -0.44 | -0.45 |
| ES12 | 41.17 | -1.60 | -1.34 | -1.34 | -0.95 | -0.95 |
| ES13 | 41.24 | -5.52 | -2.50 | -1.90 | -0.74 | -0.62 |
| ES14 | 39.31 | 0.43 | -2.21 | -2.20 | -1.27 | -1.27 |
| ES15 | 43.13 | -4.21 | -1.62 | -1.61 | -0.99 | -1.00 |
| ES16 | 43.37 | -7.41 | -2.39 | -2.39 | -1.11 | -1.11 |
| FR13 | 46.39 | 0.11 | -1.90 | -1.94 | -0.52 | -0.52 |
| FR15 | 55.18 | 0.45 | -3.09 | -3.04 | -1.51 | -1.58 |
| GB02 | 50.35 | -3.12 | -1.23 | -1.23 | -0.93 | -0.92 |
| GB13 | 54.20 | -3.42 | -1.28 | -1.32 | -0.58 | -0.55 |
| GB14 | 52.30 | -0.48 | -3.03 | -3.04 | -0.98 | -0.98 |
| GB31 | 53.23 | -3.11 | -1.74 | -1.74 | -0.90 | -0.91 |
| GB37 | 50.47 | -1.45 | -3.09 | -3.08 | -1.22 | -1.21 |
| GB38 | 51.13 | 0.10 | -2.87 | -2.78 | -1.92 | -1.72 |
| GB44 | 51.17 | -3.20 | -1.65 | -1.45 | -0.27 | -0.67 |
| GB45 | 52.17 | 0.17 | -1.80 | 0.11 | 0.20 | 0.19 |
| GB51 | 52.33 | 0.46 | -3.68 | -3.42 | -1.29 | -1.17 |
| NL91 | 52.18 | 4.30 | -4.47 | -3.51 | -1.98 | -1.86 |


Table 7: As for Table 5 except for NO₂.

| EMEP Station | Lat | Lon | DJF 3º x 2º | DJF 1º x 1º | JJA 3º x 2º | JJA 1º x 1º |
|---|---|---|---|---|---|---|
| BE32 | 50.30 | 4.59 | 10.56 | 1.27 | 1.69 | -2.13 |
| CH01 | 46.32 | 7.59 | -0.04 | -0.03 | -0.02 | -0.01 |
| CZ03 | 49.35 | 15.50 | -4.04 | 0.03 | -1.62 | 0.34 |
| DE43 | 47.48 | 11.10 | -2.37 | -2.36 | -0.49 | -0.48 |
| DK05 | 54.44 | 10.44 | 6.07 | 5.56 | 1.04 | -0.02 |
| ES07 | 58.23 | 21.49 | -1.13 | -1.79 | -0.96 | -1.15 |
| ES08 | 43.26 | -4.51 | -2.01 | -2.10 | -0.93 | -1.01 |
| ES09 | 41.16 | -3.80 | -0.95 | -0.94 | -1.07 | -1.07 |
| ES10 | 42.19 | 3.19 | 2.18 | 1.46 | 0.92 | -0.01 |
| ES11 | 38.28 | -6.55 | -1.08 | -1.08 | -0.44 | -0.44 |
| ES12 | 39.50 | -1.60 | -1.36 | -1.35 | -0.95 | -0.95 |
| ES13 | 41.17 | -5.52 | -2.51 | -0.29 | -0.74 | 0.19 |
| ES14 | 41.24 | 0.43 | -2.22 | -2.21 | -1.27 | -1.26 |
| ES15 | 39.31 | -4.21 | -1.64 | -1.63 | -0.99 | -0.99 |
| ES16 | 43.13 | -7.41 | -2.40 | -2.39 | -1.11 | -1.11 |
| FI09 | 59.46 | 21.22 | 0.79 | -0.91 | -0.10 | -0.53 |
| FI37 | 60.31 | 27.41 | 10.04 | 9.60 | 1.70 | 0.92 |
| FI96 | 62.35 | 24.11 | 0.40 | 0.34 | 0.31 | 0.17 |
| FR13 | 43.37 | 0.11 | -1.96 | -1.95 | -0.54 | -0.54 |
| FR15 | 46.39 | 0.45 | -3.84 | -3.91 | -1.78 | -1.81 |
| GB02 | 55.18 | -3.12 | 3.12 | 3.42 | 0.82 | 0.54 |
| GB13 | 50.35 | -3.42 | -2.17 | -2.13 | -1.00 | -0.98 |
| GB14 | 54.20 | -0.48 | 5.02 | 3.93 | 1.17 | 1.50 |
| GB31 | 52.30 | -3.11 | -1.71 | -1.71 | -0.90 | -0.90 |
| GB37 | 53.23 | -1.45 | -3.10 | -3.08 | -1.25 | -1.25 |
| GB38 | 50.47 | 0.10 | -4.05 | -4.03 | -2.20 | -2.19 |
| GB44 | 51.13 | -3.20 | 6.21 | 6.69 | 0.61 | 1.42 |
| GB45 | 52.17 | -0.17 | 6.28 | 6.00 | 3.48 | 1.94 |
| GB51 | 52.33 | -0.46 | 12.56 | 16.60 | 4.11 | 4.50 |
| GR01 | 41.45 | 42.49 | 0.50 | 2.07 | 0.40 | 1.05 |
| NL09 | 53.2 | 6.16 | -3.21 | -1.63 | -1.18 | -0.19 |
| NL10 | 51.32 | 5.51 | 3.48 | 3.52 | 0.92 | -0.29 |
| NL91 | 52.18 | 4.30 | 11.81 | 6.42 | 1.48 | -0.81 |



Table 8: The tropospheric chemical budget for the $CH_2O$ given in Tg $CH_2O$ yr$^{-1}$ during 2006 for the 1°
x 1° simulation. Percentage differences are shown against the corresponding 3° x 2° simulation.

| Budget Term | Global | % | SH | % | Tropics | % | NH | % |
|---|---|---|---|---|---|---|---|---|
| $CH_2O$ CP | 1919 | -1.1 | 147 | -0.3 | 1491 | -1.0 | 281 | -2.0 |
| $CH_2O$ CD | 1739 | -1.6 | 134 | -0.5 | 1349 | -1.1 | 256 | -2.3 |
| $CH_2O$ Dep. | 193 | 3.1 | 15 | 2.0 | 149 | 3.9 | 29 | - |

Table 9: The seasonal mean biases of daily [$SO_2$] (µg m$^{-3}$) at 13:00 for DJF and JJA, when taking the
difference between measurements-model values. Values are shown for both the 3° x 2° and 1° x 1°
simulations. Those with differences < 5% are considered to exhibit no discernible change in the bias.

| EMEP Station | Lat | Lon | DJF 3° x 2° | DJF 1° x 1° | JJA 3° x 2° | JJA 1° x 1° |
|---|---|---|---|---|---|---|
| AT02 | 47.46 | 16.46 | -3.34 | -3.15 | -0.89 | -0.53 |
| AT05 | 46.40 | 12.58 | -0.42 | -0.41 | -0.14 | -0.14 |
| AT48 | 47.50 | 14.26 | -0.64 | -0.63 | -0.14 | -0.15 |
| CZ03 | 49.35 | 15.50 | -3.52 | 3.65 | -0.69 | 0.64 |
| ES07 | 37.14 | -3.32 | 1.22 | 0.73 | 0.38 | 0.31 |
| ES08 | 43.26 | -4.51 | -2.98 | -3.21 | -1.19 | -1.58 |
| ES09 | 41.16 | -3.80 | -0.62 | -0.61 | -0.42 | -0.42 |
| ES10 | 42.19 | 3.19 | 2.37 | 2.45 | 1.93 | 1.53 |
| ES11 | 38.28 | -6.55 | -0.63 | -0.61 | -0.70 | -0.70 |
| ES12 | 39.50 | -1.60 | -0.47 | -0.45 | -0.32 | -0.32 |
| ES13 | 41.17 | -5.52 | -0.81 | 2.71 | -0.78 | 0.55 |
| ES14 | 41.24 | 0.43 | -0.70 | -0.67 | -0.47 | -0.47 |
| ES15 | 39.31 | -4.21 | -0.40 | -0.37 | -0.45 | -0.46 |
| ES16 | 43.13 | -7.41 | -3.84 | -3.82 | -1.66 | -1.66 |
| GB37 | 52.30 | -3.11 | -2.92 | -2.91 | -1.72 | -1.72 |
| GB38 | 53.23 | -1.45 | 2.93 | 2.75 | 0.39 | 1.33 |
| GB43 | 51.14 | -4.42 | -1.49 | 4.77 | -2.03 | -0.63 |
| GB45 | 52.17 | -0.17 | 3.87 | 7.20 | 1.01 | 1.77 |
| GR01 | 38.22 | 23.50 | 1.70 | 2.77 | 0.74 | 1.50 |
| NL07 | 52.50 | 6.34 | 2.58 | -1.67 | 0.46 | -1.03 |
| NL08 | 52.70 | 5.12 | 1.77 | 1.56 | -0.14 | -0.30 |
| NL09 | 53.2 | 6.16 | 2.53 | 2.16 | 0.47 | 0.27 |


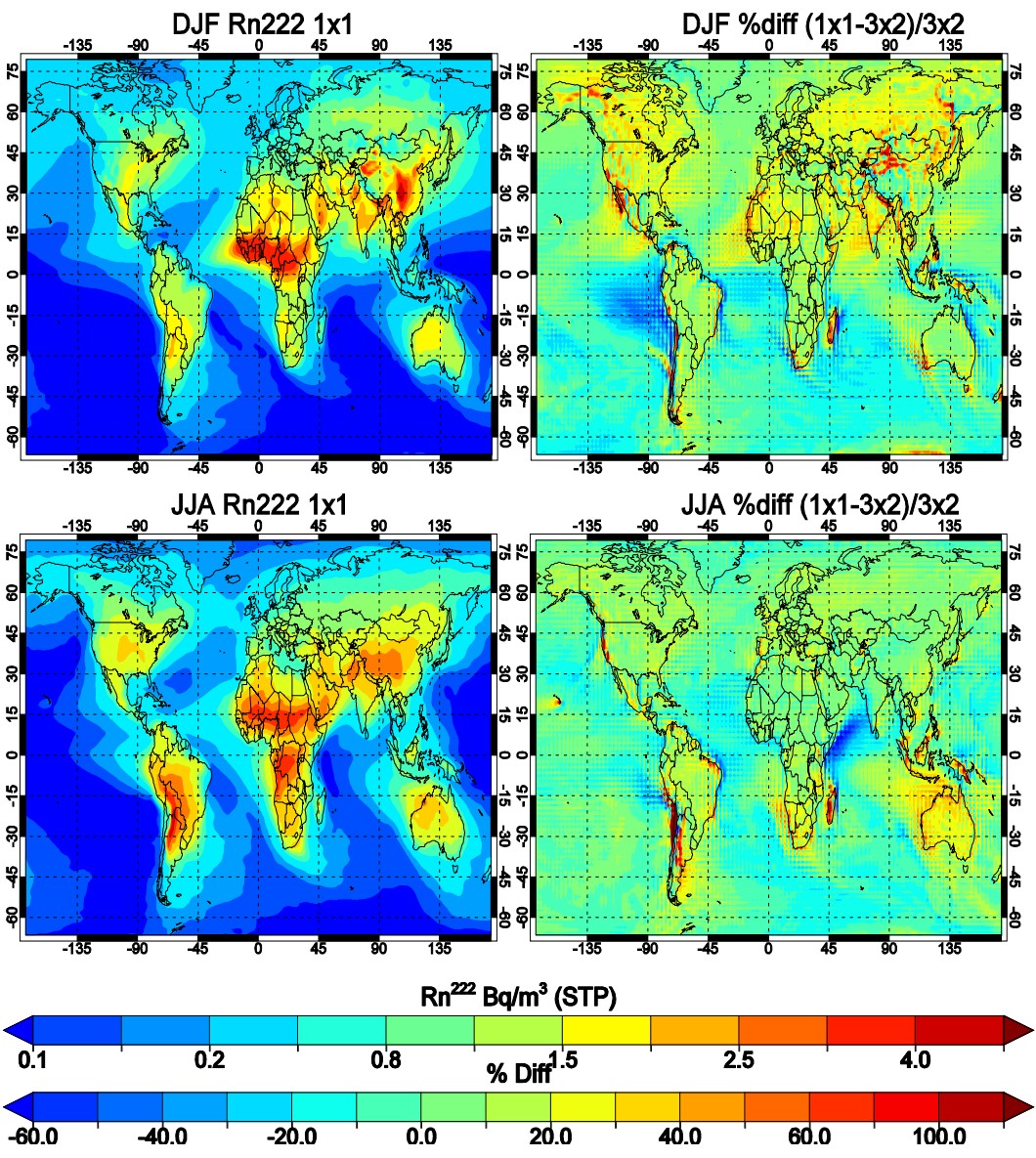

Figure 1: The seasonal distributions of Rn[222] averaged between 800 and 900hPa for DJF (top) and JJA (bottom) for the 3º x 2º (left) and 1º x 1º (right) simulation, with the associated percentage differences when compared against the 3º x 2º simulation.

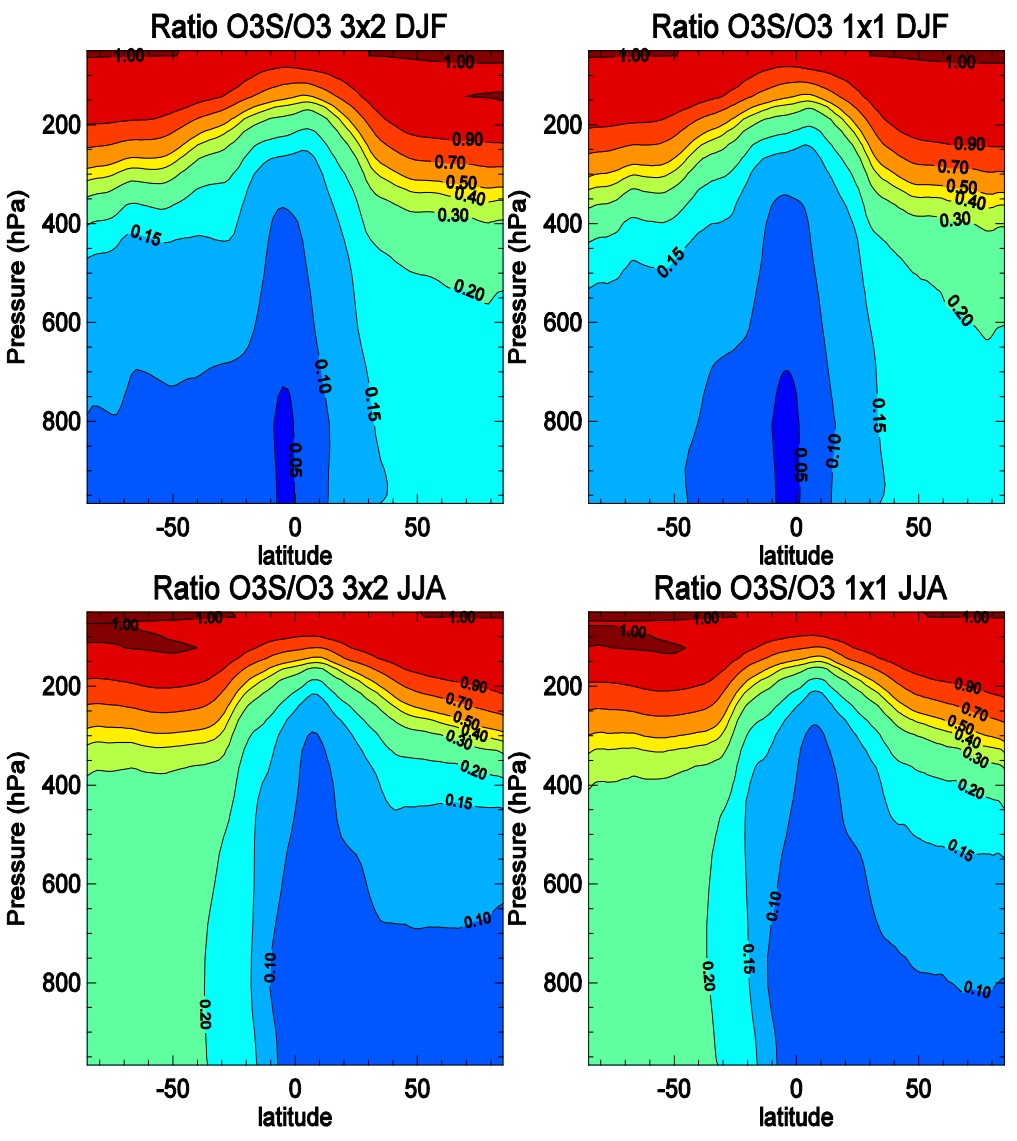

Figure 2: Zonal mean seasonal distribution of the TM5-MP $O_3S/O_3$ ratio for the 3º x 2º (left) and 1º x 1º (right) simulations.


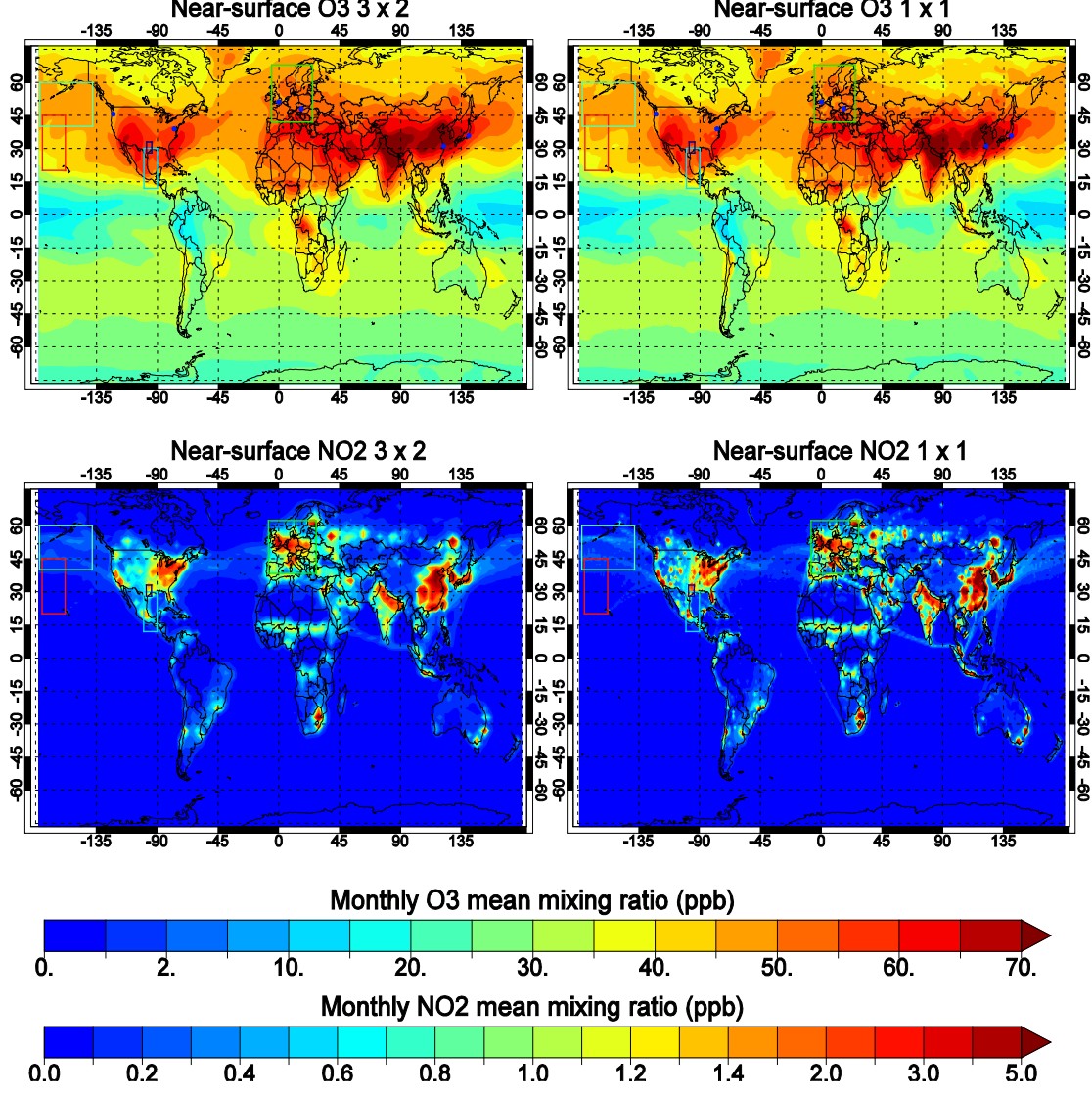

Figure 3: The near-surface distribution in tropospheric $O_3$ (top) and $NO_2$ (bottom) for May 2006 from the
3° x 2° (left) and 1° x 1° (right) TM5-MP simulations. The blue points represent the location of the
MOZAIC airports used for comparisons. Also shown are the locations of the INTEXB and Texas-AQSII
measurement campaigns, and the extent of the EMEP network in the European domain, used for the
validation of the resulting $O_3$ and $NO_2$ distributions.




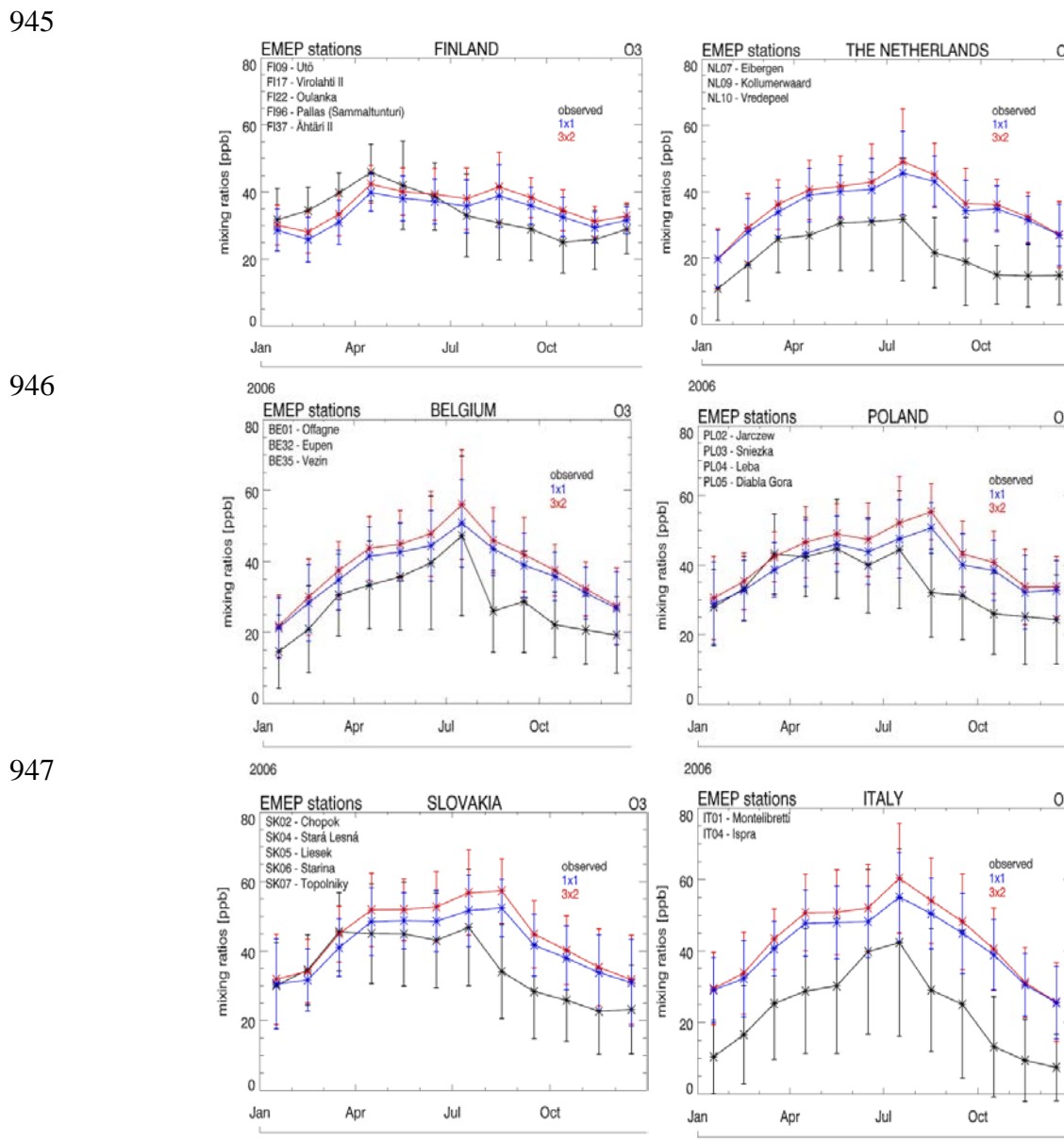

Figure 4: Comparisons of the seasonal variability in TM5-MP mass mixing ratios for
surface O$_3$ against composites of measurements taken across the EMEP monitoring
network for 2006. Both the co-located TM5-MP 3º x 2º and 1º x 1º monthly mean
values are shown, along with the 1-σ variability for Finland, The Netherlands, Belgium,
Poland, Slovakia and Italy. Individual stations that are aggregated are given in the
panels.




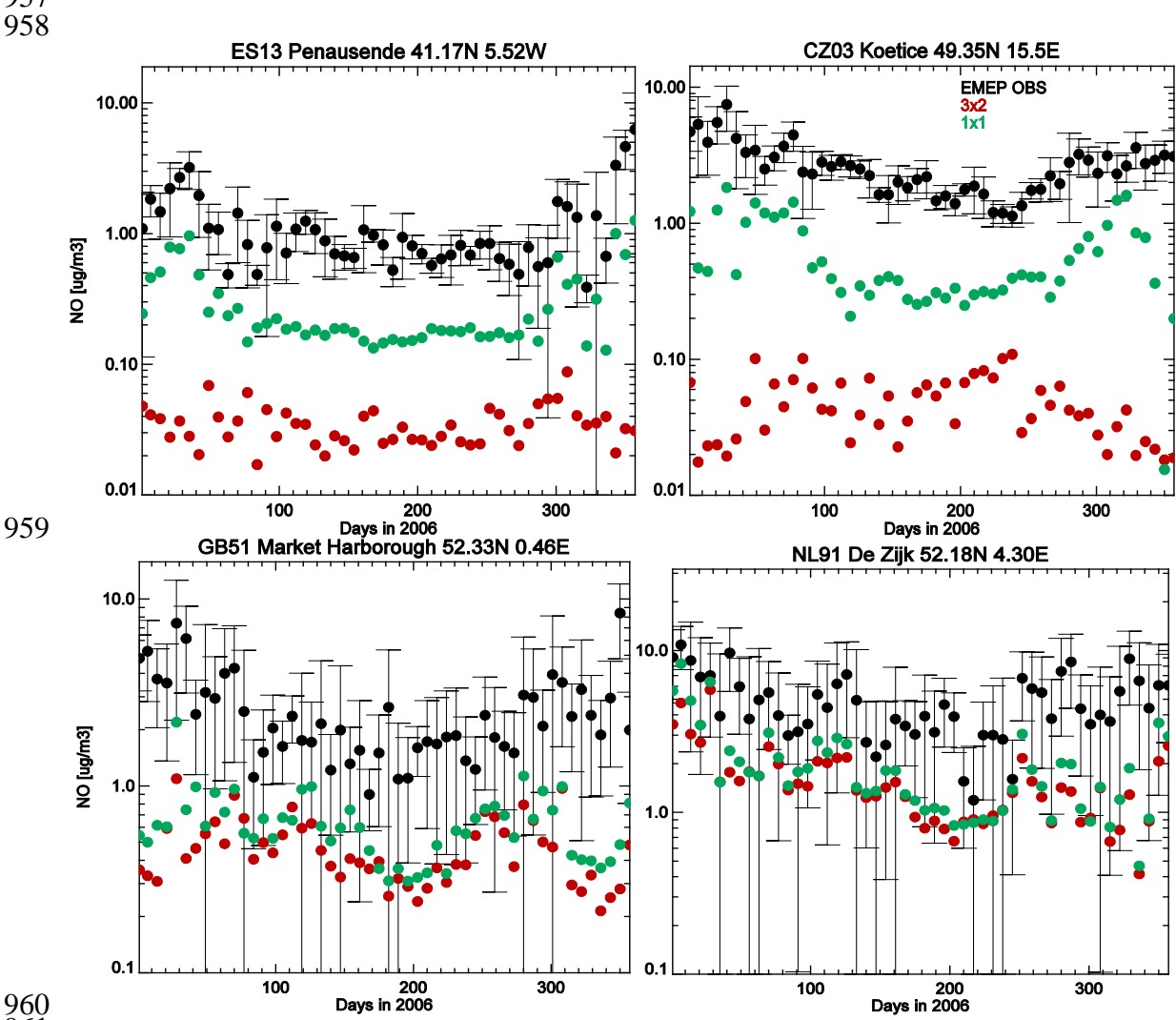

Figure 5: Comparison of TM5-MP weekly [NO] sampled at 13:00 UT each day during 2006 with observed [NO] ($\mu g\ m^{-3}$). The selected sites shown are in the Czech Republic (top left), Spain (top right), Great Britain (bottom left) and The Netherlands (bottom right).


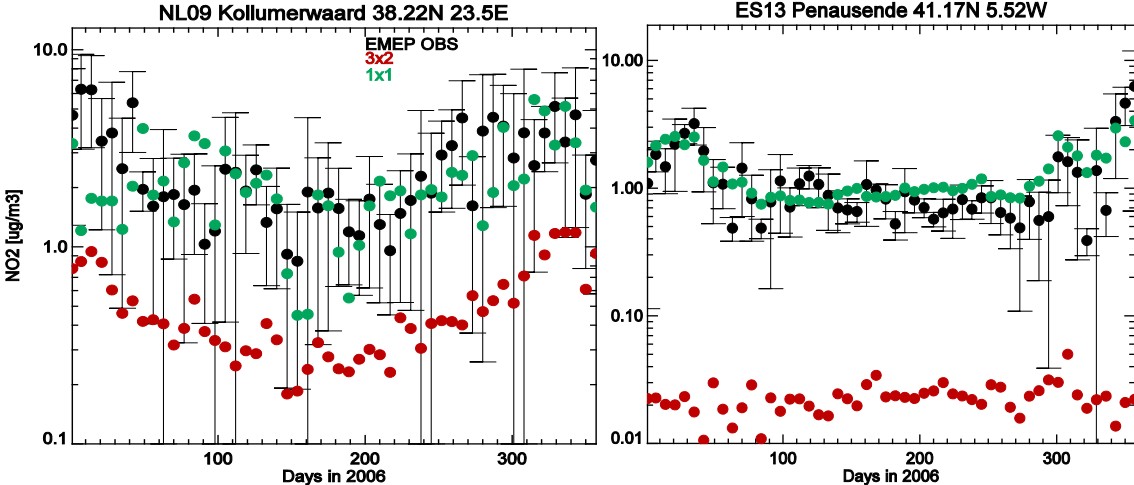

Figure 6: Comparison of weekly TM5-MP [NO₂] sampled at 13:00 UT each day during 2006 with
observed [NO₂] ($\mu$g m$^{-3}$). The selected sites shown are in The Netherlands (left) and Spain (right).

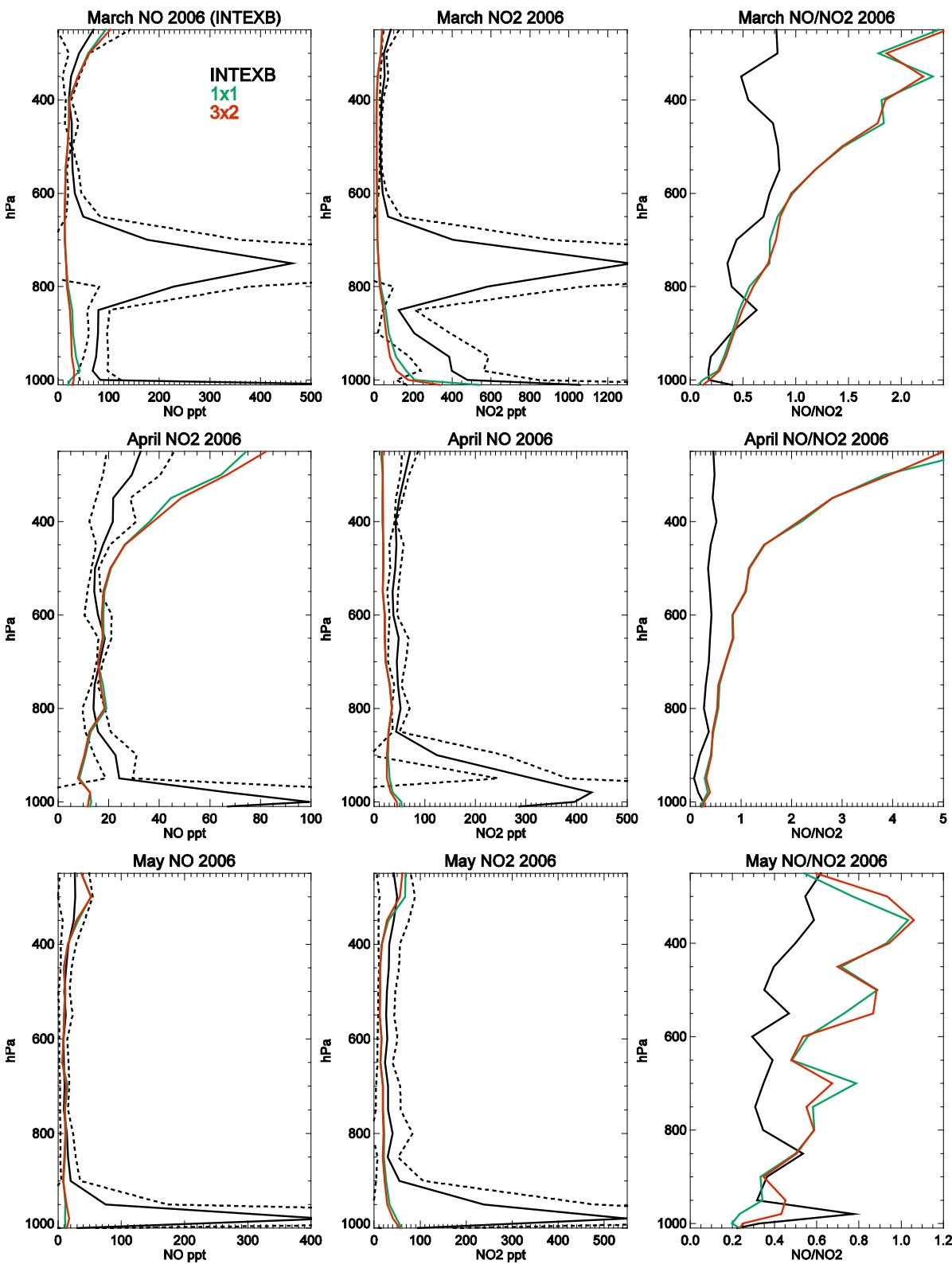

Figure 7: Monthly mean comparisons of NO (left), $NO_2$ (middle) and the resulting $NO/NO_2$ ratio (right) from INTEX-B measurements and TM5-MP simulations. The dotted line represents the 1-σ deviation in the mean of the measurements. For details of the locations for each month the reader is referred to Singh et al. (2009).

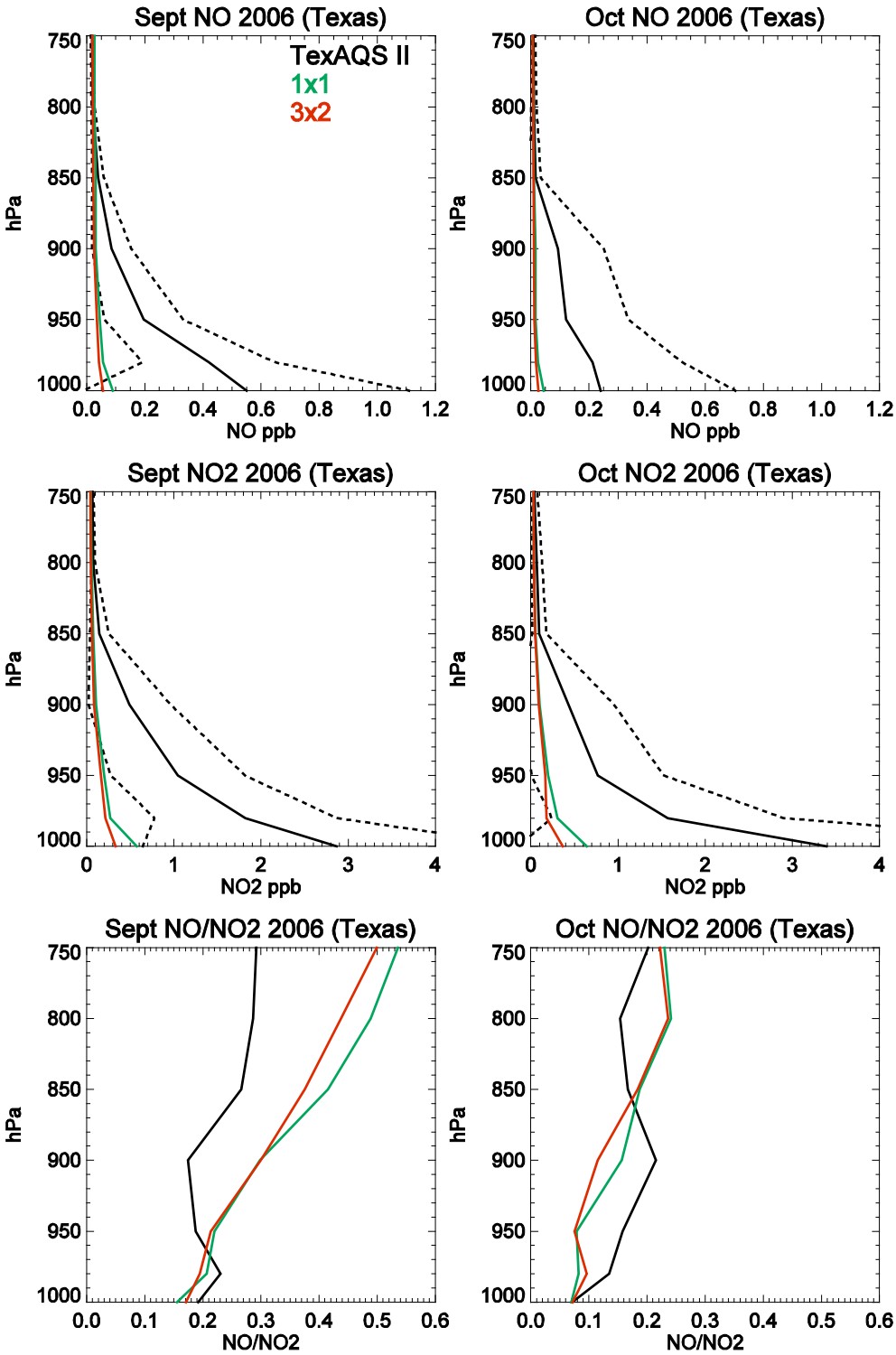

Figure 8: Monthly mean comparisons of NO (left), $NO_2$ (middle) and the resulting $NO/NO_2$ ratio (right) from the TexAQSII campaign during September and October 2006 and TM5-MP simulations. The dotted line represents the 1-σ deviation in the mean of the measurements. For details of the locations for each month the reader is referred to Parrish et al. (2009).

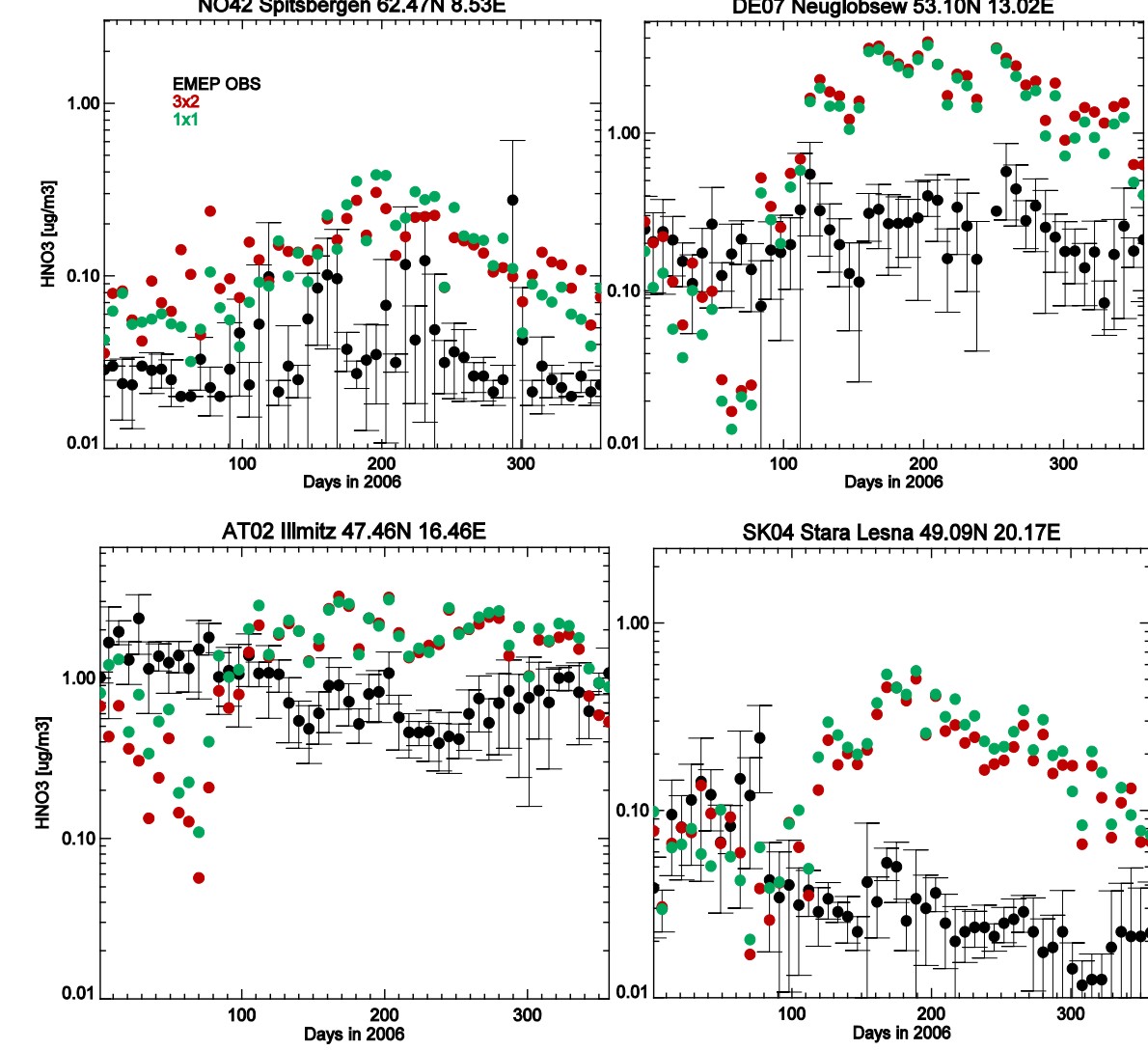




Figure 9: Comparison of weekly [HNO$_3$] (µg m$^{-3}$) from both 3° x 2° and 1° x 1° simulations at 4 selected
EMEP sites for 2006. The 1-σ deviation in the weekly observations are shown as error bars. The selected
sites shown are in Norway (top left), Germany (top right), Austria (bottom left) and Slovakia (bottom
right).


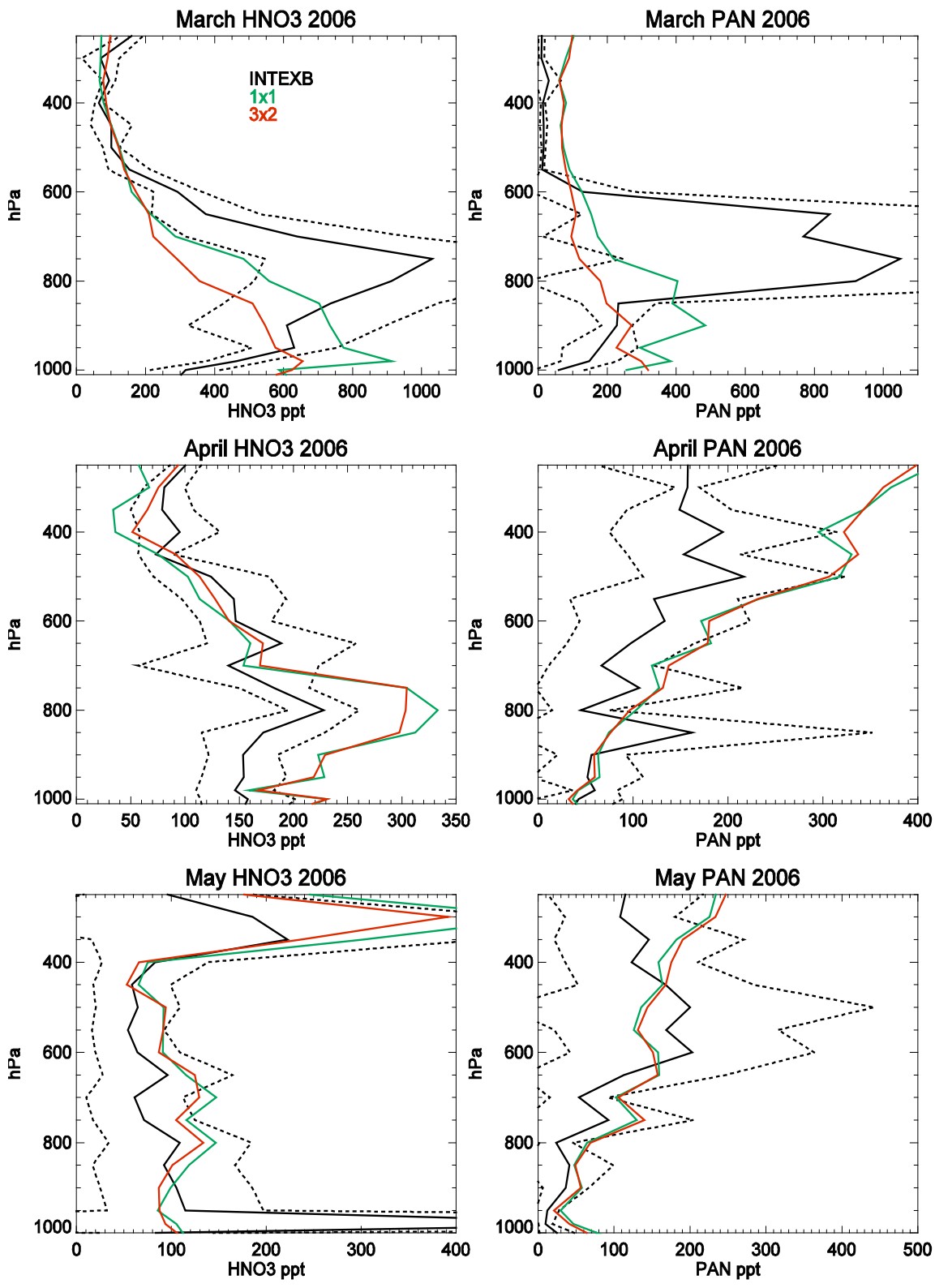

Figure 10: Monthly mean comparisons of HNO$_3$ (left) and PAN (right) from the INTEX-B measurements and TM5-MP simulations. The dotted line represents the 1-σ deviation in the mean of the measurements. For details of the locations for each month the reader is referred to Singh et al. (2009).

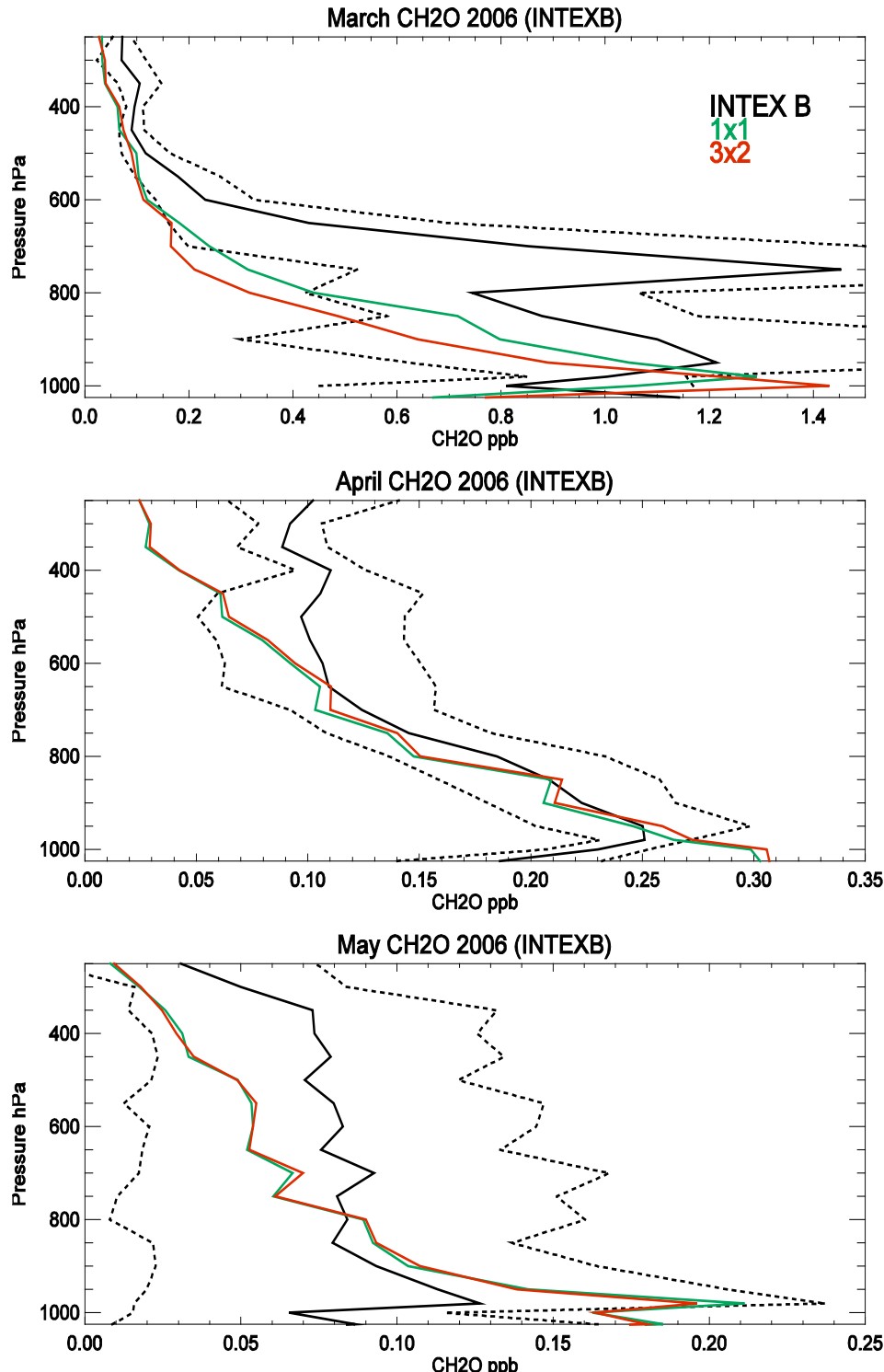

Figure 11: Comparisons of the vertical distribution of $CH_2O$ from both 3° x 2° and 1° x 1°
simulations against measurements made as part of the INTEX-B during 2006. The dotted line
represents the 1-σ deviation in the mean of the measurements. For details on the exact location of
the flights the reader is referred to Parrish et al. (2009).


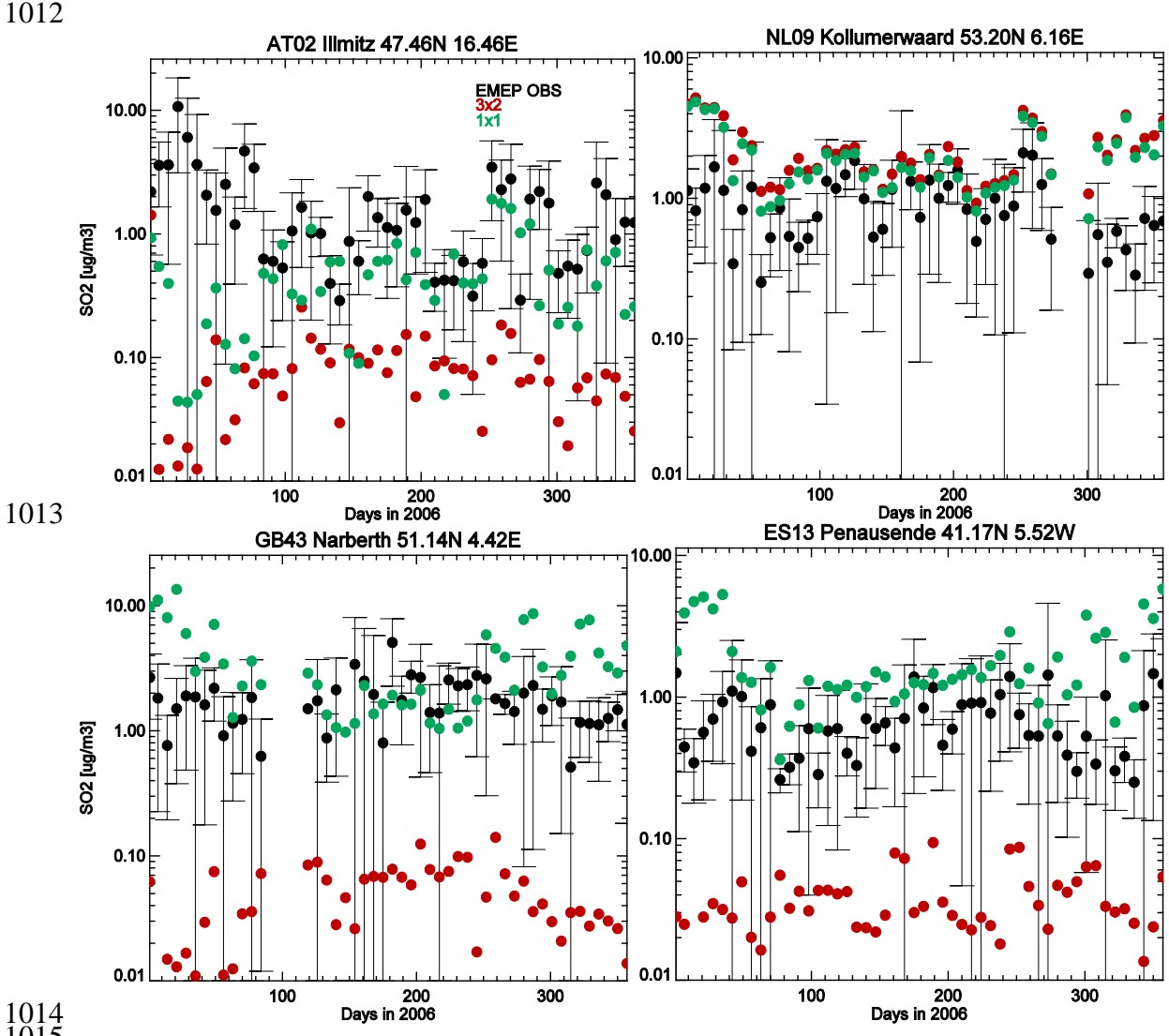


Figure 12: Comparison of weekly [SO$_2$] (µg m$^{-3}$) at 13:00 from both the 3º x 2º and 1º x 1º simulations
at 4 selected EMEP sites for 2006. The selected sites shown are in Austria (top left), the Netherlands
(top right), Great Britain (bottom left) and Spain (bottom right).


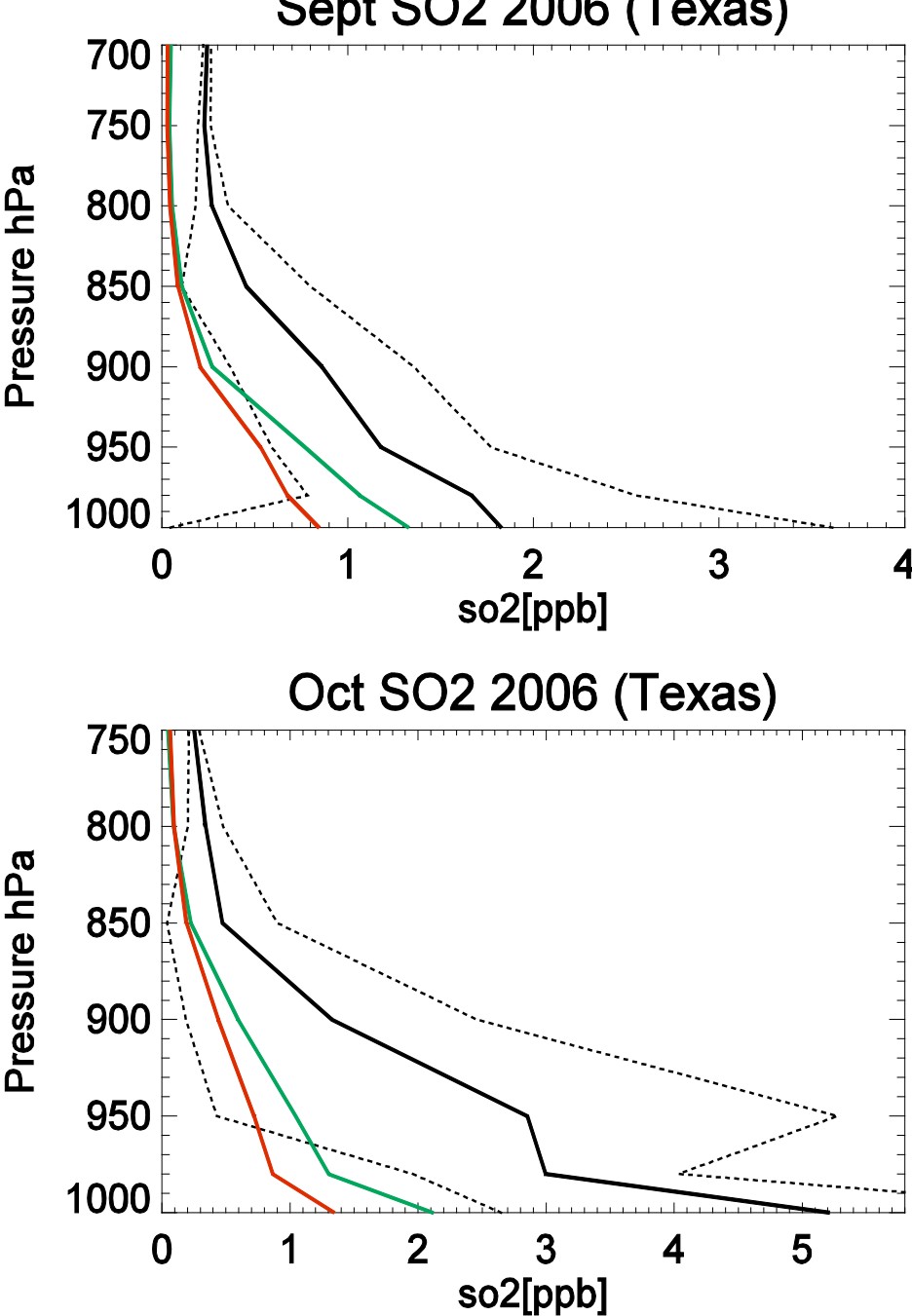

Figure 13: Comparisons of the monthly tropospheric SO$_2$ profile assembled from data
taken during September and October 2006 as part of TexAQS II. The 1-σ deviation of the
mean derived from the measurements is shown as the dotted line. For details of the flight
paths the reader is referred to Parrish et al. (2009).

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
