# Peer review of "The high-resolution version of TM5 for optimised satellite retrievals: Description and Validation"

_Geoscientific Model Development, 2016_

## Referee Comment (RC1) · Anonymous Referee #1 · 25 Aug 2016

**Review of**

**The high-resolution version of TM5-MP for optimised satellite retrievals: Description and Validation**

August 25, 2016

**Introduction**

I think the work described in the manuscript is interesting. The manuscript clearly states the larger context and final aim of the work, and motivates reasonably well the changes made to the model. The work is reasonably focused, and in general rather well explained. The flow of the manuscript makes it motivating to read it.

However, there are also some major points of critic. The analysis of the results and explanation of the results is not thorough enough. Further, the writing of the manuscript is of poor quality. In addition, some useful information which would have been useful is forgotten.

**General**

The abstract is not attractive, as it is too much a listing of separate observations.

There is too much similarity between the abstract and the conclusions. The conclusion section should benefit from including an outlook paragraph (which, e.g., should not be in the abstract). Now that a study has been done on modifying the chemistry and the resolution, and improvements in the model performance are moderate, what could be the next points of focus for further development of TM5. What would be the way forward to further improve CTMs? Another aspect which could be discussed in the conclusion : as the differences between 1x1 and 2x3 are moderate, is it an option to still use the 3x2 version profiles as a priori for retrievals?

I have the impression that the manuscript is too strong in its argumentation that emissions are an important reason for discrepancies with observations. I would suggest that the authors make this claim more solid, e.g., by doing sensitivity experiments.

This manuscript is presenting a new version of TM5, and is mainly focusing on the new aspects. For most of the other information and comparisons, it refers to earlier publications. However, as the current manuscript is aimed to be a comprehensive description of the new TM5 version, it should also contain some relevant basic information. Essential information which is missing in such a model description is :

- What bout the number of levels in the model?

- What about the horizontal resolution towards the pole : is a reduced grid used?

- What about the number of species? Number of transported tracers? I assume that several new tracers have been added, which were not used in earlier versions.

- What about the turbulent diffusion in the model? What about the dry deposition scheme?

- Some lumped species which are mentioned in the text, should be shortly described : ORGNTR, ...

- What about the tracer transport scheme? Is it mass conservative?

The analysis of the differences between the two resolutions should be improved. What about the difference in turbulent vertical transport between both model simulations? What about dry deposition parameterisations affected by the different resolutions? What about lightning $NO_x$ parameterisations? In the current manuscript, there is a lot of focus on convection, whereas other parameterizations might also play a role.

It is also not made clear how large the differences are in the meteorological fields seen by the different resolutions. E.g., is the total precipitation equal in 3x2 and 1x1? Are the convective mass fluxes, when globally averaged, equal in both versions? Is the cloud cover equal when globally averaged? And the albedo? If not, it would be informative to quantify that.

Similarly, are the $^{222}$Rn and other emission totals equal?

As one of the aims of the model is to use it for generating instantaneous a-priori profiles and columns, it is not sufficient in this study to look at biases only. One should also look at the high frequency behaviour and thus, e.g., at correlations. E.g., Table 9 gives seasonal biases, but I think it is necessary to also show correlations. In addition, as the satellite retrievals will be used globally, it is not sufficient to quantify the difference between the 3x2 and 1x1 versions only for the observation locations of the manuscript. As the (tropospheric) columns of $NO_2$, $SO_2$ and $CH_2O$ from TM5 will be important for the retrieval, one could, e.g., estimate how well the 3x2 and 1x1 distributions are correlated spatially. This could also allow to better quantify whether using 3x2 in stead of 1x1 still makes sense.

The figures in the supplementary material are of poor quality.

It is not always clear from the text whether one is comparing resolutions, or models versus observations. Mentioning it always explicitly will make the text too heavy. However, the authors should be careful that their manuscript is not confusing.

The writing of the manuscript is of poor quality. This manifests itself in different ways :

- using abbreviations without or before they have been defined

- referring to wrong figures

- incorrect numbering of the sections

- using capital letters for words which do not need it (Forest, Tundra, ...)

- poor quality of the text in figure and table captions, the captions should also be much more homogeneous,

- starting to use a symbol for concentrations from page 12 onwards (e.g., [NO] and [$NO_2$])

- using expressions like :between 800-900 hPa ...

- spelling of identical words in different ways.

Finally, it is not clear what selection criterium is used for putting some figures and tables in the main document and others in the supplementary material? If a reader decides not to read the supplementary material, he should at least have an idea of what he will miss.

**Comments**

**page 2, line 55** The text mentions current resolutions of 2–4° in latitude and 2–6° in longitude. There are however currently models with higher resolutions, see, e.g., Yu et al. [2013.]

**page 3, line 113** which replaces the parameterization of Tiedtke (1989). Be clearer about what sub-grid scale parameterizations are still calculated in TM5. E.g., are turbulent diffusion coefficients calculated, or have they been archived too?

**page 3, line 117-119** Concerning large scale transport, one mentions the CFL criterium. In addition, maybe it is interesting for the reader to mention which transport scheme is used.

**page 6, line 231** The Schery et al. (2004) reference for the $^{222}$Rn emissions is difficult to find, as it is part of a book. It would therefore be useful to describe shortly some aspects of the $^{222}$Rn emission map: is it only from continents, is there a latitudinal gradient, are there emissions at high latitudes, is it very patchy or rather homogeneous? Is it dependent on soil moisture or precipitation?

**page 8, line 290-293** Differences are attributed to the resolution of the emission data set, and the convection. It is not clear why the temporal resolution of the emissions should play a large role. Isn't it mainly the horizontal resolution which plays a role in explaining the difference between 3x2 and 1x1? In addition is written earlier in the text (page 8, line 283) that $^{222}$Rn is emitted at a steady rate.

**page 10, line 368-369** Is this relevant as only 90°S-30°S, 30°S-30°N, and 30°N-90°N are shown?

**page 10, line 372-373** The abbreviations O3S (tagged $O_3$ tracer which undergoes only ...) and BO3S (stratospheric burden of $O_3$) are confusing. If O3S is a tracer as defined above, it would be logical that BO3S would be just the total burden of that tracer, whereas here that is not the case.

**page 11, line 414-417** Differences are attributed to only 4 causes. Might there not be an impact of the resolution on the dry deposition, the turbulent mixing, the large scale transport, or the mass conservation of the transport?

**page 12, line 428** shows that differences are small : across resolutions?

**page 11-12, line 436-438** this does not seem to hold on a regional scale.

**page 13, l 501** What is ORGNTR?

**page 15, line 581** splitting the atmosphere in 3 regions (NH extra-tropics, tropics, SH extra-tropics) is much rougher than "zonally integrated"

**page 16, line 603** Fig. S13 : is this the correct figure to be referred to?

**page 16, line 616-617** Isn't it the increase in spatial resolution which helps?

**page 25, line 744-745** Those with differences <5 % are considered to exhibit no discernible change in the bias. An interpretation should not be written in the caption of a figure.

**page 31-32, Figs. 4-5** Are the observations also just the 13:00 values?

**page 35, line 823** during September 2006 : but October 2006 is apparently also shown.

**SM** Add page numbers in the supplementary material.

**Questions**

**page 1, line 15-17** differences ... differences

**page 1, line 15-17** increases/decreases : it is not clear which resolution is the reference.

**page 1, line 18** "strength" of convective activity is rather vague. Is CAPE meant, updraft velocity, updraft mass flux?

**page 1, line 19** NH is not yet defined. What is meant by "NH (tropics)"?

**page 1, line 20-21** from simulations at 1x1 horizontal resolution. Isn't it also done for 3x2?

**page 1, line 31-32** not clear whether for both resolutions.

**page 1, line 34** shouldn't 20 and 35 sum up to 100? At this stage, the reader is not yet aware of the fact that changes of less then 5 % are not accounted for.

**page 1, line 35** in TM5-PP : only the high resolution version.

**page 8, line 301-303** are the globally averaged $^{222}$Rn emissions equal in 3x2 and 1x1? Are the globally average convective mass fluxes equal in 3x2 and 1x1?

**page 8, line 304-306** I would expect that, if archived mass fluxes are used, the global total mass flux is equal, independent of the resolution (1x1 or 3x2). Secondly, if mass fluxes were stronger in 1x1, I would expect for 1x1 (compared to 3x2) lower $^{222}$Rn concentrations at the surface and higher concentrations between 900 and 700 hPa. But for DJF in Paris and London one sees the inverse.

**page 9, line 341** and around Iceland for JJA. This difference is hard to distinguish.

**page 9, line 344-346** Here no averaging is performed towards an identical hor ... : does it mean that the value of a 1x1 grid box is compared with the value of a 3x2 grid box? Is there a spatial interpolation between 3x2 ... grid points, and 1x1 grid points?

**page 10, line 390-391** "At 1x1 the largest increase in STE occurs in the SH during JJA." : This cannot be seen from the numbers in the tables. Isn't the aim explaining the 7 % reduction?

**page 10, line 391-393** "Comparing the 0.2 contour ...". I have the impression, looking at the upper panels in Fig. 2, that the change in the SH is as large as in the NH. One sees a lowered contour line between 50°N-90°N and 50°S-90°S, but a lifted contour line between 30°N-50°N and 30°S-50°S, leading to a more tilted contour line representing stronger horizontal gradients (as if horizontal gradients can be conserved better in the 1x1 simulation).

**page 13, line 482-483** In the lower troposphere (<900 hPa) : as it is written here, one interpretes it as if (<900 hPa) is the definition of the lower troposphere. It is however meant to be an extra condition.

**page 13, line 490-493** And what about October?

**page 13, line 504** to quantify the effect on higher spatial resolution.

**page 13, line 515** decrease marginally by 2-3% : looking at Table S1, shouldn't it be 1-3%?

**page 16, line 619** aggregated on a weekly basis does not matter if one looks at seasonal biases; it would have played a role if one also shows correlations.

**page 24, Table 4** Whereas most terms in this table are in units of $Tg\,O_3\,yr^{-1}$, it is unclear in what units BO3 and Strat. BO3 are. If these are burdens, one would expect Strat. BO3 to be a larger fraction of BO3, than the values shown here (e.g., on the global scale 80 and 378).

**page 24, Table 4** It is not clear where one can find "The fraction of the tropospheric burden originating from the stratosphere is also given." Does one mean Strat BO3? Are these absolute values Tg, or is it %?

**page 24, Table 4** Why no % for the NH/SH/tropical STE changes? As these values are not given, the sentence on page 10, line 379 is rather unclear : "The increase in STE in the SH, with an associated decrease in the NH ..."

**Inconsistencies**

Below a list can be found of inconsistent use of abbreviations, capital letters, etc. Please make the manuscript more self-consistent.

- Sect. and Sect

- Correct the numbering of the sections and subsections.

- Fig. versus Figure. One should use Figure at the beginning of a sentence, but Fig. within a sentence.

- Free Troposphere : should just be written "free troposphere"

- Marine Boundary Layer should be just written "marine boundary layer".

- Chemistry Transport Models (CTM) versus vertical column densities (VCD).

- The naming of the campaigns should be coherent throughout the manuscript: INTEX-B versus INTEX B, Texas-AQS versus TexAQS II.

- earth-orbiting versus earth orbiting.

- TROPOMI versus tropOMI.

- supplementary material versus Supplementary Material.

- TM5-chem-v3.0 versus TM5 v3.0.

- grid cell versus grid-cell.

- gas phase versus gas-phase.

- BL : definition of boundary layer given much later than first three appearances.

- LT : used but never defined.

- SH, NH : used but never defined.

- BO3 : used but not defined.

- Actinic Fluxes versus actinic fluxes.

- um versus $\mu$m.

- [$SO_2$] (page 27, line 759) versus NO surface concentrations (page 31, line 793).

- 10 Tg S/year (page 6, line 211) versus 49 Tg N yr$^{-1}$.

- The Netherlands versus the Netherlands.

- 500 hPa versus 500hPa.

- $J$ values versus $J_{O3}$, $J_{NO2}$.

- Tg $O_3$ versus Tg$O_3$.

- Strat. nudging versus Strat BO3 versus Trop.Chem.Prod (Table 4).

- (i.e.) versus i.e.

- TES versus MLS : for TES the full name is given, whereas not for MLS.

- methodology outline versus methodology outlined.

- Stratosphere versus stratosphere.

- monthly-mean versus monthly mean.

- recycling versus re-cycling

- TR : used but never defined.

- UT : used but not defined.

- i.e. is sometimes used where e.g. should be used.

- overestimate versus over estimate.

- underestimate versus under-estimate.

- high bias versus high-bias.

- NOy versus $NO_y$.

- Strat. Nudge versus Strat. Nudging.

- N reservoir versus N-reservoir.

- 1° x 1° versus 1° x1°.

- cloud Surface Area Data : no capital letters needed.

- UTLS : not defined.

- ug m$^{-3}$ versus $\mu$g m$^{-3}$.

- monthly mean versus monthly-mean.

- down-welling versus down welling.

- underestimation versus under-estimation.

- both * and x in mathematical expressions.

- style for labeling panels in figures : sometimes is the position of the panel (left/right/top/bottom) given before mentioning the element it describes (e.g., in Figs. S5 and S7a), and sometimes it is given after the element it describes (most other figures).

- tropical cities versus Tropical cities.

- (-) versus ().

- (1x1 - 3x2)/3x2 (Fig. 1) versus 1x1/3x2 (most other figures).

- chose vs choose.

- Marine Boundary Layer does not need capital letters.

- Di-Methyl Sulphide : no capital letters needed.

**Inappropriate or unclear language**

Below are examples of the poor language of the manuscript. Sometimes suggestions are given for improvement, otherwise it is up to the authors to find a better expression.

**page 1, line 24** by between 5-10%.

**page 1, line 39-40** vertical column densities : is column-integrated values meant?

**page 2, line 50-53** from earth-orbiting satellites including ... : The use of "including" gives the impression that a list of satellites will follow, whereas actually a list of instruments follows.

**page 2, line 56** of hundreds of kilometers in area.

**page 2, line 57** sampling is not the same as resolution.

**page 2. line 59** constraints $\rightarrow$ limitations.

**page 2, line 60** NO is not yet defined.

**page 2, line 67** the information ... are.

**page 2, line 67** the coarsening procedure.

**page 2, line 71** VCD is never used later. So I would not define it.

**page 2, line 79-80** is for placing $\rightarrow$ is to place, or is placing.

**page 2, line 80** constraints in $\rightarrow$ constraints on.

**page 3, line 83** yields.

**page 3, line 83-84** whose spatial location is also smeared via the coarsening procedure : poor language.

**page 3, line 85** deriving biases $\rightarrow$ finding/estimating biases.

**page 3, line 89** add i.e. when you start the summing of all the modifications.

**page 3, line 93** Radon : without capital letter. Or use $^{222}$Rn.

**page 3, line 118** criteria : is this meant to be plural?

**page 4, line 121** place emphasis of → on.

**page 5, line 169-170** between 0.25-0.27$\mu$m.

**page 5, line 197** between 2003/2004 and 2001-2009.

**page 5, line 195** mean ratios between $CO/O_3$.

**page 6, line 208** Non-Methane Hydrocarbons (NMVOC). The abbreviation does not seem to correspond with the full expression.

**page 6, line 237** at both 3x2 and 1x1 : add resolution(s).

**page 7, line 238** diverse ... diverse.

**page 7, line 252** insures → ensures.

**page 7, line 253** $O_3$ , → $O_3$,

**page 7, line 259** around Alaska : all those flights start at the southern tip of Alaska and continue southwards, so around is not the appropriate wording.

**page 7, line 260** and bin → and we bin.

**page 7, line 265** ... we supplement the INTEX-B comparisons with those ...

**page 7, line 266** as part of the .... . → as part of ...., or as part of the ... campaign.

**page 8, line 285** deposition → dry deposition.

**page 8, line 288** averaged between 800-900 hPa.

**page 8, line 288-289** highlighting the spatial variability in convective upwelling near the top of the convective boundary layer : As the boundary layer depth is different for different locations, just sampling the 800-900 hPa altitude does not guarantee that you sample everywhere the top of the convective boundary layer. This sentence should be improved.

**page 8, l 294** with increases/decreases : not clear whether 1x1 is compared to 3x2, or the inverse.

**page 8, l 295** coasted regions → coastal regions.

**page 8, l 295** Madagaskar → Madagascar.

**page 8, l 306-307** range between  2-10%.

**page 9, line 318-319** potential differences [=differences] ... can be considerable compared to those [=absolute profile] ...

**page 9, line 322-323** For this comparison no averaging is employed [is this absolute?], where the selected grid cells are near the centre of each urban conurbation [or is this a condition?].

**page 9, line 323-324** residual(s) → ratio.

**page 9, line 329** changes ... has → have.

**page 10, line 372-373** zonal STE.

**page 10, line 375** the multi-model STE mean.

**page 10 ,line 376** This  7 % reduction ... : the use of "this" is strange because the reduction has not been mentioned before.

**page 10, line 380** the stratospheric BO3.

**page 10, line 383** "recent" study for 2011.

**page 10, line 382** in the horizontal and the vertical.

**page 11, line 425** vertical gradients → vertical profile.

**page 12, line 446** 13:00 local time close to → which is close to.

**page 12, line 441-442** improving the bias → reducing the bias.

**page 12, line 456** implying → concluding.

**page 12, line 459** accuracy → inaccuracy.

**page 12, line 463** For a 20% of sites ...

**page 12, line 463-465** should be improved.

**page 13, line 482** their ratio. These are ... → their ratio. It is ...

**page 13, line 486-487** during ... during.

**page 13, line 488** 1-sigma variability. One should explain better what is meant.

**page 13, line 501-503** Considering ... means ...

**page 14, line 521** advective mixing terms.

**page 14, line 524** this intermediate become → becomes.

**page 15, line 571** that is it → that it is.

**page 15, line 590** 3-5 Tg less $CH_2O$ $yr^{-1}$ → 3-5 Tg $CH_2O$ $yr^{-1}$ less.

**page 16, line 607** at number of EMEP sites.

**page 16, line 607-608** Forest, Rural, ... → forest, rural, ...

**page 16, line 609-610** Energy Sector → energy sector.

**page 16, line 609-611** strange sentence.

**page 16, line 610-611** varying from ... and ... → 1) varying between ... and..., or 2) varying from ... to ...

**page 16, line 618** overview of the changes : that would be correct if the difference was shown in Table 8. However, now just the values are shown in that table.

**page 16, line 631** being described better as for that shown for $NO_2$ in Fig.6.

**page 17, line 651-652** we show that differences exist at higher resolution.

**page 17, line 653** location orography.

**page 17, line 673-674** in only a of the order of few percent.

**page 18, line 680** For $SO_2$ comparison with surface observations in Europe show → shows.

**page 18, line 681** at 20% of sites.

**page 18, line 683** associated with either precursor or direct emission terms : shouldn't there be also the word underestimation in the last part of the sentence?

**page 19, line 697** ... applied for $NO_x$ radical-radical reactions and nitrogen reservoirs.

**page 19, line 699** is taken → are taken.

**page 21, Table 21** [E] is not defined.

**page 21, Table 21** $IC_3H_7O_2$ → $C_3H_7O_2$ or i-$C_3H_7O_2$.

**page 21, line 705 and 707** Branching ratio and Rate have a capital letter, whereas assumed has not.

**page 24, line 722-723** The definition ... and the ... are defined.

**page 24, line 724-725** nudging (=constraining) to constraints : one nudges to values.

**page 24, line 725-726** The contribution ... are provided.

**page 24, line 732** The RO2 term is ... $\rightarrow$ The NO + RO2 term ...

**page 25, line 742-743** The seasonal mean absolute biases of weekly [NO] ($\mu$g m$^{-3}$) composed from daily measurements at 13:00 for DJF and JJA. This sentence is hard to grasp, as there are four different references to time.

**page 25, line 743** (measurements-model).

**page 26, line 750** except.

**page 27, line 759-760** taking the difference between measurements-model values.

**page 28, line 766** between 800-900 hPa.

**page 28, line 766-768** (right) is mentioned, but not (left).

**page 33, line 814-817** (left) and (middle) are mentioned, but not (right).

**page 37, line 846** as part of the INTEX B $\rightarrow$ 1) as part of INTEX B, or 2) as part of the INTEX B campaign.

**page 38, line 857-859** Is "weekly" missing?

**page 39, line 866** as part of the Texas-AQS II.

**page 39, line 867** for each of the days?

**page 39, line 867-878** details ... details.

**SM, Fig. S1** residual $\rightarrow$ ratio.

**SM, Fig. S1** differences in ratio $\rightarrow$ ratios between.

**SM, Fig. S1** between 1x1/3x2.

**SM, Fig. S1** for January and July during 2006 $\rightarrow$ in 2006.

**SM, Fig. S1** red-line $\rightarrow$ red line.

**SM, Fig. S3** 1x1/1x1 (Tiedtke) (poor description).

**SM, Fig. S4** Monthly mean comparisons of $J_{O3}$ ... $\rightarrow$ Comparison of monthly mean $J_{O3}$ ...

**SM, Fig. S4** type of scenario : does this refer to "High Arctic", "Tundra", "Industrial", ...? I think scenario is not the correct word.

**SM, Fig. S6a** Residual $\rightarrow$ Ratio.

**SM, Figs. S7a, S7b, and S8** Why expressing "Comparisons are shown for volume mixing ratios." in Figs. S7a and S7b, and not in Fig. S8? Why not just mentioning the units in the first sentence after $O_3$?

**SM, Figs. S8 and S9** Is there a difference in interpretation between "The dotted line represents the 1-sigma variability associated with the measurements." and "The 1-sigma deviation from the measurements is shown as the dotted line for each of the days." If not, I suggest to homogenize the captions.

**SM, Fig. S11** except for.

**SM, Fig. S12** The units for the panels in the left column should be mentioned.

**SM, Fig. S14** details ... details.

**SM, Fig. S15** also October is shown.

**Additional corrections**

Below can be found a list of additional errors which should be corrected.

**page 2, line 47** (CTM) → (CTMs).

**page 2, line 49** earth-orbiting → Earth-orbiting.

**page 2, line 53** (Valks et al., 2011)) → (Valks et al., 2011).

**page 2, line 50** (TES, Worden et al., 2007) → (TES, Worden et al., 2007),

**page 2, line 68-69** an associated uncertainty of 2 → an associated uncertainty of a factor of 2.

**page 3, line 103** regional → region.

**page 3, line 105-106** massivelyntersta .

**page 4, line 123 and 125** is → are.

**page 4, line 145** Details → For details

**page 4, line 149** land and ocean ... 40 and 900, respectively. I presume 40 and 900 should be inversed.

**page 6, line 221** 6Tg/N yr$^{-1}$ → 6Tg N yr$^{-1}$.

**page 6, line 225** SO$_2$ (117 Tg S yr$^{-1}$) : this seems a lot. Could it be meant SO$_2$ (117 Tg SO$_2$ yr$^{-1}$)?

**page 6, line 234-235** (VOC) → (VOCs).

**page 9, line 323** that the significant differences exist.

**page 9, line 342** Figure S3 refers to $^{222}$Rn, whereas here $J$-values are discussed

**page 10, line 362** the also the efficiency.

**page 10, line 396** from the Stratospheric.

**page 11, line 379** (Verstraeten et al. (2015)) → (Verstraeten et al., 2015).

**page 12, line 444** Figures 4 and 5 shows → show.

**page 12, line 453** due → due to.

**page 12, line 475** are the order of.

**page 13, line 487-488** in NO and NO$_2$, mixing ratios.

**page 15, line 564** Goncalves et al, 2012 → Goncalves et al., 2012.

**page 15, line 596-597** or that the chemical production term is too.

**page 16, line 626** significantly → significant

**page 16, line 628** Figure S14 → Figure S16.

**page 13, line 637** Figure 13 → Figure 12.

**page 16, line 637** September 2006 is mentioned in the text, whereas also October is shown in the figures.

**page 21, line 705** 1/(1+498.*exp(-1160/T) → 1/(1+498.*exp(-1160/T)).

**page 34, line 823 and page 13, line 487** October is also shown.

**page 34, line 822-825** (left), (middle) → (top), (middle), and (bottom).

**SM, Table S1** Tg/N.

**SM, Tables S1 and S2** chemical troposphere → chemical tropopause.

---

## Referee Comment (RC2) · Anonymous Referee #2 · 28 Sep 2016

*Review of "The high-resolution version of TM5-MP for optimised satellite retrievals: Description and Validation"*

The work described in this paper is certainly interesting and the motivation for the work is clearly explained. The general structure and flow is good. However, in the current state, it is difficult to assess the reproducibility and scientific quality of this work for two main reasons: the experimental setup and differences between the two model resolutions are not clearly explained; the analysis and presentation of the results is not well structured and leaves a number of unanswered questions.

**Major Comments**

1) A much more rigorous description of the two model setups and their differences is required. There is currently no mention of the vertical resolution although this is later reported as being important for e.g. STE fluxes. In its current format the model description in Section 2 is lacking information and is written in a way that makes it unclear which modifications are applied to the new TM5 model version and which to the higher resolution.

2) A paragraph should be added in Section 2 to clearly describe all details for the two model integrations used for this work. In particular, the authors should specify:
- start date of integrations and run length
- chemical initial conditions and spin up periods
- details of the analysis used (horizontal, vertical and temporal resolution).
Are these the same for both model resolutions or are they different? Are the simulated model years the same used for the emissions and observational datasets?

3) The use of EMEP observations in it's current state is confusing.
In line 242-243 the authors state that sites in "Norway, Finland, The Netherlands, Belgium, Poland, Germany, Spain, Italy and Portugal" are used for comparison. Why just use EMEP stations from the above and not the ones in other countries? Or is this a mistake (see later)?
The same uncertainty regarding only selected sites being used and lack of explanation on why they are used is present throughout the manuscript and affects the interpretation of the results.

In Fig 3, EMEP sites from " Finland, The Netherlands, Belgium, Poland, Slovakia and Italy" are used, aggregated by nation. The authors should explain why just these six countries? Why aggregate the sites? Poland and Slovakia should be mentioned in the list of sites in Section 2!

In Fig 4, four EMEP sites are selected for comparison. Why these 4 sites? Again the sites in the Czech Republic and Great Britain used in this figure are not listed in Section 2.

In Fig 5, two sites are selected for comparison and again no explanation as to why those specific sites are used.

Similarly, Table 6 and 7 use yet two different subsets of EMEP stations for comparison without explaining the reason for their choice.

In Fig 8, four selected EMEP sites are shown (from Norway, Germany, Austria and Slovakia). Same issues as above.

If the purpose of the comparison with EMEP is to evaluate the model performance in the new configuration, as well as addressing the differences in model resolution, the current analysis is not convincing. Comparison of model data with tropospheric ozone column from satellite would help better evaluate model performance on the global scale. This could also lead to better evaluate the model ozone profiles which currently show significant discrepancies with MOZAIC data. Further comparison with EMEP surface sites (and other campaign data) would then add to the analysis, so far as the comparison is done across all suitable sites and a clear explanation is given if only a subset of sites is selected.

4) In Section 5 the authors provide an analysis of budget terms for tropospheric ozone and compare these at the two different model resolutions. They state: "the chemical tropopause calculated for 3x2 is applied for the analysis of 1x1 budget terms to ensure that a valid comparison is performed". However, if convection and convective transport is significantly different in the two model resolutions (as the authors suggest) the position of the chemical tropopause at 1x1 should be at a higher altitude in the tropics compared to 3x2. Using the 3x2 chemical tropopause to analyse 1x1 budget terms is in my view inconsistent and the reduction in STE term with increasing resolution is likely to change if the 1x1 chemical tropopause is used.

**Minor Comments**

line 16: change 'coastal' to 'coastal regions'

line 66: 'and the extent of mixing by convective upwelling (i.e. land type)'; it is not clear why
    land type is mentioned here

line 75: change 'where' to 'in which'

line 103: change 'regional' to 'regional domain'

line 105: 'massivelyntersta' ? Correct.

line 218-219: Explain which method is used for coarsening of emissions (area averaged?
    Linear interpolation?) and comment on its suitability

line 220-222: not clear from the text how the lightning Nox parameterisation works. Does
    it use convective precipitation (line 220) or convective flux (line 221)? And if it uses
    convective flux (from ERA?) why does it need rescaling?

line 235: Fisher et al., 2015 is not present in the References section

line 292-293: 'more accurate temporal distribution of regional  222Rn emissions at 1x1'; is
    the timing of the emission different at 1x1 compared to 3x2? If so it should be
    clearly stated and it should be explained why this would be desirable or necessary.

line 295: 'coasted' to 'coastal'

line 308: '(i.e.) orography and land type' to '(i.e. orography and land type)'

line 323: 'that the significant differences exist' remove 'the'

line 362: 'the also the efficiency' remove 'the'

line 409: 'the surface deposition flux to e.g. should...' correct

line 514: 'related to lower [OH] (i.e.) chemical production'  remove '(i.e.)'

line 596-597: 'that the chemical production term is too.' ? Not clear, rephrase

line 665: 'across diverse global regions'? Either 'globally' or 'across various regions'

---

## Author Comment (AC1) · 14 Oct 2016

**Response to anonymous referee #2:**

**We thank the referee for their review of our manuscript and provide responses to the questions and suggestions below:**

A much more rigorous description of the two model setups and their differences is required. There is currently no mention of the vertical resolution although this is later reported as being important for e.g. STE fluxes. In its current format the model description in Section 2 is lacking information and is written in a way that makes it unclear which modifications are applied to the new TM5 model version and which to the higher resolution.

We now add more details regarding the vertical resolution employed which is identical to that described in Huijnen et al. (2010). In the interests of brevity we did not include these specific details but will address this point by adding the following sentence:

*Although TM5-MP can adopt all 60 vertical levels provided by the ECMWF ERA-Interim analysis, we employ 34 vertical levels for this study with higher resolution in the troposphere and the upper-troposphere-lower stratosphere (UTLS).*

An identical model version is compared in our study, with the only change between simulations being the horizontal resolution that is employed. This allows us to attribute the changes shown to the use of increased resolution. We now add the following sentence to the end of the first paragraph in Sect. 2.

*The following model description pertains to both 3° x 2° and 1° x 1° simulations discussed in this manuscript.*

A paragraph should be added in Section 2 to clearly describe all details for the two model integrations used for this work. In particular, the authors should specify:

- start date of integrations and run length

- chemical initial conditions and spin up periods

- details of the analysis used (horizontal, vertical and temporal resolution).

Are these the same for both model resolutions or are they different? Are the simulated model years the same used for the emissions and observational datasets?

We now provide extra details regarding the simulations as requested by the referee. Again the only model parameter changed between the simulations is the horizontal resolution, increasing from 3° x 2° to 1° x 1°. We only use 2006 observational data and clarify this in Sect. 2.3:

*We choose a range of ground-based and airborne measurements taken at diverse locations during the year 2006 representing different chemical regimes.*

The use of EMEP observations in it's current state is confusing. In line 242-243 the authors state that sites in "Norway, Finland, The Netherlands, Belgium, Poland, Germany, Spain, Italy and Portugal" are used for comparison. Why just use EMEP stations from the above and not the ones in other countries? Or is this a mistake (see later)?

This was an oversight and has now been corrected in the text, thus:

*… where we exploit measurements taken at various background sites in Norway, Finland, The Netherlands, Belgium, Poland, the Czech republic, Germany, Great Britain, Spain, Slovakia, Italy and Portugal. The number of sites used for comparisons of trace species other than $O_3$ is smaller due to data availability.*

In Fig 3, EMEP sites from " Finland, The Netherlands, Belgium, Poland, Slovakia and Italy" are used, aggregated by nation. The authors should explain why just these six countries? Why aggregate the sites? Poland and Slovakia should be mentioned in the list of sites in Section 2!

For the sake of brevity, we choose not to show individual stations but rather aggregates as has been presented in other studies (e.g. Williams et al., 2013), since we feel a station by station decomposition is not the ideal presentational form. The aggregates shown cover a significant range of latitudes throughout Europe and we wish to show comparisons for the entire European domain to provide confidence in the model performance. We now change the text to:

*Figure 3 shows comparisons of simulated and observed mass mixing ratios of surface $O_3$ at EMEP sites across Europe ([www.emep.int](www.emep.int); Aas et al. 2001), with countries chosen so to cover a range of latitudes.*

In Fig 4, four EMEP sites are selected for comparison. Why these 4 sites? Again the sites in the Czech Republic and Great Britain used in this figure are not listed in Section 2.

These sites were chosen to show the diverse changes that can occur for different locations. Again the range in latitudes and longitudes shown for the European domain is broad, where only a limited number of EMEP stations measure NO and $NO_2$ therefore identical composites as those for $O_3$ cannot be presented. The biases for all stations are shown in Table 6.

In Fig 5, two sites are selected for comparison and again no explanation as to why those specific sites are used.

For brevity we choose to show a high and low $NO_x$ rather than an extended set. Table 7 does provide the seasonal biases across all stations if the reader is curious as to the behavior at other sites.

Similarly, Table 6 and 7 use yet two different subsets of EMEP stations for comparison without explaining the reason for their choice.

Tables 6 and 7 present seasonal biases for **all** EMEP stations which measure NO and $NO_2$. Therefore the selection is dictated by data availability rather than by the authors. We modify the table heading accordingly:  *Values are shown for both the 3° x 2° and 1° x 1° simulations for all stations with available data.*

In Fig 8, four selected EMEP sites are shown (from Norway, Germany, Austria and Slovakia). Same issues as above.

See explanations given for previous referee comments related to EMEP sites above. The number of stations measuring $HNO_3$ is a small subset of the total number of stations  in the EMEP network.

If the purpose of the comparison with EMEP is to evaluate the model performance in the new configuration, as well as addressing the differences in model resolution, the current analysis is not convincing. Comparison of model data with tropospheric ozone column from satellite would help better evaluate model performance on the global scale. This could also lead to better evaluate the model ozone profiles which currently show significant discrepancies with MOZAIC data. Further comparison with EMEP surface sites (and other campaign data) would then add to the analysis, so far as the comparison is done across all suitable sites and a clear explanation is given if only a subset of sites is selected.

Due to the lack of stratospheric chemistry and microphysics in TM5MP we actually employ tropospheric ozone columns from the Multi-Sensor Re-analysis (van der A et al., 2010) for constraining the overhead ozone column (i.e.) the total column is nudged towards the observed value. Details of this method are given in Sect 2.1. This means an independent comparison against satellite data is not feasible as a large fraction of $O_3$ exists above the tropopause, which is the threshold where the nudging constraint is applied. The tropospheric component of any total column value is notoriously difficult to retrieve (e.g. de Laat et al., ACP, 2009) with a high uncertainty in the value, further compounded by sampling (totally cloudy skies). EMEP comparisons are regularly used to assess the accuracy of air quality models and have high temporal coverage throughout the day for all seasons therefore act as an excellent dataset for evaluating near surface $O_3$ as long as interpolation is done well. We also provide comparisons against two independent campaigns which cover multiple days and locations, allowing an assessment of the vertical profile in the tropopause which the referee fails to mention. We feel that our conclusions are robust as to the effect of higher horizontal resolution of tropospheric $O_3$, with similar behavior seen across independent comparisons.

In Section 5 the authors provide an analysis of budget terms for tropospheric ozone and compare these at the two different model resolutions. They state: "the chemical tropopause calculated for 3x2 is applied for the analysis of 1x1 budget terms to ensure that a valid comparison is performed". However, if convection and convective transport is significantly different in the two model resolutions (as the authors suggest) the position of the chemical tropopause at 1x1 should be at a higher altitude in the tropics compared to 3x2. Using the 3x2 chemical tropopause to analyse 1x1 budget terms is in my view inconsistent and the reduction in STE term with increasing resolution is likely to change If the 1x1 chemical tropopause is used.

Initially we did not impose the 3° x 2° tropopause definition onto the 1° x 1° budget analysis using the 150ppb gradient to diagnose the chemical tropopause for both simulations. The resulting burdens are entirely different as a different total mass of air is compared, making the STE component unrealistically large and the analysis incompatible with the profile comparisons shown throughout the manuscript. Many CTM studies adopt a climatological tropopause such as that provided by e.g. Lawrence et al., ACP, 2001 for their analysis in order to address this total mass issue.

---

## Author Comment (AC2) · 25 Oct 2016

Response to anonymous referee #1:

We thank the referee for their comprehensive and detailed review of our manuscript and provide responses to the questions and suggestions below:

Major comments:

There is too much similarity between the abstract and the conclusions. The conclusion section should benefit from including an outlook paragraph (which, e.g., should not be in the abstract). Now that a study has been done on modifying the chemistry and the resolution, and improvements in the model performance are moderate, what could be the next points of focus for further development of TM5. What would be the way forward to further improve CTMs? Another aspect which could be discussed in the conclusion : as the differences between 1x1 and 2x3 are moderate, is it an option to still use the 3x2 version profiles as a priori for retrievals?

We have now limited our abstract to focus on chemical trace species that will be retrieved by TM5-MP. We modify the abstract accordingly:

*We provide a comprehensive description of the high-resolution version of the TM5-MP global Chemistry-Transport Model, which is to be employed for deriving highly resolved vertical profiles of nitrogen dioxide ($NO_2$), formaldehyde ($CH_2O$), and sulphur dioxide ($SO_2$) for use in satellite retrievals from platforms such as the Ozone Monitoring Instrument (OMI) and the Sentinel-5 Precursor, the TROPOspheric Monitoring Instrument (tropOMI). Comparing simulations conducted at horizontal resolutions of 3º x 2º and 1º x 1º reveals differences of $\pm 20\%$ exist in the global seasonal distribution of $^{222}Rn$, being larger near specific coastal locations and tropical oceans. For tropospheric ozone ($O_3$), analysis of the chemical budget terms shows that the impact on globally integrated photolysis rates is rather low, in spite of the higher spatial variability of meteorological data fields from ERA-Interim at $1° x 1°$. Surface concentrations of $O_3$ in high-$NO_x$ regions decrease between 5-10% at $1° x 1°$ due to a reduction in $NO_x$ recycling terms and an increase in the associated titration term of $O_3$ by NO. At $1° x 1°$, the net global stratosphere-troposphere exchange of $O_3$ decreases by ~7%, with an associated shift in the hemispheric gradient. By comparing NO, $NO_2$, $HNO_3$ and PAN profiles against measurement composites, we show that TM5-MP captures the vertical distribution of $NO_x$ and long-lived $NO_x$ reservoirs at background locations, again with modest changes at $1° x 1°$. We show that surface mixing ratios in both NO and $NO_2$ are generally underestimated in both low and high $NO_x$ scenarios. For Europe, a negative bias exists for [NO] at the surface across the whole domain, with lower biases at $1° x 1°$ at only ~20% of sites. For $NO_2$, biases are more variable, with lower (higher) biases at $1° x 1°$ occurring at ~35% (~20%) of sites, with the remainder showing little change. For $CH_2O$, the impact of higher resolution on the chemical budget terms is rather modest, with changes less than 5%. The simulated vertical distribution of $CH_2O$ agrees reasonably well with measurements in pristine locations, although column-integrated values are generally underestimated relative to satellite measurements in polluted regions. For $SO_2$, the performance at $1° x 1°$ is principally governed by the quality of the emission inventory, with limited improvements in the site specific biases with most showing no significant improvement. For the vertical column, improvements near strong source regions occur which reduce the biases in the integrated column.*

Previous studies have quantified retrieval errors with respect to horizontal resolution (e.g. Boersma et al., 2007; Heckel et al., ACP, 2011) and considering the small footprint of the new tropOMI instrument, it seems disingenuous to use a 3° x 2° model grid to perform such retrievals considering the progress made in the instrument resolution.

I have the impression that the manuscript is too strong in its argumentation that emissions are an important reason for discrepancies with observations. I would suggest that the authors make this claim more solid, e.g., by doing sensitivity experiments.

Performing such sensitivity studies would then turn our manuscript into a scientific paper rather than a model description and validation paper, whereas the purpose of our submission to GMD is to provide a peer-reviewed benchmarking reference. It is envisaged that studies related to emission estimates and retrievals from tropOMI will occur once the satellite is launched (spring 2017). Other independent studies have placed discrepancies between models and measurements almost entirely on missing emission terms, therefore allowing inversion studies to be performed (e.g. Elburn et al., ACP, 2007; Kim et al., ACP, 2011; Manning et al., JGR, 2011). Moreover, the basis of emission trend studies from Earth-orbiting satellites relies on the missing component being almost entirely due to emission fluxes (e.g. Schneider et al., 2015). Given the large discrepancy between e.g. lower tropospheric $NO_x$ in Texas, an area subject to high Anthropogenic emissions, the first-order impact is also thought to be from under-estimates in emissions (e.g. de Gouw et al., Env. Sci. Tech., 2011).

The analysis of the differences between the two resolutions should be improved. What about the difference in turbulent vertical transport between both model simulations? What about dry deposition parameterisations affected by the different resolutions? What about lightning NOx parameterisations? In the current manuscript, there is a lot of focus on convection, whereas other parameterizations might also play a role.

We focus on the convective aspect as the source of the convective mass-fluxes has changed in this version of the model compared to previous versions, rather than the e.g. turbulent mixing scheme, which is identical. Resolution effects on turbulent mixing would require a separate study and, again, we consider this a model validation paper with a focus on retrievable trace gases, and therefore present the cumulative result of all resolution induced changes. Additional tuning was performed between simulations so that the lightning $NO_x$ is constrained to an annual global total of 6 Tg N $yr^{-1}$ throughout, as described in the text, thus:

*For lightning $NO_x$ we use the parameterization which uses convective precipitation fields (Meijer et al., 2001) and constrain the annual global emission term at ~6 Tg N $yr^{-1}$. This uses the convective flux values meaning that re-scaling of the nudging term was necessary in order to achieve similar total lightning $NO_x$ across simulations.*

This ensures that the $NO_x$ emission total is the same between runs allowing a valid comparison.

The vertical grid is identical between 3° x 2° and 1° x 1° simulations. For the dry deposition, although regional terms may exhibit larger differences, the small change in the $O_3$ deposition term in the Northern Hemisphere given in Table 4 implies this is not a dominating source of the modest differences found.

It is also not made clear how large the differences are in the meteorological fields seen by the different resolutions. E.g., is the total precipitation equal in 3x2 and 1x1? Are the convective mass fluxes, when globally averaged, equal in both versions? Is the cloud cover equal when globally averaged? And the albedo? If not, it would be informative to quantify that.

For details on the use of meteorological fields in TM5-MP the referee is pointed to Bregman et al., ACP, 2003 and Huijnen et al., GMD, 2010. The similarity in both the regional photolysis frequencies (where clouds dominate the total Optical Depth; Figure S4) and the wet and dry deposition fluxes shows that there are no significant changes in the global and zonal mean terms for such quantities. Transport will be better defined using higher resolution wind fields, but this is one of the benefits of increasing horizontal resolution evident in the March INTEX-B comparisons of e.g. $O_3$. We feel that a comparison of such meteorological fields would detract from the real focus of our paper, which is whether the integrated effect of the change in resolution alters the chemical composition of the troposphere significantly.

As one of the aims of the model is to use it for generating instantaneous a-priori profiles and columns, it is not sufficient in this study to look at biases only. One should also look at the high frequency behaviour and thus, e.g., at correlations. E.g., Table 9 gives seasonal biases, but I think it is necessary to also show correlations. In addition, as the satellite retrievals will be used globally, it is not sufficient to quantify the difference between the 3° x 2° and 1° x 1° versions only for the observation locations of the manuscript. As the (tropospheric) columns of $NO_2$, $SO_2$ and $CH_2O$ from TM5 will be important for the retrieval, one could, e.g., estimate how well the 3° x 2° and 1° x 1° distributions are correlated spatially. This could also allow to better quantify whether using 3° x 2° instead of 1° x 1° still makes sense.

As well as presenting the biases at EMEP surface sites in Europe, we also present comparisons of vertical profiles across a wide area from the INTEX-B and Texas-AQSII campaigns (Singh et al., ACP, 2009; Parrish et al., JGR, 2009). The locations chosen for validation are significantly restricted by data availability during 2006. However, the main findings are consistent across all selected regions, therefore we feel confident that as we have compared surface values and vertical profiles in both remote and urban scenarios (i.e. over different chemical regimes) the main biases in any a-priori fields have been sufficiently quantified.

At the request of the referee, we have examined the Pearson correlation co-efficients for the seasonal biases given between observations and instantaneous values at 13:00hrs in Tables 7 and 8. For $NO_2$, only a few sites exhibit significant correlations with $r > 0.65$ (i.e.) with many more exhibiting anti-correlations i.e. negative $r$ values, especially during DJF, or $r$ values between -0.3-0.3 indicating no meaningful correlation between model and measurements at all. There is typically a marked difference in $r$ between seasons at sites for both simulations, with JJA generally exhibiting higher correlations. Looking across sites reveals increasing resolution does not necessarily increase correlation though, with 1/3 of the sites exhibiting less correlation at 1° x 1°and ¼ being relatively unaffected. Comparing $r$ values at 1° x 1° using the Tiedke convective scheme shows that although there is some impact, there is not a consistent increase in correlation when using the ERA-interim archived mass-fluxes, with many sites exhibiting significant decreases. Therefore, similar to the conclusions regarding seasonal biases, the use of 1° x 1° does not lead to a systematic improvement

in correlation showing the constraints of using monthly mean estimates for emissions towards capturing variability.

We include the following text to summarise this:

*Analyzing the corresponding seasonal correlation co-efficients (not shown) shows in ~25% of the cases there is little seasonal correlation between the weekly [NO$_2$] in TM5-MP and the measurements regardless of resolution for both seasons (Pearson's r in the range -0.3-0.3). In ~30% of cases there is actually a degradation in r between resolutions, the changes somewhat reflect those seen in the seasonal biases i.e. simultaneous changes to both the meteorology and local emission fluxes do not necessarily improve the performance of the model. Comparing 1º x 1º values both with and without the Tiedtke convection scheme shows that for the most convective regions (e.g. south of 45ºN) increases in r generally occur during JJA when employing the ERA-interim mass-fluxes. Conversely for e.g. Finland the correlation becomes worse.*

Finally, it is not clear what selection criterium is used for putting some figures and tables in the main document and others in the supplementary material? If a reader decides not to read the supplementary material, he should at least have an idea of what he will miss.

The authors selected which Figures they find most revealing i.e. that show the most interesting findings typical of most manuscripts. We reference the Supplementary Material many times in the text of the manuscript, so assume that the reader has the opportunity to look at all Figures shown if he/she is interested in any particular trace gas.

Specific comments:

**page 1, line 15-17** differences ... differences

**page 1, line 15-17** increases/decreases : it is not clear which resolution is the reference.

We modify the sentence thus: *Differences of $\pm$20% exist in the global seasonal distribution of $^{222}$Rn between simulations conducted at 3° x 2° and 1° x 1°, being larger near specific coastal locations and tropical oceans.*

**page 1, line 18** "strength" of convective activity is rather vague. Is CAPE meant, updraft velocity, updraft mass flux?

The archived convective mass-fluxes and detrainment rates are the new meteorological fields employed in TM5-MP from the ECMWF meteorological dataset as described in Sect. 2.1 of the manuscript. We refer to the cumulative changes in convection determined using the $^{222}$Rn tracer, which come from a combination of parameters in the meteorological dataset, now summarized as the term "convective transport".

**page 1, line 19** NH is not yet defined. What is meant by "NH (tropics)"?

We remove this abbreviation from the abstract and change the text accordingly: *Analyzing vertical profiles of $^{222}$Rn above source regions, differences in the strength of the convective transport of between 2 and 10% (~10 and 20%) occur below 700hPa (200hPa) in the Northern Hemisphere around the tropics.*

**page 1, line 20-21** from simulations at 1x1 horizontal resolution. Isn't it also done for 3x2?

To determine any difference in J values requires the comparison of two different runs. We clarify this in the abstract thus: *For tropospheric ozone ($O_3$) analysis of the chemical budget terms between simulations shows that the impact on globally integrated photolysis rates is rather low, in spite of the higher spatial variability of meteorological data fields from ERA-Interim at 1°x 1°.*

**page 1, line 31-32** not clear whether for both resolutions.

We change the text accordingly: "*By comparing NO, $NO_2$, $HNO_3$ and PAN profiles from both simulations against a host of measurements … *"

**page 1, line 34** shouldn't 20 and 35 sum up to 100? At this stage, the reader is not yet aware of the fact that changes of less then 5% are not accounted for.

We imply that at 45% of the sites there is no significant change in the bias. We change the text accordingly: *For $NO_2$, biases are more variable, with lower (higher) biases at 1°x 1° occurring at ~35% (~20%) of sites, with the remainder showing little change.*

**page 1, line 35** in TM5-PP : only the high resolution version.

Figure 8 shows that there is a seasonal cycle in [$HNO_3$] for both simulations, where there is a strong correlation for between simulations.

**page 2, line 55** The text mentions current resolutions of 2–4∘ in latitude and 2–6∘ in longitude. There are however currently models with higher resolutions, see, e.g., Yu et al. [2013.]

We thank the referee for this information and update the text accordingly.

**page 3, line 113** which replaces the parameterization of Tiedtke (1989). Be clearer about what sub-grid scale parameterizations are still calculated in TM5. E.g., are turbulent diffusion coefficients calculated, or have they been archived too?

We now include the following text: "*The vertical diffusion in the free troposphere is calculated according to Louis (1979), and in the BL by the approach of Holtslag and Boville (1993). Diurnal variability in the BL height is determined using the parameterization of Vogelezang and Holtslag (1996).*"

**page 3, line 117-119** Concerning large scale transport, one mentions the CFL criterium. In addition, maybe it is interesting for the reader to mention which transport scheme is used.

We change the text accordingly: "*We use the first-order moments scheme with an iterative time-step to prevent too much mass being transported out of any particular grid-cell…*"

**page 6, line 231** The Schery et al. (2004) reference for the 222Rn emissions is difficult to find, as it is part of a book. It would therefore be useful to describe shortly some aspects of the 222Rn emission map: is it only from continents, is there a latitudinal gradient, are there emissions at high latitudes, is it very patchy or rather homogeneous? Is it dependent on soil moisture or precipitation?

The distribution of global $^{222}$Rn emissions is shown in Zhang et al, ACP, 2011. We now reference this publication for readers interested in the specifics of the $^{222}$Rn distribution. We feel that an in-depth discussion of the emission inventories used detracts from the main focus of our manuscript.

**page 8, line 290-293** Differences are attributed to the resolution of the emission data set, and the convection. It is not clear why the temporal resolution of the emissions should play a large role. Isn't it mainly the horizontal resolution which plays a role in explaining the difference between 3°x2° and 1°x1°? In addition is written earlier in the text (page 8, line 283) that 222Rn is emitted at a steady rate.

The emissions, which are typically provided at 0.5-1° resolution, are distributed onto the working model grid. Therefore more heterogeneity occurs on a higher resolution as urban and rural centers are differentiated more acutely. Emission at a steady rate means there is no variability in the monthly mean emission flux representing meteorological factor or diurnal variability.

**page 8, line 301-303** are the globally averaged 222Rn emissions equal in 3° x 2° and 1° x 1°? Are the globally average convective mass fluxes equal in 3° x 2° and 1° x 1°?

**page 8, line 304-306** I would expect that, if archived mass fluxes are used, the global total mass flux is equal, independent of the resolution (1° x 1° or 3° x 2°). Secondly, if mass fluxes were stronger in 1° x 1°, I would expect for 1° x 1° (compared to 3° x 2°) lower $^{222}$Rn concentrations at the surface and higher concentrations between 900 and 700 hPa. But for DJF in Paris and London one sees the inverse

All globally integrated emission fluxes of $^{222}$Rn are identical between simulations allowing a valid comparison of results, similar to the other emissions introduced into TM5-MP. The values at specific locations do change though due to the degree of coarsening of the 0.5° x 0.5° ECMWF data needed for the different resolutions (although the area-weighted total is equal to the original ECMWF data in both cases). For comprehensive details on the use of meteorological datasets in TM5 the referee is referred to Huijnen et al. (2010), which for the sake of brevity we do not include in our manuscript. As would be expected, the global mean of the convective mass fluxes calculated using 3° x 2° and 1° x 1° values can be slightly different due to potentially wider spread in the 1° x 1° values (more members of the data array), although the summed total will be equal. This holds for other tropospheric parameters such as temperature and surface albedo. Here we are more interested in regional differences. To remove the variability in emission fluxes above point locations averaging of the 1° x 1° profiles is necessary (where decomposition of the 3° x 2° profile at sub-grid scale is not possible), thus being able to differentiate the impact of the meteorology. Under instances of weak convective activity (DJF), our results show that indeed the coarsened 3 x 2 convective mass-flux can result in more uplift than at 1° x 1°, due to the variability in the averaged 1° x 1° values being high.

**page 9, line 341** and around Iceland for JJA. This difference is hard to distinguish.

We remove this from the text and will provide a higher resolution version of the diagram to improve clarity.

**page 9, line 344-346** Here no averaging is performed towards an identical hor ... : does it mean that the value of a 1° x 1° grid box is compared with the value of a 3° x 2° grid box? Is there a spatial interpolation between 3° x 2° grid points, and 1° x 1° grid points?

We use the geographical location of the cities to perform interpolation in both cases, as for all the profile comparisons for trace gases shown in the manuscript.

**page 10, line 368-369** Is this relevant as only 90°S-30°S, 30°S-30°N, and 30°N-90°N are shown?

Yes, because these three zones are comprised of cumulative sums from the 10° bands, therefore we inform the reader as to the resolution of the budget terms.

**page 10, line 372-373** The abbreviations O3S (tagged O3 tracer which undergoes only ...) and BO3S (stratospheric burden of $O_3$) are confusing. If O3S is a tracer as defined above, it would be logical that BO3S would be just the total burden of that tracer, whereas here that is not the case.

BO3S is the burden of O3S, thus : *"The stratospheric burden of $O_3$ ($BO_3S$) exhibits a strong hemispheric gradient … "*

**page 10, line 390-391** "At 1°x1° the largest increase in STE occurs in the SH during JJA." : This cannot be seen from the numbers in the tables. Isn't the aim explaining the 7% reduction?

Figure 2 is introduced at the start of this paragraph (line 387) and we refer to this when discussing the change in the latitudinal gradient in Stratospheric $O_3$ with respect to the downwelling.

**page 11, line 414-417** Differences are attributed to only 4 causes. Might there not be an impact of the resolution on the dry deposition, the turbulent mixing, the large scale transport, or the mass conservation of the transport?

The fact that the cumulative deposition velocities for e.g. $O_3$ (Table 3) are essentially the same between the 3° x 2° and 1° x 1° simulations shows that dry deposition effects are minimal. We implicitly examine the differences in the turbulent mixing with the $Rn^{222}$ comparisons, which hold for $O_3$ considering the tropospheric lifetime is typically > 20 days. We modify the text accordingly. The large scale transport does change as shown in the INTEXB comparison and we comment on it there. Without performing tagged $O_3$ experiments we cannot fully quantify changes in the long-range transport component.

**page 12, line 428** shows that differences are small : across resolutions?

Between the different simulations thus resolutions. We change the text thus:" … *shows that differences are small between simulations, and typically mimic those which occur at the surface."*

**page 11-12, line 436-438** this does not seem to hold on a regional scale.

All budget terms we show are for the global or a zonal domain. There is no 3D budget file output during a run due to computational constraints. We realise that providing analyzing results in this way will not provide exact changes in clean/polluted regions. However, the comparison of $O_3$ mixing ratios in Europe again EMEP measurements shows that differences between resolutions are small,

therefore changes are no so large as to lead to first-order reductions in resident [$O_3$] due to much higher [NO].

**page 13, l 501** What is ORGNTR?

This is the tracer name for lumped alkyl nitrates. We now include a definition at the end of the introduction along with $HNO_3$ and PAN.

**page 13, line 482-483** In the lower troposphere (<900 hPa) : as it is written here, one interpretes it as if (<900 hPa) is the definition of the lower troposphere. It is however meant to be an extra condition.

Rather than referring to a designated definition, we only use the terms to describe our conclusions on what is shown in Fig.6.

**page 13, line 490-493** And what about October?

This is now corrected.

**page 13, line 504** to quantify the effect on higher spatial resolution.

Now changed.

**page 13, line 515** decrease marginally by 2-3% : looking at Table S1, shouldn't it be 1-3%?

Now corrected.

**page 15, line 581 splitting the atmosphere in 3 regions (NH extra-tropics, tropics, SH extra-tropics) is much rougher than "zonally integrated"**

Zonally integrated refers to the cumulative values across all longitudes.

**page 16, line 603** Fig. S13 : is this the correct figure to be referred to?

We have now corrected to text referencing the correct Figure.

**page 16, line 616-617** Isn't it the increase in spatial resolution which helps?

Any improvement in the temporal distribution comes from an using a higher horizontal resolution on which gridded emission estimates are applied. We now change the text to: " *At 1º x 1º significant improvements occur as a result of the better temporal resolution of the emission sources as a result of increasing horizontal resolution.*"

**page 16, line 619** aggregated on a weekly basis does not matter if one looks at seasonal biases; it would have played a role if one also shows correlations.

The value presented is a mean seasonal bias as derived using the bias values from weekly points rather than a single seasonal value.

**page 24, Table 4** Whereas most terms in this table are in units of TgO3 yr−1, it is unclear in what units BO3 and Strat. BO3 are. If these are burdens, one would expect Strat. BO3 to be a larger fraction of BO3, than the values shown here (e.g., on the global scale 80 and 378).

We now include in the Table heading : " *.. with all quantities being given in Tg $O_3$ $yr^{-1}$.*" We take our number from the individual budget files in order to quantify our Strat. BO3. Figure 2 shows that the zonal mean ratio is between 0.05-0.7 in the troposphere, with the higher ratio correlating with lower air pressure thus less mass. Given that through most of the troposphere the ratio changes between 0.05-0.3, 80 Tg seems a reasonable total.

**page 24, Table 4** It is not clear where one can find "The fraction of the tropospheric burden originating from the stratosphere is also given." Does one mean Strat BO3? Are these absolute values Tg, or is it %?

We now remove this from the Table legend.

**page 24, Table 4** Why no % for the NH/SH/tropical STE changes? As these values are not given, the sentence on page 10, line 379 is rather unclear : "The increase in STE in the SH, with an associated decrease in the NH ..."

We now add the percentage differences for each chosen zone.

**page 25, line 744-745** Those with differences <5% are considered to exhibit no discernible change in the bias. An interpretation should not be written in the caption of a figure.

The final print version of this Table will include coloring such that the number of positive and negative biases >5% can be discerned quickly. The policy of GMD is not to include colour in the text in the first instance. The <5% comment relates to the fact that the (black) entries in the table represent stations that are essentially unchanged. Therefore, rather than a definition is pertains to entries in the Table.

**page 31-32, Figs. 4-5** Are the observations also just the 13:00 values?

Yes, please see Sect. 2.3.

**page 35, line 823** during September 2006 : but October 2006 is apparently also shown.

Figure legend now corrected.

---

## Author Response (AR2)

We thank the Editor for assessing our manuscript with respect to the referees suggestions and answer her queries below.

*You have adequately addressed the similarity between the abstract and the conclusions and indeed, included a few sentences on future developments of TM5. However, I'd also be keen that you comment here on the STE findings in relation to the NH flux. How do you propose to investigate this further? What about plans to address the low bias in O₃ deposition. Equally, I don't think you fully addressed the reviewer's point of "What would be the way forward to further improve CTMs?", which requires a more general response than solely focussing on TM5. Given the moderate improvement as a result of increasing resolution and given the reviewer's question above on using 3 x 2 profiles as a priori in retrievals, I would also suggest that you include your response in the conclusions section, should other readers have similar thoughts.*

We now modify the text included at the end of the paper to include some aspects mentioned above.

"Future updates to TM5-MP will most likely focus on developing an online Secondary Organic Aerosol scheme, tropospheric halogen chemistry and incorporating an updated isoprene oxidation scheme involving more intermediate species. It will also be applied in the context of an Earth System Model (EC-EARTH) for allowing future studies concerning chemistry-climate feedbacks. When computing resources allow more expensive simulations can be performed using the applying 60 vertical levels as defined in the ERA-Interim meteorological dataset, approximately doubling the resolution of the simulations presented here. An additional update to improve the STE would be to apply the second-order moments scheme (Prather, 1986), whose application has been shown to capture the seasonality and magnitude of STE exchange to a better degree (Bönisch et al., 2008). In terms of oxidative capacity, one means of reducing the tropospheric near-surface $O_3$ mixing ratios would be improve loss to land-surfaces (Hardacre et al., 2010), although mixing ratios have been shown to be insensitive to the additional loss term to oceans, which is currently missing from many CTM's (Ganzeveld et al., 2009). Our comparisons of $CH_2O$ and $SO_2$ show that there is a significant uncertainty of chemical processes that affect distributions in the pristine marine environment. For instance, the physical process of deposition seems to be under-represented possibly due to too low surface area of the surface i.e. lack of a flat surface. The significant underestimates in $SO_2$ suggest missing biogenic sources terms, therefore more understanding of biogenic emission terms is necessary."

It is not our aim to provide a review of the all of the current issues surrounding CTM models. The diverse range of CTM's available in the community means that different models include different ways of accounting for chemistry and microphysics, therefore commonality between models is not so striking and subject to individual focus as to the potential importance of missing components. This provides a range in the performance skill, with different models exhibiting different biases. What we have therefore done is provide some ideas of how to improve CTM modelling based on our findings presented here.

*I have to agree with the reviewer here. Could I please ask that you include some sensitivity experiments, looking at the impact of reducing/increasing emissions by a fixed amount that represents the uncertainty in the emission inventories? Although you argue that including such experiments would turn the manuscript into a scientific paper rather than a description and validation paper, I believe that a key part of benchmarking a model is understanding the cause of the known biases. Such experiments would add to the current manuscript, would address a major concern by this reviewer, and are well within the scope of a GMD paper.*

As discussed previously by e-mail and in contact with the Lead Editor this step in not feasible because of funding restrictions, where the main body of the work was finished during Summer 2015. Sensitivity experiments are envisaged during future research projects where this version of TM5-MP will be applied.

*Although you argue here that the turbulent vertical transport scheme itself hasn't changed between the 3x2 version and the 1x1 version of TM5, its behaviour will potentially have changed due to the difference in resolution of the input fields. Therefore, could I please ask that you include a discussion on the differences, even if small, in the modelled turbulent vertical transport? In addition, you've added information on scaling the global annual lightning emissions totals between resolutions, differences in the global and vertical distributions of lightning NOx emissions are potentially important and would be worth including in the manuscript. Differences between Tiedtke and the mass flux simulations at 1x1 resolution would also be of relevance here.*

There is currently no way of sampling turbulent fluxes in TM5 and this would require a dedicated body of work for the extraction and analysis of these fields. However, a paper on tropospheric transport in TM5 has recently been published (Koffi et al., 2016), which allows the assessment of the performance of TM5 at $1° \times 1°$, with focus on the diurnal behavior of the boundary layer and vertical transport. We now reference this paper throughout our manuscript to provide more clarity. For lightning $NO_x$ we now include a global horizontal cross-section of Lightning $NO_x$ distributions for the month of July 2006 for the $3° \times 2°$, $1° \times 1°$ and $1° \times 1°$ (Tiedtke) simulations, so the reader can see both the changes in the regional distributions and the totals to the LiNOx emissions. We also include 1-D cross sections of the differences in LiNOx emissions due to changing the convective scheme in TM5 (c.f. new FigS1).

*I think the reviewer makes a very valid point here and some indication of how the meteorological fields themselves differ should be included in the manuscript (you've already included some of that information in your responses). It need not detract from the main focus of the paper, but would rather add to it.*

The performance of the transport in TM5 using both ERA-interim and Tiedtke convective mass-fluxes has recently been published by Koffi et al. (2016), which we now reference throughout the manuscript. There are no real strong conclusions as to which convective scheme performs the best when analysis [222]Rn distributions. We now include further comments regarding ground albedo and also now include an additional diagram related to the global distribution in cloud cover between $3°$ x $2°$ and $1°$ x $1°$ simulations. We then use it during the discussion of the impact of resolution on monthly-mean photolysis rates.

*I think you have partly addressed the reviewer's comments here. However, one aspect which you haven't addressed is "it is not sufficient to quantify the difference between the 3° x 2° and 1° x 1° versions only for the observation locations of the manuscript." In order to fully address this comment, could I please ask you to put the comparisons at the observation locations into a global context?*

We have now included diagrams of the lower tropospheric distribution of $O_3$, $NO_2$, $CH_2O$ and $SO_2$ for the month of May 2006, indicating the regional domains where the validation of each trace species takes place. This allows the reader to see the latitudinal variability in mixing ratios at global scale.

*Again, this comment partly relates to a comment by Reviewer 1 about placing the comparisons with observations within a global context. Could I please ask you to consider including global comparisons between the two resolutions and/or using MOPITT O3 or multi-model output (e.g. ACCENT, ACCMIP) as a somewhat independent global O3 dataset?*

We have now added two additional diagrams to allow the reader to compare the global distributions of $O_3$, $NO_2$, $CH_2O$ and $SO_2$ in the $3°$ x $2°$ and $1°$ x $1°$ TM5-MP simulations, where we also place the regional comparisons in context on a global scale. We also refer to these diagrams during the discussions of each of the trace gas species.

*Page 24, Table 3: Add units for emissions of CO, HCHO, CH₃OH and NOx to Table caption for completeness. Other species in the table whose emissions are NOT in units of TgC/year are DMS, SO₂ and NH₃. Again, can you add emissions units to Table caption? For example, it is currently unclear whether the SO2 emissions are in terms of S or SO2.*

We now put more details regarding the units used for listing the different emissions terms in the Table header.

*Page 9, Lines 334-340: Significant differences exist between Rn profiles simulated with the Tiedtke scheme and those simulated with the convective mass fluxes from ERA-Interim. Apart from saying there are differences, can you use the observations to conclude about which simulation compares better to the observations?*

This had been performed in the recent Koffi et al. (2016) paper, which we now reference throughout the manuscript. In summary, the study was rather inconclusive as to which of the convective schemes performs better with respect to $^{222}$Rn distributions when compared against measurements in the European domain.

*Pg 10, Line 364: Can you change the word "residual" to use "ratio" instead (and be explict about what ratio it is i.e. 3x2/1x1 or vice versa) and add this explicitly to the captions for Figure S6a and S6b?*

Now corrected.

*Pg 10, Line 377: Correct the grammar in "Olszyna et al., 1994) and the also the efficiency of the NOx recycling terms"*

Now corrected.

*Page 25, Table 4: Can you be explicit about the percentage difference in the caption i.e.(ResolutionA-ResolutionB)/Resolution A, rather than referring to it as ResA/ResB?*

We now provide the formulation used for calculating the differences.

*Pg 16, Line 648-649: The sentence "At 1º x 1º significant improvements occur as a result of the better temporal resolution of the emission sources as a result of increasing horizontal resolution." doesn't make sense. Do you mean that the temporal variability of modelled SO2 has improved as a result of increasing horizontal resolution? Or do you mean that you've altered the time profile of the so2 emissions in 1x1?*

We have now re-written the sentence thus: "At 1º x 1º significant improvements occur in the correlations as a result of the better temporal distribution of anthropogenic emission sources."

We thank the Editor for assessing our manuscript with respect to the referees suggestions and answer her queries below.

*You have adequately addressed the similarity between the abstract and the conclusions and indeed, included a few sentences on future developments of TM5. However, I'd also be keen that you comment here on the STE findings in relation to the NH flux. How do you propose to investigate this further? What about plans to address the low bias in $O_3$ deposition. Equally, I don't think you fully addressed the reviewer's point of "What would be the way forward to further improve CTMs?", which requires a more general response than solely focussing on TM5. Given the moderate improvement as a result of increasing resolution and given the reviewer's question above on using 3 x 2 profiles as a priori in retrievals, I would also suggest that you include your response in the conclusions section, should other readers have similar thoughts.*

We now modify the text included at the end of the paper to include some aspects mentioned above.

"Future updates to TM5-MP will most likely focus on developing an online Secondary Organic Aerosol scheme, tropospheric halogen chemistry and incorporating an updated isoprene oxidation scheme involving more intermediate species. It will also be applied in the context of an Earth System Model (EC-EARTH) for allowing future studies concerning chemistry-climate feedbacks. When computing resources allow more expensive simulations can be performed using the applying 60 vertical levels as defined in the ERA-Interim meteorological dataset, approximately doubling the resolution of the simulations presented here. An additional update to improve the STE would be to apply the second-order moments scheme (Prather, 1986), whose application has been shown to capture the seasonality and magnitude of STE exchange to a better degree (Bönisch et al., 2008). In terms of oxidative capacity, one means of reducing the tropospheric near-surface $O_3$ mixing ratios would be improve loss to land-surfaces (Hardacre et al., 2010), although mixing ratios have been shown to be insensitive to the additional loss term to oceans, which is currently missing from many CTM's (Ganzeveld et al., 2009). Our comparisons of $CH_2O$ and $SO_2$ show that there is a significant uncertainty of chemical processes that affect distributions in the pristine marine environment. For instance, the physical process of deposition seems to be under-represented possibly due to too low surface area of the surface i.e. lack of a flat surface. The significant underestimates in $SO_2$ suggest missing biogenic sources terms, therefore more understanding of biogenic emission terms is necessary."

It is not our aim to provide a review of the all of the current issues surrounding CTM models. The diverse range of CTM's available in the community means that different models include different ways of accounting for chemistry and microphysics, therefore commonality between models is not so striking and subject to individual focus as to the potential importance of missing components. This provides a range in the performance skill, with different models exhibiting different biases. What we have therefore done is provide some ideas of how to improve CTM modelling based on our findings presented here.

*I have to agree with the reviewer here. Could I please ask that you include some sensitivity experiments, looking at the impact of reducing/increasing emissions by a fixed amount that represents the uncertainty in the emission inventories? Although you argue that including such experiments would turn the manuscript into a scientific paper rather than a description and validation paper, I believe that a key part of benchmarking a model is understanding the cause of the known biases. Such experiments would add to the current manuscript, would address a major concern by this reviewer, and are well within the scope of a GMD paper.*

As discussed previously by e-mail and in contact with the Lead Editor this step in not feasible because of funding restrictions, where the main body of the work was finished during Summer 2015. Sensitivity experiments are envisaged during future research projects where this version of TM5-MP will be applied.

*Although you argue here that the turbulent vertical transport scheme itself hasn't changed between the 3x2 version and the 1x1 version of TM5, its behaviour will potentially have changed due to the difference in resolution of the input fields. Therefore, could I please ask that you include a discussion on the differences, even if small, in the modelled turbulent vertical transport? In addition, you've added information on scaling the global annual lightning emissions totals between resolutions, differences in the global and vertical distributions of lightning NOx emissions are potentially important and would be worth including in the manuscript. Differences between Tiedtke and the mass flux simulations at 1x1 resolution would also be of relevance here.*

There is currently no way of sampling turbulent fluxes in TM5 and this would require a dedicated body of work for the extraction and analysis of these fields. However, a paper on tropospheric transport in TM5 has recently been published (Koffi et al., 2016), which allows the assessment of the performance of TM5 at $1 \times 1$, with focus on the diurnal behavior of the boundary layer and vertical transport. We now reference this paper throughout our manuscript to provide more clarity. For lightning $NO_x$ we now include a global horizontal cross-section of Lightning $NO_x$ distributions for the month of July 2006 for the $3 \times 2$, $1 \times 1$ and $1 \times 1$ (Tiedtke) simulations, so the reader can see both the changes in the regional distributions and the totals to the LiNOx emissions. We also include 1-D cross sections of the differences in LiNOx emissions due to changing the convective scheme in TM5 (c.f. new FigS1).

*I think the reviewer makes a very valid point here and some indication of how the meteorological fields themselves differ should be included in the manuscript (you've already included some of that information in your responses). It need not detract from the main focus of the paper, but would rather add to it.*

The performance of the transport in TM5 using both ERA-interim and Tiedtke convective mass-fluxes has recently been published by Koffi et al. (2016), which we now reference throughout the manuscript. There are no real strong conclusions as to which convective scheme performs the best when analysis $^{222}$Rn distributions. We now include further comments regarding ground albedo and also now include an additional diagram related to the global distribution in cloud cover between 3 x 2 and 1 x 1 simulations. We then use it during the discussion of the impact of resolution on monthly-mean photolysis rates.

*I think you have partly addressed the reviewer's comments here. However, one aspect which you haven't addressed is "it is not sufficient to quantify the difference between the 3° x 2° and 1° x 1° versions only for the observation locations of the manuscript." In order to fully address this comment, could I please ask you to put the comparisons at the observation locations into a global context?*

We have now included diagrams of the lower tropospheric distribution of $O_3$, $NO_2$, $CH_2O$ and $SO_2$ for the month of May 2006, indicating the regional domains where the validation of each trace species takes place. This allows the reader to see the latitudinal variability in mixing ratios at global scale.

*Again, this comment partly relates to a comment by Reviewer 1 about placing the comparisons with observations within a global context. Could I please ask you to consider including global comparisons between the two resolutions and/or using MOPITT O3 or multi-model output (e.g. ACCENT, ACCMIP) as a somewhat independent global O3 dataset?*

We have now added two additional diagrams to allow the reader to compare the global distributions of $O_3$, $NO_2$, $CH_2O$ and $SO_2$ in the 3° x 2° and 1° x 1° TM5-MP simulations, where we also place the regional comparisons in context on a global scale. We also refer to these diagrams during the discussions of each of the trace gas species.

*Page 24, Table 3: Add units for emissions of CO, HCHO, CH$_3$OH and NOx to Table caption for completeness. Other species in the table whose emissions are NOT in units of TgC/year are DMS, SO$_2$ and NH$_3$. Again, can you add emissions units to Table caption? For example, it is currently unclear whether the SO2 emissions are in terms of S or SO2.*

We now put more details regarding the units used for listing the different emissions terms in the Table header.

*Page 9, Lines 334-340: Significant differences exist between Rn profiles simulated with the Tiedtke scheme and those simulated with the convective mass fluxes from ERA-Interim. Apart from saying there are differences, can you use the observations to conclude about which simulation compares better to the observations?*

This had been performed in the recent Koffi et al. (2016) paper, which we now reference throughout the manuscript. In summary, the study was rather inconclusive as to which of the convective schemes performs better with respect to $^{222}$Rn distributions when compared against measurements in the European domain.

*Pg 10, Line 364: Can you change the word "residual" to use "ratio" instead (and be explict about what ratio it*
*is i.e. 3x2/1x1 or vice versa) and add this explicitly to the captions for Figure S6a and S6b?*

Now corrected.

*Pg 10, Line 377: Correct the grammar in "Olszyna et al., 1994) and the also the efficiency of the NOx*
*recycling terms"*

Now corrected.

*Page 25, Table 4: Can you be explicit about the percentage difference in the caption i.e.(ResolutionA-*
*ResolutionB)/Resolution A, rather than referring to it as ResA/ResB?*

We now provide the formulation used for calculating the differences.

*Pg 16, Line 648-649: The sentence "At 1° x 1° significant improvements occur as a result of the better temporal*
*resolution of the emission sources as a result of increasing horizontal resolution." doesn't make sense. Do you*
*mean that the temporal variability of modelled SO2 has improved as a result of increasing horizontal*
*resolution? Or do you mean that you've altered the time profile of the so2 emissions in 1x1?*

[revised manuscript text omitted]

Formatted Table

| Page 33: [2] Formatted | Williams, Jason (KNMI) | 13/12/2016 15:21:00 |
|---|---|---|

Indent: Left:  -0.19 cm

| Page 33: [3] Formatted | Williams, Jason (KNMI) | 13/12/2016 15:21:00 |
|---|---|---|

Indent: Hanging:  0.27 cm

| Page 33: [4] Formatted | Williams, Jason (KNMI) | 13/12/2016 15:21:00 |
|---|---|---|

Indent: Left:  -0.19 cm

| Page 33: [5] Formatted | Williams, Jason (KNMI) | 13/12/2016 15:21:00 |
|---|---|---|

Indent: Left:  -0.44 cm, First line:  0.25 cm

| Page 33: [6] Formatted | Williams, Jason (KNMI) | 13/12/2016 15:21:00 |
|---|---|---|

Indent: Left:  -0.19 cm

| Page 33: [7] Formatted | Williams, Jason (KNMI) | 13/12/2016 15:21:00 |
|---|---|---|

Indent: Left:  -0.21 cm

| Page 33: [8] Formatted | Williams, Jason (KNMI) | 13/12/2016 15:21:00 |
|---|---|---|

Indent: Left:  -0.22 cm

| Page 33: [9] Formatted | Williams, Jason (KNMI) | 13/12/2016 15:21:00 |
|---|---|---|

Indent: Left:  -0.19 cm

| Page 33: [10] Formatted | Williams, Jason (KNMI) | 13/12/2016 15:21:00 |
|---|---|---|

Indent: Left:  -0.19 cm

| Page 33: [11] Formatted | Williams, Jason (KNMI) | 13/12/2016 15:21:00 |
|---|---|---|

Centered, Indent: Left:  -0.19 cm, First line:  0 cm

| Page 33: [12] Formatted | Williams, Jason (KNMI) | 13/12/2016 15:21:00 |
|---|---|---|

Indent: Left:  -0.44 cm

| Page 33: [13] Formatted | Williams, Jason (KNMI) | 13/12/2016 15:21:00 |
|---|---|---|

Indent: Left:  -0.19 cm

| Page 33: [14] Formatted | Williams, Jason (KNMI) | 13/12/2016 15:21:00 |
|---|---|---|

Indent: Left:  -0.21 cm

| Page 33: [15] Formatted | Williams, Jason (KNMI) | 13/12/2016 15:21:00 |
|---|---|---|

Indent: Left:  -0.22 cm

| Page 33: [16] Formatted | Williams, Jason (KNMI) | 13/12/2016 15:21:00 |
|---|---|---|

Indent: Left:  -0.19 cm

| Page 33: [17] Formatted | Williams, Jason (KNMI) | 13/12/2016 15:21:00 |
|---|---|---|

Indent: Left:  -0.19 cm

| Page 33: [18] Formatted | Williams, Jason (KNMI) | 13/12/2016 15:21:00 |
|---|---|---|

Indent: Left:  -0.19 cm, Right:  -0.19 cm

| Page 33: [19] Formatted | Williams, Jason (KNMI) | 13/12/2016 15:21:00 |
|---|---|---|

Indent: Left:  -0.46 cm, Right:  -0.19 cm

| Page 33: [20] Formatted | Williams, Jason (KNMI) | 13/12/2016 15:21:00 |
|---|---|---|

Indent: Left:  -0.19 cm

| Page 33: [21] Formatted | Williams, Jason (KNMI) | 13/12/2016 15:21:00 |
|---|---|---|

Indent: Left:  -0.21 cm

| Page 33: [22] Formatted | Williams, Jason (KNMI) | 13/12/2016 15:21:00 |
|---|---|---|

Indent: Left: -0.22 cm

| Page 33: [23] Formatted | Williams, Jason (KNMI) | 13/12/2016 15:21:00 |
|---|---|---|

Indent: Left: -0.19 cm

| Page 33: [24] Formatted | Williams, Jason (KNMI) | 13/12/2016 15:21:00 |
|---|---|---|

Indent: Left: -0.19 cm

| Page 33: [25] Formatted | Williams, Jason (KNMI) | 13/12/2016 15:21:00 |
|---|---|---|

Indent: Left: -0.19 cm, Right: -0.19 cm

| Page 33: [26] Formatted | Williams, Jason (KNMI) | 13/12/2016 15:21:00 |
|---|---|---|

Indent: Left: -0.46 cm, Right: -0.19 cm

| Page 33: [27] Formatted | Williams, Jason (KNMI) | 13/12/2016 15:21:00 |
|---|---|---|

Indent: Left: -0.19 cm

| Page 33: [28] Formatted | Williams, Jason (KNMI) | 13/12/2016 15:21:00 |
|---|---|---|

Indent: Left: -0.21 cm

| Page 33: [29] Formatted | Williams, Jason (KNMI) | 13/12/2016 15:21:00 |
|---|---|---|

Indent: Left: -0.22 cm

| Page 33: [30] Formatted | Williams, Jason (KNMI) | 13/12/2016 15:21:00 |
|---|---|---|

Indent: Left: -0.19 cm

| Page 33: [31] Formatted | Williams, Jason (KNMI) | 13/12/2016 15:21:00 |
|---|---|---|

Indent: Left: -0.19 cm

| Page 33: [32] Formatted | Williams, Jason (KNMI) | 13/12/2016 15:21:00 |
|---|---|---|

Indent: Left: -0.19 cm, Right: -0.19 cm

| Page 33: [33] Formatted | Williams, Jason (KNMI) | 13/12/2016 15:21:00 |
|---|---|---|

Indent: Left: -0.46 cm, Right: -0.19 cm

| Page 33: [34] Formatted | Williams, Jason (KNMI) | 13/12/2016 15:21:00 |
|---|---|---|

Indent: Left: -0.19 cm

| Page 33: [35] Formatted | Williams, Jason (KNMI) | 13/12/2016 15:21:00 |
|---|---|---|

Indent: Left: -0.21 cm

| Page 33: [36] Formatted | Williams, Jason (KNMI) | 13/12/2016 15:21:00 |
|---|---|---|

Indent: Left: -0.22 cm

| Page 33: [37] Formatted | Williams, Jason (KNMI) | 13/12/2016 15:21:00 |
|---|---|---|

Indent: Left: -0.19 cm

| Page 33: [38] Formatted | Williams, Jason (KNMI) | 13/12/2016 15:21:00 |
|---|---|---|

Indent: Left: -0.19 cm

| Page 33: [39] Formatted | Williams, Jason (KNMI) | 13/12/2016 15:21:00 |
|---|---|---|

Indent: Left: -0.19 cm, Right: -0.19 cm

| Page 33: [40] Formatted | Williams, Jason (KNMI) | 13/12/2016 15:21:00 |
|---|---|---|

Indent: Left: -0.46 cm, Right: -0.19 cm

| Page 33: [41] Formatted | Williams, Jason (KNMI) | 13/12/2016 15:21:00 |
|---|---|---|

Indent: Left: -0.19 cm

| Page 33: [42] Formatted | Williams, Jason (KNMI) | 13/12/2016 15:21:00 |
|---|---|---|

Indent: Left: -0.21 cm

| Page 33: [43] Formatted | Williams, Jason (KNMI) | 13/12/2016 15:21:00 |

Indent: Left:  -0.22 cm

| Page 33: [44] Formatted | Williams, Jason (KNMI) | 13/12/2016 15:21:00 |

Indent: Left:  -0.19 cm

| Page 33: [45] Formatted | Williams, Jason (KNMI) | 13/12/2016 15:21:00 |

Indent: Left:  -0.19 cm

| Page 33: [46] Formatted | Williams, Jason (KNMI) | 13/12/2016 15:21:00 |

Indent: Left:  -0.19 cm, Right:  -0.19 cm

| Page 33: [47] Formatted | Williams, Jason (KNMI) | 13/12/2016 15:21:00 |

Indent: Left:  -0.46 cm, Right:  -0.19 cm

| Page 33: [48] Formatted | Williams, Jason (KNMI) | 13/12/2016 15:21:00 |

Centered

| Page 33: [49] Formatted | Williams, Jason (KNMI) | 13/12/2016 15:21:00 |

Indent: Left:  -0.19 cm

| Page 33: [50] Formatted | Williams, Jason (KNMI) | 13/12/2016 15:21:00 |

Indent: Left:  -0.21 cm

| Page 33: [51] Formatted | Williams, Jason (KNMI) | 13/12/2016 15:21:00 |

Indent: Left:  -0.22 cm

| Page 33: [52] Formatted | Williams, Jason (KNMI) | 13/12/2016 15:21:00 |

Indent: Left:  -0.19 cm

| Page 33: [53] Formatted | Williams, Jason (KNMI) | 13/12/2016 15:21:00 |

Indent: Left:  -0.19 cm

| Page 33: [54] Formatted | Williams, Jason (KNMI) | 13/12/2016 15:21:00 |

Indent: Left:  -0.19 cm, Right:  -0.19 cm

| Page 33: [55] Formatted | Williams, Jason (KNMI) | 13/12/2016 15:21:00 |

Indent: Left:  -0.46 cm, Right:  -0.19 cm

| Page 33: [56] Formatted | Williams, Jason (KNMI) | 13/12/2016 15:21:00 |

Indent: Left:  -0.19 cm

| Page 33: [57] Formatted | Williams, Jason (KNMI) | 13/12/2016 15:21:00 |

Indent: Left:  -0.21 cm

| Page 33: [58] Formatted | Williams, Jason (KNMI) | 13/12/2016 15:21:00 |

Indent: Left:  -0.22 cm

| Page 33: [59] Formatted | Williams, Jason (KNMI) | 13/12/2016 15:21:00 |

Indent: Left:  -0.19 cm

| Page 33: [60] Formatted | Williams, Jason (KNMI) | 13/12/2016 15:21:00 |

Indent: Left:  -0.19 cm

| Page 33: [61] Formatted | Williams, Jason (KNMI) | 13/12/2016 15:21:00 |

Indent: Left:  -0.19 cm, Right:  -0.19 cm

| Page 33: [62] Formatted | Williams, Jason (KNMI) | 13/12/2016 15:21:00 |

Indent: Left:  -0.46 cm, Right:  -0.19 cm

| Page 33: [63] Formatted | Williams, Jason (KNMI) | 13/12/2016 15:21:00 |

Indent: Left:  -0.19 cm

| Page 33: [64] Formatted | Williams, Jason (KNMI) | 13/12/2016 15:21:00 |
|---|---|---|

Indent: Left: -0.21 cm

| Page 33: [65] Formatted | Williams, Jason (KNMI) | 13/12/2016 15:21:00 |
|---|---|---|

Indent: Left: -0.22 cm

| Page 33: [66] Formatted | Williams, Jason (KNMI) | 13/12/2016 15:21:00 |
|---|---|---|

Indent: Left: -0.19 cm

| Page 33: [67] Formatted | Williams, Jason (KNMI) | 13/12/2016 15:21:00 |
|---|---|---|

Indent: Left: -0.19 cm

| Page 33: [68] Formatted | Williams, Jason (KNMI) | 13/12/2016 15:21:00 |
|---|---|---|

Indent: Left: -0.19 cm, Right: -0.19 cm

| Page 33: [69] Formatted | Williams, Jason (KNMI) | 13/12/2016 15:21:00 |
|---|---|---|

Indent: Left: -0.46 cm, Right: -0.19 cm

| Page 33: [70] Formatted | Williams, Jason (KNMI) | 13/12/2016 15:21:00 |
|---|---|---|

Position: Horizontal: 0.69 cm, Relative to: Margin

| Page 33: [71] Formatted Table | Williams, Jason (KNMI) | 13/12/2016 15:21:00 |
|---|---|---|

Formatted Table

| Page 33: [72] Formatted | Williams, Jason (KNMI) | 13/12/2016 15:21:00 |
|---|---|---|

Indent: Left: -0.37 cm, Hanging: 0.19 cm, Position: Horizontal: 0.69 cm, Relative to: Margin

| Page 33: [73] Formatted | Williams, Jason (KNMI) | 13/12/2016 15:21:00 |
|---|---|---|

Indent: Left: -0.19 cm, Position: Horizontal: 0.69 cm, Relative to: Margin

| Page 33: [74] Formatted | Williams, Jason (KNMI) | 13/12/2016 15:21:00 |
|---|---|---|

Indent: Left: -0.19 cm, Position: Horizontal: 0.69 cm, Relative to: Margin

| Page 33: [75] Formatted | Williams, Jason (KNMI) | 13/12/2016 15:21:00 |
|---|---|---|

Justified, Indent: Left: -0.23 cm, Position: Horizontal: 0.69 cm, Relative to: Margin

| Page 33: [76] Formatted | Williams, Jason (KNMI) | 13/12/2016 15:21:00 |
|---|---|---|

Indent: Left: -0.25 cm, Position: Horizontal: 0.69 cm, Relative to: Margin

| Page 33: [77] Formatted | Williams, Jason (KNMI) | 13/12/2016 15:21:00 |
|---|---|---|

Position: Horizontal: 0.69 cm, Relative to: Margin

| Page 33: [78] Formatted | Williams, Jason (KNMI) | 13/12/2016 15:21:00 |
|---|---|---|

Indent: Left: -0.19 cm, Position: Horizontal: 0.69 cm, Relative to: Margin

| Page 33: [79] Formatted | Williams, Jason (KNMI) | 13/12/2016 15:21:00 |
|---|---|---|

Position: Horizontal: 0.69 cm, Relative to: Margin

| Page 33: [80] Formatted | Williams, Jason (KNMI) | 13/12/2016 15:21:00 |
|---|---|---|

Indent: Left: -0.19 cm, Position: Horizontal: 0.69 cm, Relative to: Margin

| Page 33: [81] Formatted | Williams, Jason (KNMI) | 13/12/2016 15:21:00 |
|---|---|---|

Indent: Left: -0.19 cm, Position: Horizontal: 0.69 cm, Relative to: Margin

| Page 33: [82] Formatted | Williams, Jason (KNMI) | 13/12/2016 15:21:00 |
|---|---|---|

Indent: Left: -0.19 cm, Position: Horizontal: 0.69 cm, Relative to: Margin

| Page 33: [83] Formatted | Williams, Jason (KNMI) | 13/12/2016 15:21:00 |
|---|---|---|

Indent: Left: -0.25 cm, Right: -0.18 cm, Position: Horizontal: 0.69 cm, Relative to: Margin

| Page 33: [84] Formatted | Williams, Jason (KNMI) | 13/12/2016 15:21:00 |
|---|---|---|

Position: Horizontal: 0.69 cm, Relative to: Margin

| Page 33: [85] Formatted | Williams, Jason (KNMI) | 13/12/2016 15:21:00 |

Indent: Left: -0.19 cm, Position: Horizontal: 0.69 cm, Relative to: Margin

| Page 33: [86] Formatted | Williams, Jason (KNMI) | 13/12/2016 15:21:00 |

Position: Horizontal: 0.69 cm, Relative to: Margin

| Page 33: [87] Formatted | Williams, Jason (KNMI) | 13/12/2016 15:21:00 |

Indent: Left: -0.19 cm, Position: Horizontal: 0.69 cm, Relative to: Margin

| Page 33: [88] Formatted | Williams, Jason (KNMI) | 13/12/2016 15:21:00 |

Indent: Left: -0.19 cm, Position: Horizontal: 0.69 cm, Relative to: Margin

| Page 33: [89] Formatted | Williams, Jason (KNMI) | 13/12/2016 15:21:00 |

Indent: Left: -0.19 cm, Position: Horizontal: 0.69 cm, Relative to: Margin

| Page 33: [90] Formatted | Williams, Jason (KNMI) | 13/12/2016 15:21:00 |

Indent: Left: -0.25 cm, Right: -0.18 cm, Position: Horizontal: 0.69 cm, Relative to: Margin

| Page 33: [91] Formatted | Williams, Jason (KNMI) | 13/12/2016 15:21:00 |

Position: Horizontal: 0.69 cm, Relative to: Margin

| Page 33: [92] Formatted | Williams, Jason (KNMI) | 13/12/2016 15:21:00 |

Indent: Left: -0.19 cm, Position: Horizontal: 0.69 cm, Relative to: Margin

| Page 33: [93] Formatted | Williams, Jason (KNMI) | 13/12/2016 15:21:00 |

Position: Horizontal: 0.69 cm, Relative to: Margin

| Page 33: [94] Formatted | Williams, Jason (KNMI) | 13/12/2016 15:21:00 |

Indent: Left: -0.19 cm, Position: Horizontal: 0.69 cm, Relative to: Margin

| Page 33: [95] Formatted | Williams, Jason (KNMI) | 13/12/2016 15:21:00 |

Indent: Left: -0.19 cm, Position: Horizontal: 0.69 cm, Relative to: Margin

| Page 33: [96] Formatted | Williams, Jason (KNMI) | 13/12/2016 15:21:00 |

Indent: Left: -0.19 cm, Position: Horizontal: 0.69 cm, Relative to: Margin

| Page 33: [97] Formatted | Williams, Jason (KNMI) | 13/12/2016 15:21:00 |

Indent: Left: -0.25 cm, Right: -0.18 cm, Position: Horizontal: 0.69 cm, Relative to: Margin

| Page 33: [98] Formatted | Williams, Jason (KNMI) | 13/12/2016 15:21:00 |

Position: Horizontal: 0.69 cm, Relative to: Margin

| Page 33: [99] Formatted | Williams, Jason (KNMI) | 13/12/2016 15:21:00 |

Indent: Left: -0.19 cm, Position: Horizontal: 0.69 cm, Relative to: Margin

| Page 33: [100] Formatted | Williams, Jason (KNMI) | 13/12/2016 15:21:00 |

Position: Horizontal: 0.69 cm, Relative to: Margin

| Page 33: [101] Formatted | Williams, Jason (KNMI) | 13/12/2016 15:21:00 |

Indent: Left: -0.19 cm, Position: Horizontal: 0.69 cm, Relative to: Margin

| Page 33: [102] Formatted | Williams, Jason (KNMI) | 13/12/2016 15:21:00 |

Indent: Left: -0.19 cm, Position: Horizontal: 0.69 cm, Relative to: Margin

| Page 33: [103] Formatted | Williams, Jason (KNMI) | 13/12/2016 15:21:00 |

Indent: Left: -0.19 cm, Position: Horizontal: 0.69 cm, Relative to: Margin

| Page 33: [104] Formatted | Williams, Jason (KNMI) | 13/12/2016 15:21:00 |

Indent: Left: -0.25 cm, Right: -0.18 cm, Position: Horizontal: 0.69 cm, Relative to: Margin

| Page 33: [105] Formatted | Williams, Jason (KNMI) | 13/12/2016 15:21:00 |

Position: Horizontal:  0.69 cm, Relative to: Margin

Page 33: [106] Formatted          Williams, Jason (KNMI)          13/12/2016 15:21:00

Indent: Left:  -0.19 cm, Position: Horizontal:  0.69 cm, Relative to: Margin

Page 33: [107] Formatted          Williams, Jason (KNMI)          13/12/2016 15:21:00

Position: Horizontal:  0.69 cm, Relative to: Margin

Page 33: [108] Formatted          Williams, Jason (KNMI)          13/12/2016 15:21:00

Indent: Left:  -0.19 cm, Position: Horizontal:  0.69 cm, Relative to: Margin

Page 33: [109] Formatted          Williams, Jason (KNMI)          13/12/2016 15:21:00

Indent: Left:  -0.19 cm, Position: Horizontal:  0.69 cm, Relative to: Margin

Page 33: [110] Formatted          Williams, Jason (KNMI)          13/12/2016 15:21:00

Indent: Left:  -0.19 cm, Position: Horizontal:  0.69 cm, Relative to: Margin

Page 33: [111] Formatted          Williams, Jason (KNMI)          13/12/2016 15:21:00

Indent: Left:  -0.25 cm, Right:  -0.18 cm, Position: Horizontal:  0.69 cm, Relative to: Margin

Page 33: [112] Formatted          Williams, Jason (KNMI)          13/12/2016 15:21:00

Position: Horizontal:  0.69 cm, Relative to: Margin

Page 33: [113] Formatted          Williams, Jason (KNMI)          13/12/2016 15:21:00

Indent: Left:  -0.19 cm, Position: Horizontal:  0.69 cm, Relative to: Margin

Page 33: [114] Formatted          Williams, Jason (KNMI)          13/12/2016 15:21:00

Position: Horizontal:  0.69 cm, Relative to: Margin

Page 33: [115] Formatted          Williams, Jason (KNMI)          13/12/2016 15:21:00

Indent: Left:  -0.19 cm, Position: Horizontal:  0.69 cm, Relative to: Margin

Page 33: [116] Formatted          Williams, Jason (KNMI)          13/12/2016 15:21:00

Indent: Left:  -0.19 cm, Position: Horizontal:  0.69 cm, Relative to: Margin

Page 33: [117] Formatted          Williams, Jason (KNMI)          13/12/2016 15:21:00

Indent: Left:  -0.19 cm, Position: Horizontal:  0.69 cm, Relative to: Margin

Page 33: [118] Formatted          Williams, Jason (KNMI)          13/12/2016 15:21:00

Indent: Left:  -0.25 cm, Right:  -0.18 cm, Position: Horizontal:  0.69 cm, Relative to: Margin